# Latent Guided Sampling for Combinatorial Optimization

## Abstract

Combinatorial Optimization problems are widespread in domains such as logistics, manufacturing, and drug discovery, yet their NP-hard nature makes them computationally challenging. Recent Neural Combinatorial Optimization (NCO) methods leverage deep learning to learn policies for constructing solutions, trained via Supervised or Reinforcement Learning. While promising, these approaches often rely on task-specific augmentations, perform poorly on out-of-distribution instances, and lack robust inference mechanisms. Moreover, existing latent space models either require labeled data or use an instance-independent latent distribution. In this work, we propose LGS-Net, a novel latent space model that conditions on problem instances, and introduce an efficient inference method, Latent Guided Sampling (LGS), based on Markov Chain Monte Carlo and Stochastic Approximation. We show that the iterations of our method form a time-inhomogeneous Markov Chain and provide rigorous theoretical convergence guarantees. Empirical results on benchmark routing tasks show that our method achieves state-of-the-art performance among NCO baselines.

## 1 Introduction

Combinatorial Optimization (CO) consists of finding the best solution from a discrete set of possibilities by optimizing a given objective function subject to constraints. It has widespread applications across various domains, including vehicle routing (Veres & Moussa, 2019), production planning (Dolgui et al., 2019), and drug discovery (Liu et al., 2017). However, its NP-hard nature and the complexity of many problem variants make solving CO problems highly challenging. Traditional heuristic methods (e.g., (Kirkpatrick et al., 1983; Glover, 1989; Mladenović & Hansen, 1997)) rely on hand-crafted rules to guide the search, providing near-optimal solutions with significantly lower computational costs. Inspired by the success of deep learning in computer vision (Krizhevsky et al., 2012; He et al., 2016) and natural language processing (Vaswani et al., 2017; Devlin et al., 2019), recent years have seen a surge in Neural Combinatorial Optimization (NCO) approaches for solving CO problems, including the Travelling Salesman Problem (TSP) and the Capacitated Vehicle Routing Problem (CVRP).

These methods leverage neural networks to learn a policy that generates solutions, trained via either Supervised Learning (SL) (Vinyals et al., 2015; Joshi et al., 2019; Hottung et al., 2021; Fu et al., 2021; Joshi et al., 2022; Kool et al., 2022) or Reinforcement Learning (RL) (Bello et al., 2017; Nazari et al., 2018; Kool et al., 2019; Chen & Tian, 2019; Kwon et al., 2020; Hottung & Tierney, 2020; Grinsztajn et al., 2023; Chalumeau et al., 2023). SL-based methods often struggle to obtain sufficient high-quality labeled data, whereas RL-based approaches can surpass them by exploring solutions autonomously. Despite their success on in-distribution problem instances, these methods often generalize poorly to out-of-distribution cases, limiting their applicability in real-world scenarios. Moreover, once a policy is trained, inference typically relies on relatively simple strategies such as stochastic sampling (Kool et al., 2019; Kwon et al., 2020), beam search (Steinbiss et al., 1994), or Monte Carlo Tree Search (Browne et al., 2012). A popular alternative, based on online fine-tuning (Bello et al., 2017; Hottung et al., 2022), is to actively adapt the policy for each new problem instance. However, this approach introduces significant computational and practical challenges.

Meanwhile, latent space models have proven effective across diverse tasks, including image generation (Rombach et al., 2022), text generation (Bowman et al., 2016), anomaly detection (An & Cho,

2015), and molecule design (Gómez-Bombarelli et al., 2018). More recently, Hottung et al. (2021); Chalumeau et al. (2023) have explored learning a continuous latent space for discrete routing problems. Importantly, Chalumeau et al. (2023) have shown that RL-based latent methods can achieve competitive results without relying on the augmentation trick (Kwon et al., 2020), which is used in several prior inference methods. This augmentation strategy enhances performance by generating variations of the same problem, such as rotating the coordinates, but is task-specific and only applicable to certain routing problems in Euclidean space. However, existing latent space methods still have important limitations: the approach of Hottung et al. (2021) relies on labeled training data for SL, whereas Chalumeau et al. (2023) lacks instance-dependent structure in its latent space and therefore does not adapt to the specific problem instance. In contrast, we introduce a latent space model that conditions the latent representation on problem instances and is trained with RL, addressing these limitations and enabling more effective latent space optimization. We further propose a novel guided inference method designed for latent-based models, based on Markov Chain Monte Carlo (MCMC) and Stochastic Approximation (SA). Our method comes with theoretical convergence guarantees and outperforms most state-of-the-art NCO methods.

More precisely, our contributions are summarized as follows.

- We introduce LGS-Net, a novel latent space model for Neural Combinatorial Optimization that does not require labeled data, learns a structured, instance-conditioned latent representation, and is supported by a rigorous mathematical framework.
- We propose LGS, a new inference scheme based on interacting MCMC, which jointly samples from the learned distribution while updating parameters via Stochastic Approximation. Moreover, we establish that its iterates form a joint time-inhomogeneous Markov Chain over the latent and solution spaces, converging to the desired target distribution.
- We empirically show that our inference method is effective in low-budget regimes, where gradient-based approaches tend to slow down. Furthermore, we establish that it achieves state-of-the-art performance among NCO methods, consistently outperforming existing techniques across diverse problem types, both with and without the augmentation trick.

## 2 RELATED WORK

**Neural Combinatorial Optimization.** The application of neural networks to CO problems dates back to Hopfield & Tank (1985), who used a Hopfield network for small TSP instances. With advancements in deep learning, a major breakthrough came with Pointer Networks (Vinyals et al., 2015), inspired by sequence-to-sequence models (Sutskever et al., 2014). Pointer Networks enabled variable-length outputs but relied on supervised training and thus could not surpass the quality of the provided solutions. To address this limitation, Bello et al. (2017) adapted the model for RL. For the CVRP, Nazari et al. (2018) extended Pointer Networks with element-wise projections, followed by the Attention Model (AM) of Kool et al. (2019), based on transformers (Vaswani et al., 2017). Since then, numerous AM variants have been proposed (Kwon et al., 2020; Xin et al., 2020; Kim et al., 2021; Xin et al., 2021; Kwon et al., 2021; Hottung et al., 2025). For instance, Kwon et al. (2020) introduced POMO, an attention-based model that incorporates a more robust learning and inference strategy grounded in multiple optimal policies. In addition, graph neural network approaches have been explored, notably graph embeddings (Khalil et al., 2017), attention networks (Deudon et al., 2018), and convolutional networks (Joshi et al., 2019) for the TSP.

Several approaches combine heuristic algorithms, such as local search (Papadimitriou & Steiglitz, 1998; De Moura & Bjørner, 2008), with machine learning techniques to tackle routing problems. For example, Chen & Tian (2019); Lu et al. (2019) propose an RL-based improvement method that iteratively selects a region of the solution and applies a local heuristic determined by a trainable policy. To improve the generalization ability of constructive methods during inference, various strategies have been introduced, such as Efficient Active Search (EAS) (Hottung et al., 2022) and Simulation-guided Beam Search (SGBS) (Choo et al., 2022). Notably, EAS builds on POMO by fine-tuning a subset of model parameters at inference time using Gradient Descent, in contrast to Active Search (Bello et al., 2017), which updates all model parameters. More recently, generalization-boosting methods have also been explored (Gao et al., 2024; Zhou et al., 2024). Recent state-of-the-art NCO methods include memory-enhanced approaches (Chalumeau et al., 2025) as well as latent space and search-based methods such as Chalumeau et al. (2023); Hottung et al. (2022).

Among the latent space models designed to map discrete routing problems to continuous spaces, Hottung et al. (2021) used a conditional variational autoencoder (CVAE) (Sohn et al., 2015) that maps solutions to a continuous latent space. However, their approach relies on supervised training, which is hindered by the significant cost of acquiring high-quality labeled data. To overcome this limitation, Chalumeau et al. (2023) proposed COMPASS, an RL-based approach that learns a latent space on top of a policy. However, the latent distribution in COMPASS lacks instance-dependent structure: it is defined via a fixed prior in latent space and does not adapt to the specific problem instance, making it conceptually similar to a GAN (Goodfellow et al., 2014) without a discriminator. In contrast, we introduce a latent space model that conditions the latent representation on problem instances and is trained with RL from cost feedback, without requiring labeled data. Our approach can be interpreted as a VAE with a modified encoder that conditions only on problem instances, rather than on both problem instances and solutions. Furthermore, we propose an inference method based on MCMC and SA, rather than using Differential Evolution (DE) (Storn & Price, 1997) or Covariance Matrix Adaptation (CMA) (Hansen & Ostermeier, 2001), which were used in previous works on latent space models.

**Markov Chain Monte Carlo.** The Metropolis-Hastings algorithm (Metropolis et al., 1953; Hastings, 1970) is one of the most widely used MCMC methods. Since its introduction, numerous variants have been developed, including the Gibbs sampler (Geman & Geman, 1984) and Hamiltonian Monte Carlo (Duane et al., 1987). These methods have been theoretically studied, particularly in terms of geometric ergodicity, under well-established drift and minorization conditions (Meyn & Tweedie, 1994; Baxendale, 2005). A crucial factor influencing the performance of MCMC methods is the choice of the proposal distribution, which significantly affects the convergence rate (Gelman et al., 1997). Traditionally, tuning the proposal distribution relies on heuristics and manual adjustments. To address this limitation, adaptive MCMC methods have been developed, where the proposal distribution is adjusted dynamically based on previous samples (Haario et al., 2001). The ergodicity of adaptive MCMC methods incorporating Stochastic Approximation (Robbins & Monro, 1951) has been studied by Andrieu & Atchadé (2007); Andrieu & Moulines (2006). Beyond adaptive MCMC, time-inhomogeneous MCMC methods, which extend adaptive MCMC, have also been explored (Andrieu et al., 2001; Douc et al., 2004). In our setting, both the proposal and target distributions evolve dynamically, and existing results are therefore insufficient to establish convergence guarantees. Indeed, prior analyses (Andrieu et al., 2001; Douc et al., 2004) focus on continuous spaces and rely heavily on regularity assumptions, such as strict convexity and coercivity of the objective function, which do not hold in our case. To address this gap, we introduce new results for time-inhomogeneous MCMC methods, enabling us to derive convergence guarantees.

## 3 Notation and Background

### 3.1 Notation

In the following, for all distribution $\mu$ (resp. probability density $p$) we write $\mathbb{E}_\mu$ (resp. $\mathbb{E}_p$) the expectation under $\mu$ (resp. under $p$). We may also write $\mathbb{E}_{x \sim \mu}$ for the expectation under $\mu$. Given a measurable space $(\mathsf{X}, \mathcal{X})$, where $\mathcal{X}$ is a countably generated $\sigma$-algebra, let $\mathsf{F}(\mathsf{X})$ denote the set of all measurable functions defined on $(\mathsf{X}, \mathcal{X})$. Let $\mathsf{M}(\mathsf{X})$ be the set of $\sigma$-finite measures on $(\mathsf{X}, \mathcal{X})$, and $\mathsf{M}_1(\mathsf{X}) \subset \mathsf{M}(\mathsf{X})$ the probability measures. For all $f \in \mathsf{F}(\mathsf{X})$ and $\mu \in \mathsf{M}(\mathsf{X})$, we write $\mu(f) = \int f(x)\mu(dx)$. For a Markov kernel $P$ on $(\mathsf{X}, \mathcal{X})$ and $\mu \in \mathsf{M}_1(\mathsf{X})$, the composition $\mu P$ is defined as $\mu P : \mathcal{X} \ni A \mapsto \int \mu(dx)P(x, dy)\mathbf{1}_A(y)$. For probability measures $\mu$ and $\nu$ defined on the same measurable space, the Total Variation (TV) is defined as $\|\mu - \nu\|_{\mathrm{TV}} := \sup_{A \in \mathcal{X}} |\mu(A) - \nu(A)|$. The $\mathrm{L}_2$-norm of a random variable $X$ is defined as $\|X\|_{\mathrm{L}_2} := \left(\mathbb{E}[\|X\|^2]\right)^{1/2}$. The Hadamard product of two vectors $u$ and $v$ is denoted by $u \odot v$. For a sequence $(a_m)_{m \in \mathbb{N}}$, and all $u \leq v$, we write $a_{u:v} = \{a_u, \ldots, a_v\}$. Table 3 provides a summary of the notations used throughout the paper for ease of reference.

### 3.2 Problem Setting

In a Combinatorial Optimization problem, the objective is to determine the best assignment of discrete variables that satisfies the constraints of the problem. Let $x$ represent a given problem instance and $y$ denote a solution. An instance $x = \{\mathsf{x}_i\}_{i=1}^n \in \mathsf{X} \subset \mathbb{R}^{n \times d_x}$ consists of a set of $n$ nodes, each

represented by a feature vector $x_i \in \mathbb{R}^{d_x}$, which encodes relevant information about the node. For a given instance $x$, we aim to find a solution $y^*$ that minimizes the associated cost function $C$:

$$y^* \in \underset{y \in \mathsf{Y}}{\arg\min}\, C(y, x) \,, \tag{1}$$

where $\mathsf{Y}$ denotes the discrete set of all feasible solutions for the given problem $x$. This setting covers problems such as the TSP, CVRP, Knapsack, and Job Scheduling. For instance, in TSP, $x$ represents the coordinates of all nodes and $\mathsf{Y}$ consists of all possible node permutations. In CVRP, $x$ additionally includes demands, and $\mathsf{Y}$ comprises all feasible routes satisfying the capacity constraints. In both cases, the cost corresponds to the cumulative distance of the route. Further details on both problems are provided in Appendix A.1.

### 3.3 CONSTRUCTIVE NCO METHODS

Constructive NCO methods (Vinyals et al., 2015; Bello et al., 2017; Nazari et al., 2018; Kool et al., 2019; Kwon et al., 2020) generate solutions sequentially using a stochastic policy $p_\theta(y|x)$, which defines the probability of selecting a solution $y$ given a problem instance $x$. This policy is parameterized by $\theta \in \Theta$, where $\Theta$ is a parameter space, and factorized as:

$$p_\theta(y|x) = \prod_{t=1}^{T} p_\theta(y_t|x, y_{1:t-1}) \,,$$

with the convention $p_\theta(y_1|x, y_{1:0}) = p_\theta(y_1|x)$, where $y_t \in \{0, \ldots, n\}$ is the selected node at step $t$, $y_{1:t-1}$ denotes the sequence of nodes selected up to step $t-1$, and $T$ is the total number of decoding steps. Following Bello et al. (2017), writing $\mathbb{P}_x$ the distribution of the problem instances, the policy is trained via RL by minimizing an empirical estimate of the expected cost:

$$J(\theta) = \mathbb{E}_{x \sim \mathbb{P}_x, y \sim p_\theta(.|x)} \left[ C(y, x) \right] \,.$$

where $C(y, x)$ denotes the tour cost. This objective is optimized using RL techniques, such as REINFORCE (Williams, 1992) or Actor-Critic methods (Konda & Tsitsiklis, 1999).

## 4 LATENT GUIDED SAMPLING

### 4.1 MODEL

The proposed model introduces a continuous latent search space for routing problems similar to Hottung et al. (2021); Chalumeau et al. (2023), which can be efficiently explored by any continuous optimization method at inference time. To achieve this, we model the target distribution that generates a solution $y$ given a problem instance $x$ as a latent-variable model:

$$p_{\phi,\theta}(y|x) = \int p_\phi(z|x) p_\theta(y|x, z) \, \mathrm{d}z \,.$$

The encoder $p_\phi(z|x)$ maps the problem instance $x$ to a continuous $d_z$-dimensional latent representation $z$. The decoder $p_\theta(y|x, z)$ then generates a solution $y$ conditioned on both $z$ and $x$. Both the encoder and decoder are parameterized by neural networks with learnable parameters $\phi \in \Phi$ and $\theta \in \Theta$, respectively. The probability of the decoder generating a solution $y$ is then factorized as:

$$p_\theta(y|x, z) = \prod_{t=1}^{T} p_\theta(y_t|y_{1:t-1}, z, x) \,,$$

where $y_t \in \{0, \ldots, n\}$ is the selected node at step $t$, and $y_{1:t-1}$ denotes the sequence of nodes selected up to step $t-1$. It is important to note that the encoder differs from the variational distribution $q_\phi(z|x, y)$ (Kingma & Welling, 2014); it corresponds to a CVAE-Opt (Hottung et al., 2021), which requires labeled data for training.

Our encoder architecture follows the general structure of Kool et al. (2019) but includes additional layers to compute the parameters of the encoder distribution. To compute output probabilities, we use a single decoder layer with multi-head attention to enable efficient inference. At step $0 \le t \le T$, for all $i \in \{0, \ldots, n\}$, this layer computes the probability $p_\theta(y_t = i|y_{1:t-1}, x, z)$ while masking nodes that lead to infeasible solutions. Details on the encoder and decoder are provided in Appendix A.2.

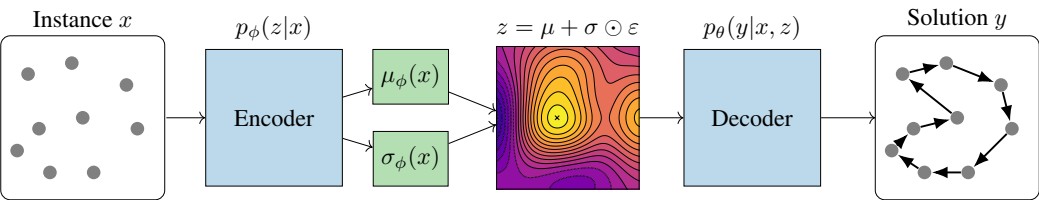

Figure 1: Our Latent Model Architecture with the Gaussian Reparameterization Trick

## 4.2 TRAINING

During training, our objective is to minimize the cost $C$ while encouraging diversity in the generated solutions to improve inference efficiency. To achieve this, we introduce an entropic regularization term (Ziebart et al., 2008; Haarnoja et al., 2018) controlled by a parameter $\beta$. The training loss is given by:

$$\mathcal{L}(\phi, \theta; x) = \sum_{k=1}^{K} \mathbb{E}_{z^k \sim p_\phi(\cdot|x)} \left[ \mathbb{E}_{y^k \sim p_\theta(\cdot|x,z^k)} \left[ w^k C(y^k, x) \right] + \beta \mathcal{H}(p_\theta(\cdot \mid x, z^k)) \right] \ . \tag{2}$$

where $\mathcal{H}(p_\theta(\cdot|x,z))$ denotes the entropy of the conditional decoder distribution $p_\theta(y|x,z)$. The loss in (2) can be interpreted as a cost-weighted extension of Maximum Entropy RL (Ziebart et al., 2008), with weights $w^k = \exp(-C(y^k, x)/\tau)$ acting as importance factors that emphasize low-cost solutions. This formulation is inspired by IWAE (Burda et al., 2016) and the "best-of-many" objective (Bhattacharyya et al., 2018; Grinsztajn et al., 2023), interpolating between uniform weighting (large $\tau$) that encourages exploration and near-greedy weighting (small $\tau$) that prioritizes exploitation. In practice, $\tau$ is gradually decreased to encourage broad exploration early on, followed by concentration on promising regions of the latent space. A more detailed derivation and the complete training procedure are provided in Appendix A.3.

## 4.3 INFERENCE

Any search procedure strategy can be used at inference time to find the best solution while keeping the computational cost manageable. Possible approaches include evolutionary algorithms such as DE and CMA-ES, as well as learnable methods like Active Search (Bello et al., 2017) and Efficient Active Search (Hottung et al., 2022). We formulate inference as a sampling problem: given an instance $x$, and the learned encoder and decoder parameters $\theta$ and $\phi$, our goal is to sample from the distribution:

$$\pi_\theta(y|x) \propto \int \pi_\theta(z, y|x) \mathrm{d}z \ , \quad \text{where} \quad \pi_\theta(z, y|x) \propto p_\phi(z|x) p_\theta(y|z, x) e^{-\lambda C(y,x)} \ . \tag{3}$$

We omit the explicit dependence on $\phi$ since $p_\phi$ serves as a prior over latent variables. To favor lower-cost solutions, we introduce the reweighting factor $\exp(-\lambda C(y, x))$, where $\lambda$ controls the trade-off between likelihood and cost. However, incorporating this reweighting renders the distribution in (3) intractable to sample from directly. While methods such as MCMC (Hastings, 1970) can be used to approximate it, they are often inefficient in practice. To address this challenge, we propose Latent Guided Sampling (LGS), a novel inference method designed for latent space models. LGS constructs sequences of latent samples and corresponding solutions by running multiple interacting Markov Chains to encourage better exploration, while simultaneously updating the model parameters $\theta$ via Stochastic Approximation to minimize the following test objective:

$$\mathcal{L}_{test}(\theta; x) = \mathbb{E}_{\pi_\theta(\cdot|x)} \left[ C(y, x) \right] \ . \tag{4}$$

Since the solution quality depends on the trained parameters, it is natural to iteratively update $\theta$. In contrast, the encoder parameters $\phi$ are kept fixed to avoid the high computational cost associated with backpropagating through them. The gradient is estimated using previously sampled latent variables:

$$H_\theta \left( x, \{(z^k, y^k)\}_{k=1}^{K} \right) = \frac{1}{K} \sum_{k=1}^{K} \left( C(y^k, x) - b(x) \right) \nabla_\theta \log p_\theta(y^k|x, z^k) \ , \tag{5}$$

where $b(x)$, defined in (9), serves as a baseline to reduce the variance. The proposed method is detailed in Algorithm 1. The value of $K$ should be selected to balance stability (reducing variance in gradient estimates), effective exploration of the latent space, and computational efficiency. Although standard gradient updates are used in Algorithm 1, any other optimizer such as Adam (Kingma & Ba, 2015) could be used. In the next section, we establish that the sequence generated by our algorithm forms a Markov Chain and converges to the target distribution, depending on the optimality of $\theta$.

---

**Algorithm 1** Latent Guided Sampling

---

1: **Input:** Problem instance $x$, pretrained encoder $p_\phi$, pretrained decoder $p_{\theta_0}$, proposal distribution $q$, number of particles $K$, number of iterations $M$, cost function $C$, and temperature $\lambda$.
2: **Initialize:**
3: Sample initial particles: $z_0^k \sim p_\phi(\cdot|x)$ for all $k = 1, \ldots, K$.
4: **for** $m = 0, 1, \ldots, M - 1$ **do**
5:      Propagate new particles: $\tilde{z}_{m+1}^k \sim q(\cdot|z_m)$ for all $k = 1, \ldots, K$.
6:      Generate candidate solutions: $y_{m+1}^k \sim p_{\theta_m}(\cdot|\tilde{z}_{m+1}^k, x)$ for all $k = 1, \ldots, K$.
7:      Compute acceptance probabilities:

$$\alpha_{m+1}^k = \min\left(1, \frac{p_\phi(\tilde{z}_{m+1}^k|x)}{p_\phi(z_m^k|x)} \times e^{-\lambda(C(y_{m+1}^k, x) - C(y_m^k, x))}\right).$$

8:      Accept $z_{m+1}^k = \tilde{z}_{m+1}^k$ with probability $\alpha_{m+1}^k$, otherwise $z_{m+1}^k = z_m^k$ for all $k = 1, \ldots, K$.
9:      Compute the gradient estimate $H_{\theta_m}\left(x, \{(z_{m+1}^k, y_{m+1}^k)\}_{k=1}^K\right)$ using (5).
10:      Update parameters: $\theta_{m+1} = \theta_m - \gamma_{m+1} H_{\theta_m}\left(x, \{(z_{m+1}^k, y_{m+1}^k)\}_{k=1}^K\right)$.
11: **end for**

---

## 5    THEORETICAL RESULTS

Let $Z \subset \mathbb{R}^{d_z}$ be the latent space and $Y$ the solution space. In this section, we present theoretical results on our inference method described in Algorithm 1.

### 5.1    CONVERGENCE ANALYSIS FOR FIXED $\theta$

We first analyze convergence in the absence of the Stochastic Approximation step, that is, without updating the parameter $\theta$ (line 10 in Algorithm 1). Specifically, we show that the sequences generated by our algorithm form a Markov Chain and exhibit geometric convergence to the joint target distribution defined in (3).

**Proposition 5.1.** *The sequence $\{(Z_m, Y_m) : m \in \mathbb{N}\}$ generated by Algorithm 1 for a fixed parameter $\theta$ forms a Markov Chain with transition kernel $P_\theta$.*

The explicit expression for $P_\theta$ is provided in Proposition C.1. Consider the following assumptions.

**Assumption 1.** The cost function $C$ is bounded. For all $\phi \in \Phi$, the encoder distribution $p_\phi$ is positive. Furthermore, the decoder probability satisfies for all $1 \leq t \leq T$, and all $(y_{1:t-1}, x, z)$, $p_\theta(y_t = i|y_{1:t-1}, x, z) > 0$ for all feasible nodes $i \in \{0, \ldots, n\}$.

Since $Y$ is discrete, the boundedness of $C$ is a natural assumption, analogous to bounded rewards in RL (Fallah et al., 2021). A Gaussian choice for $p_\phi$ is standard, enabling the reparameterization trick (Kingma & Welling, 2014) for efficient gradient backpropagation. To ensure positivity of the decoder, a common approach is to use a softmax function in the final layer of the neural network, which is a standard practice in most architectures.

**Assumption 2.** The proposal density $q$ is positive and symmetric.

Assumption 2 on the proposal density $q$ is commonly used in various sampling-based methods, such as Importance Sampling and MCMC (Douc et al., 2004). It holds for a wide range of distributions, including Gaussian, Laplace, and Uniform.

**Theorem 5.2.** *Let Assumptions 1 and 2 hold. Then, the Markov kernel $P_\theta$ admits a unique invariant probability measure $\pi_\theta$, defined in (3). There exist constants $\rho_1, \rho_2 \in (0, 1)$ and $\kappa_1, \kappa_2 \in \mathbb{R}_+$ such*

*that for all $\mu \in \mathsf{M}_1(\mathsf{Z} \times \mathsf{Y})$, $\theta \in \Theta$, and $m \in \mathbb{N}$,*

$$\|\mu P_\theta^m - \pi_\theta\|_{TV} \le \kappa_1 \rho_1^m \quad and \quad \|\mu P_\theta^m - \pi_\theta\|_{\mathrm{L}_2} \le \kappa_2 \rho_2^m \|\mu - \pi_\theta\|_{\mathrm{L}_2} .$$

Theorem 5.2 shows that the Markov Chain generated by our algorithm with a fixed $\theta$ converges geometrically to the joint target distribution in both Total Variation and $\mathrm{L}_2$-distance. Specifically, when the initial distribution is $\mu = p_\phi p_\theta$, the theorem guarantees rapid mixing of the Markov Chain, provided that the model is well-trained.

## 5.2 CONVERGENCE ANALYSIS FOR ADAPTIVE $\theta$

Incorporating SA steps introduce a time-inhomogeneous Markov Chain, where both the Markov kernel and the target distribution evolve dynamically. While only a few results are available in this setting (Douc et al., 2004), we establish new convergence results for time-inhomogeneous Markov Chains under assumptions adapted to our setting, without relying on strict convexity or coercivity of the cost function $C$. For simplicity, we only present the results relating to our setting here; more general results are provided in Appendix D.1. To analyze the convergence, we introduce the following additional assumptions.

**Assumption 3.** There exists $L \in \mathsf{F}(\mathsf{X} \times \mathsf{Z} \times \mathsf{Y})$ such that for all $x \in \mathsf{X}, y \in \mathsf{Y}, z \in \mathsf{Z}$ and $\theta \in \Theta$,

$$\|\nabla_\theta \log p_\theta(y|z, x)\| \le L(x, y, z) .$$

Assumption 3 is commonly used in the analysis of convergence rates of policies (Papini et al., 2018; Surendran et al., 2025). In Adaptive MCMC, the Lipschitz condition is often applied to the Markov kernel rather than the target distribution (Andrieu & Atchadé, 2007; Andrieu & Moulines, 2006). However, since here both the kernel and the target distribution evolve dynamically, this Lipschitz condition with respect to the target distribution is more appropriate.

**Assumption 4.** There exists $\theta_\infty \in \Theta$ and a positive sequence $(a_m)_{m \in \mathbb{N}}$, with $a_m \to 0$ as $m \to \infty$ such that

$$\|\theta_m - \theta_\infty\|_{\mathrm{L}_2}^2 = O(a_m) .$$

Notably, we do not require $\theta_\infty$ to be a unique minimizer; it can simply be a critical point, which is often the case when the objective function in inference is non-convex. With additional regularity assumptions on the objective, this condition can be verified (see Appendix D.3).

**Theorem 5.3.** *Let Assumptions 1 - 4 hold. Then, there exist a constant $\rho \in (0, 1)$ and a positive sequence $(b_m)_{m \in \mathbb{N}}$ such that for all $\mu \in \mathsf{M}_1(\mathsf{Z} \times \mathsf{Y})$, and $m \in \mathbb{N}$,*

$$\mathbb{E}\big[\|\mu P_{\theta_1} \cdots P_{\theta_m} - \pi_{\theta_\infty}\|_{\mathrm{TV}}\big] = \mathcal{O}\bigg(\rho^{b_m} + \sum_{j=m-b_m}^{m-1} \gamma_{j+1} + a_m\bigg) .$$

*Furthermore, if $\limsup_{m \to \infty} \big(b_m^{-1} + b_m/m + b_m \gamma_m\big) = 0$, then:*

$$\mathbb{E}\left[\|\mu P_{\theta_1} \cdots P_{\theta_m} - \pi_{\theta_\infty}\|_{\mathrm{TV}}\right] \xrightarrow[m \to \infty]{} 0 .$$

Theorem 5.3 establishes that the time-inhomogeneous Markov Chain generated by our algorithm converges to the joint target distribution $\pi_{\theta_\infty}$. The bound in Theorem 5.3 has three key components: $(i)$ the mixing error, which reflects how well the Markov Chain mixes from an arbitrary initial distribution; $(ii)$ the tracking error, which quantifies how much the stationary distribution shifts over time due to changes in the parameters; and $(iii)$ the optimization error, which measures the difference between the current parameters and their limiting value $\theta_\infty$. These terms are interdependent: choosing a larger $b_m$ accelerates the convergence of the mixing error but may slow the convergence of the parameters, while the step size sequence $\gamma_m$ affects the convergence rate of the parameters $a_m$. If $\limsup_{m \to \infty} \big(b_m^{-1} + b_m/m + b_m \gamma_m\big) = 0$, the expected total variation distance between the Markov Chain and the target distribution tends to zero as $m \to \infty$, ensuring convergence. If $\gamma_m = m^{-\gamma}$, then choosing $b_m = \lfloor -\gamma \log(m)/\log(\rho) \rfloor$ yields a convergence rate of $\mathcal{O}\left(m^{-\gamma} \log m + a_m\right)$.

## 6 EXPERIMENTS

In this section, we illustrate our method using two classic CO problems: the TSP and the CVRP. We evaluate performance using benchmark datasets from the literature (Hottung et al., 2021), consisting of 1,000 instances drawn from a training distribution—100 nodes uniformly sampled within the unit square. To evaluate generalization, we also test on two out-of-distribution datasets with larger sizes of 125 and 150 nodes. All experiments use a latent space of dimension $d_z = 100$ and are run on a single NVIDIA RTX 6000 GPU. Details of the training and inference setup are provided in Appendix F. In our experiments, we perform SA steps at selected intervals, rather than at every iteration, to reduce the computational cost and enable more extensive exploration of the latent space (see Appendix F.2).

**Baselines.** We compare our model to a range of state-of-the-art learning-based NCO methods and industrial solvers. These include Concorde (Applegate et al., 2006), an exact solver specialized for the TSP, LKH3 (Helsgaun, 2017), a leading solver for CO problems, and Google OR-Tools (Perron & Furnon, 2019), a widely used suite of optimization tools. Among the NCO methods, we evaluate our approach against POMO (Kwon et al., 2020), CVAE-Opt (Hottung et al., 2021), EAS (Hottung et al., 2022), COMPASS (Chalumeau et al., 2023), ELG (Gao et al., 2024), and CNF (Zhou et al., 2024).

Table 1: Experimental results on TSP and CVRP without the augmentation trick. "Obj." denotes the average total travel distance, and "Time" indicates the total runtime for solving 1000 instances.

| | Method | Training distribution | | | Generalization | | | | | |
| | | n = 100 | | | n = 125 | | | n = 150 | | |
| | | Obj. | Gap | Time | Obj. | Gap | Time | Obj. | Gap | Time |
|---|---|---|---|---|---|---|---|---|---|---|
| **TSP** | Concorde | 7.752 | 0.00% | 8M | 8.583 | 0.00% | 12M | 9.346 | 0.00% | 17M |
| | LKH3 | 7.752 | 0.00% | 47M | 8.583 | 0.00% | 73M | 9.346 | 0.00% | 99M |
| | POMO (greedy) | 7.785 | 0.429% | <1M | 8.640 | 0.664% | <1M | 9.442 | 1.022% | <1M |
| | POMO (sampling) | 7.768 | 0.206% | 20M | 8.614 | 0.361% | 30M | 9.406 | 0.642% | 40M |
| | CVAE-Opt | 7.779 | 0.348% | 15H | 8.646 | 0.736% | 21H | 9.482 | 1.454% | 30H |
| | EAS | 7.767 | 0.197% | 20M | 8.607 | 0.280% | 30M | 9.387 | 0.434% | 40M |
| | COMPASS | 7.753 | 0.014% | 20M | 8.586 | 0.035% | 30M | 9.358 | 0.128% | 40M |
| | ELG | 7.783 | 0.399% | 20M | 8.634 | 0.594% | 30M | 9.427 | 0.867% | 40M |
| | CNF | 7.766 | 0.181% | 20M | 8.607 | 0.279% | 30M | 9.394 | 0.514% | 40M |
| | LGS-Net (ours) | **7.752** | **0.002%** | 20M | **8.584** | **0.012%** | 30M | **9.354** | **0.081%** | 40M |
| **CVRP** | LKH3 | 15.54 | 0.00% | 17H | 17.50 | 0.00% | 19H | 19.22 | 0.00% | 20H |
| | OR Tools | 17.084 | 9.936% | 38M | 18.036 | 3.063% | 64M | 21.209 | 10.349% | 73M |
| | POMO (greedy) | 15.740 | 1.287% | <1M | 17.905 | 2.314% | <1M | 19.882 | 3.444% | <1M |
| | POMO (sampling) | 15.607 | 0.431% | 40M | 17.652 | 0.869% | 1H | 19.539 | 1.659% | 1H30 |
| | CVAE-Opt | 15.752 | 1.364% | 32H | 17.864 | 2.080% | 36H | 19.843 | 3.240% | 46H |
| | EAS | 15.563 | 0.148% | 40M | 17.541 | 0.234% | 1H | 19.319 | 0.515% | 1H30 |
| | COMPASS | 15.561 | 0.135% | 40M | 17.546 | 0.263% | 1H | 19.358 | 0.718% | 1H30 |
| | ELG | 15.736 | 1.261% | 40M | 17.729 | 1.308% | 1H | 19.516 | 1.540% | 1H30 |
| | CNF | 15.591 | 0.328% | 40M | 17.682 | 1.040% | 1H | 19.998 | 4.047% | 1H30 |
| | LGS-Net (ours) | **15.524** | **-0.102%** | 40M | **17.496** | **-0.022%** | 1H | **19.286** | **0.343%** | 1H30 |

The average performance of each method on TSP and CVRP, where the gap to the best known solution is defined in (16), is reported in Table 1, while the results with the augmentation trick of Kwon et al. (2020) are presented in Tables 4 and 5. Overall, our approach achieves state-of-the-art performance across most settings. For the TSP, our method produces near-optimal solutions and consistently reaches optimality when the augmentation trick is applied (Table 4), while also outperforming other methods on out-of-distribution instances. Latent space models trained via RL (ours and COMPASS) outperform other baselines, highlighting the effectiveness of learned latent representations for capturing solution diversity. Importantly, our method surpasses COMPASS in all TSP settings. Although EAS achieves reasonable performance, it is considerably more expensive due to gradient computations at each iteration. For the CVRP, our model again outperforms all baselines, including both COMPASS and EAS. Furthermore, our method also surpasses LKH3 on instances with $n = 100$ and $n = 125$, and remains the only learning-based model that outperforms LKH3.

Our method also consistently outperforms both generalization-boosting methods (ELG, CNF) on TSP and CVRP. While each improves upon POMO, both still fall short of our approach. Their benefits are more pronounced under the stricter inference budgets of their original works; with our slightly larger budget, their advantage diminishes. In summary, our method achieves the best results across all TSP and CVRP settings and remains the top performer. While COMPASS is the closest competitor for TSP, EAS performs more strongly than COMPASS on CVRP, but still falls short of our method.

Table 2: Comparison of different inference methods with our model on CVRP with $n = 100$

| Method | Obj. | Gap |
|---|---|---|
| Sampling | 15.652 | 0.721% |
| DE | 15.561 | 0.135% |
| CMA-ES | 15.582 | 0.271% |
| EAS | 15.685 | 0.933% |
| Adam | 15.632 | 0.592% |
| SGLD | 15.607 | 0.431% |
| Single MCMC | 15.649 | 0.701% |
| Parallel MCMC (ours) | 15.557 | 0.109% |
| Interacting MCMC (ours) | 15.535 | -0.032% |
| LGS (ours) | **15.524** | **-0.102%** |

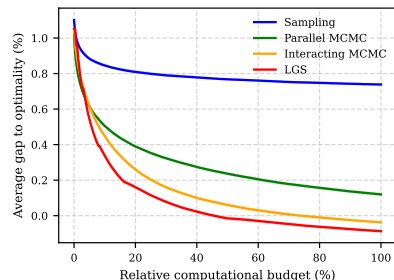

Figure 2: Performance of sampling-based methods on CVRP with $n = 100$, with bold lines indicating the mean over 10 inference runs

Table 2 and Figure 2 present the results of ablation studies, focusing on the comparison of different inference methods with our model on CVRP instances with $n = 100$ for a fixed time. Among the methods evaluated, Parallel MCMC, Interacting MCMC, and LGS consistently outperform other techniques, particularly DE and CMA-ES, which are commonly used inference methods in continuous spaces. In contrast, Single MCMC struggles due to limited exploration, leading to poor performance. Surprisingly, even gradient-based approaches such as Stochastic Gradient Langevin Dynamics (SGLD) and Adam perform poorly, as the high cost of gradient computations limits their effectiveness. EAS also underperforms, as the initial particles are insufficiently effective, and adjusting the parameters does not significantly improve the solution. This highlights the critical importance of particle propagation and parameter learning in improving solution quality. Notably, our method is the only one with negative gaps, yielding lower-cost solutions than LKH3 on CVRP.

Additionally, the "Sampling" method corresponds to direct sampling from the distribution defined in (3), without the reweighting factor $\exp(-\lambda C(y, x))$. Figure 2 highlights the advantage of incorporating this reweighting factor: the blue curve shows sampling without this factor, whereas the green curve shows sampling with this factor using parallel MCMC but with no learning or interaction between chains. Adding interaction and learning yields notable further improvements. Figure 3 further illustrates how our method explores the continuous latent space to discover high-quality solutions. While interacting MCMC benefits from faster mixing (Theorem 5.2), it is nevertheless outperformed by LGS, which achieves better cost convergence despite potentially slower mixing (Theorem 5.3).

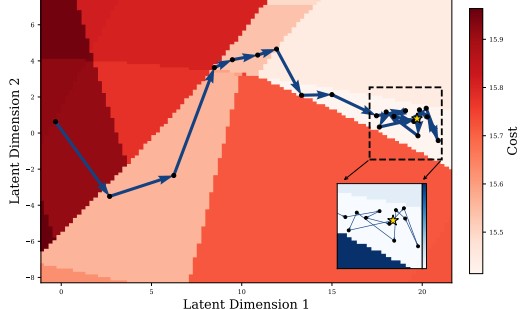

Figure 3: Visualization of the 2-dimensional latent space ($d_z = 2$) learned by our model on a problem instance. The plotted path illustrates the search trajectory leading to the best-found solution.

This contrast highlights the importance of updating $\theta$ during inference: without the SA step, interacting MCMC may converge to samples from a possibly inaccurate distribution, resulting in suboptimal solutions. Nonetheless, interacting MCMC remains competitive due to the mitigating effect of the reweighting factor.

## 7 CONCLUSION

This paper introduces LGS-Net, a novel latent space model for Neural Combinatorial Optimization that conditions directly on problem instances, thereby removing the need for labeled data and pretrained policies. We further propose a guided inference method that generates sequences of latent samples and corresponding solutions based on MCMC and SA. We establish that the iterates of our method form a time-inhomogeneous Markov Chain, with theoretical convergence guarantees. We evaluate our approach on TSP and CVRP, setting a new benchmark for learning-based NCO methods, both with and without domain-specific augmentations. A promising direction for future work is to explore how frequently the parameters of the target distribution should be updated to balance between optimal convergence rate and computational efficiency.

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

# Supplementary Material for "Latent Guided Sampling for Combinatorial Optimization"

## Table of Contents

NOTATION

Table 3: Summary of notation used throughout the paper.

| Object | Description |
|---|---|
| $x = \{x_i\}_{i=1}^n \in \mathsf{X} \subset \mathbb{R}^{n \times d_x}$ | Problem instance |
| $y = (y_1, \cdots, y_T) \in \mathsf{Y} \subset \{0, \cdots, n\}^T$ | Solution |
| $z \in \mathsf{Z} \subset \mathbb{R}^{d_z}$ | Latent variable |
| $(\mathsf{X}, \mathcal{X})$ | Problem instance space $\mathsf{X}$ with Borel $\sigma$-algebra $\mathcal{X} = \mathcal{B}(\mathsf{X})$ |
| $(\mathsf{Y}, \mathcal{Y})$ | Discrete solution space $\mathsf{Y}$ with the power set $\mathcal{Y} = \mathcal{P}(\mathsf{Y})$ |
| $(\mathsf{Z}, \mathcal{Z})$ | Latent space $\mathsf{Z}$ with Borel $\sigma$-algebra $\mathcal{Z} = \mathcal{B}(\mathsf{Z})$ |
| $\mathbb{P}_x$ | Distribution over problem instances |
| $p_\phi(z\|x)$ | Encoder distribution |
| $p_\theta(y\|x, z)$ | Decoder distribution |
| $C$ | Cost function |
| $n$ | Number of nodes in the instance |
| $t, T$ | Decoding step index and horizon |
| $m, M$ | Inference step index and total iterations |
| $k, K$ | Particle index and total number of particles |
| $B$ | Batch size used during training |

For a given batch of problem instances, we denote by $x_{(i)}$ the $i$-th input in a training batch. The corresponding solution and latent variable samples are denoted by $y_{(i)}^k$ and $z_{(i)}^k$ respectively, where $k$ indexes multiple samples drawn for the same input $x_{(i)}$. During inference, we denote by $y_m^k$ and $z_m^k$ the solution and latent variable of the $k$-th particle at the $m$-th inference iteration.

# A    PROBLEM AND MODEL DESCRIPTION

## A.1    PROBLEM SETTING

**Traveling Salesman Problem (TSP).** A TSP instance $x = \{x_i\}_{i=1}^n$ consists of a set of $n$ nodes, where the feature $x_i$ corresponds to its coordinates $c_i \in \mathbb{R}^2$. The objective is to find a permutation $y = (y_1, \ldots, y_n)$ of the nodes, where $y_t \in \{1, \ldots, n\}$ and $y_t \neq y_{t'}$ for all $t \neq t'$, that minimizes the total tour length:

$$C(y, x) = \sum_{i=1}^{n-1} \|x_{y_{i+1}} - x_{y_i}\| + \|x_{y_n} - x_{y_1}\| , \qquad (6)$$

where $\|\cdot\|$ denotes the Euclidean norm. Note that the number of decoder steps $T$ equals the number of nodes $n$, i.e., $T = n$.

**Capacitated Vehicle Routing Problem (CVRP).** CVRP generalizes TSP by introducing a depot (indexed as $0$) and multiple routes, each starting and ending at the depot. Each customer $i \in \{1, \ldots, n\}$ has a demand $d_i > 0$ and a location $c_i \in \mathbb{R}^2$, while the depot has $d_0 = 0$. A fleet of vehicles, each with a capacity $D > 0$, serves the customers. The goal is to determine the minimum number of vehicles and the corresponding routes, ensuring that each customer is visited exactly once and that the total demand in each route does not exceed $D$: for any route $j$,

$$\sum_{i \in R_j} d_i \leq D ,$$

where $R_j$ denotes the set of customers assigned to route $j$.

## A.2 MODEL ARCHITECTURE DETAILS

### A.2.1 ENCODER

Given $d_x$-dimensional input features $\mathsf{x}_i$, the encoder initially computes $d_h$-dimensional node embeddings $h_i^{(0)}$ through a learned linear projection using parameters $W_0$ and $b_0$:

$$h_i^{(0)} = W_0 \mathsf{x}_i + b_0 \ .$$

The embeddings are updated using $L$ attention layers, each consisting of two sublayers: a multi-head attention (MHA) layer followed by a node-wise fully connected feed-forward (FF) layer. Each sublayer adds a skip connection (He et al., 2016) and instance normalization (InstanceNorm) (Huang & Belongie, 2017). Denoting $h_i^{(l)}$ as the node embeddings produced by layer $l \in \{1, \dots, L\}$, the updates are defined as follows:

$$\hat{h}_i^{(l+1)} = \mathsf{InstanceNorm}\left(h_i^{(l)} + \mathsf{MHA}\left(h_1^{(l)}, \dots, h_n^{(l)}\right)\right),$$

$$h_i^{(l+1)} = \mathsf{InstanceNorm}\left(\hat{h}_i^{(l+1)} + \mathsf{FF}(\hat{h}_i^{(l+1)})\right) \ .$$

Then, it computes an aggregated embedding $\bar{h}^{(L)}$ of the input graph as the mean of the final node embeddings $h_i^{(L)}$. Finally, the encoder generates a latent space vector using a reparameterization trick:

$$z = \mu_\phi(x) + \sigma_\phi(x) \odot \varepsilon, \quad \varepsilon \sim \mathcal{N}(0, I_{d_z}) \ ,$$

where the mean $\mu_\phi(x)$ and log-variance $\log \sigma_\phi(x)^2$ of the conditional distribution $p_\phi(z|x)$ are given by:

$$\mu_\phi(x) = \mathsf{FF}\left(\bar{h}^{(L)}\right) \qquad \text{and} \qquad \log \sigma_\phi(x)^2 = \mathsf{FF}\left(\bar{h}^{(L)}\right) \ .$$

### A.2.2 TSP DECODER

The context $c_t$ for the TSP decoder at time $t$ is derived by combining the latent vector $z$ and the output up to time $t-1$. Specifically, the context is defined as:

$$c_t = \left[z, h_{\mathsf{y}_{t-1}}^{(L)}, h_{\mathsf{y}_0}^{(L)}\right] \ ,$$

where $h_{\mathsf{y}_0}^{(L)}$ and $h_{\mathsf{y}_{t-1}}^{(L)}$ represent the embeddings of the starting node and the previously selected node, respectively.

**Computation of Probabilities.** The output probabilities for the TSP decoder are computed using a single decoder layer with multi-head attention. This layer computes probabilities while incorporating a masking mechanism:

$$p_\theta(\mathsf{y}_t = i | \mathsf{y}_{0:t-1}, x, z) = \begin{cases} \mathsf{softmax}\left(\omega \tanh\left(\frac{q_{(c_t)}^T k_i}{\sqrt{d_k}}\right)\right) & \text{if } i \neq \mathsf{y}_s \quad \forall s < t \ , \\ 0 & \text{otherwise} \ , \end{cases}$$

where the query and key are given by $q_{(c_t)} = \mathsf{MHA}\left(c_t, \{h_i^{(L)}\}_{i=1}^n, \{h_i^{(L)}\}_{i=1}^n\right)$ and $k_i = W^K h_i^{(L)}$ respectively.

### A.2.3 CVRP DECODER

Similar to the TSP decoder, the context $c_t$ for the CVRP decoder at time $t$ is derived by combining the latent vector $z$ and the output up to time $t-1$. Specifically, the context is defined as:

$$c_t = \left[z, h_{\mathsf{y}_{t-1}}^{(L)}, \widehat{D}_t\right] \ ,$$

where we keep track of the remaining vehicle capacity $\widehat{D}_t$ at time $t$. At $t = 1$, this is initialized as $\widehat{D}_t = D$, after which it is updated as follows:

$$\widehat{D}_{t+1} = \begin{cases} \max(\widehat{D}_t - d_{\mathsf{y}_t, t}, 0) & \text{if } \mathsf{y}_t \neq 0 \ , \\ D & \text{if } \mathsf{y}_t = 0 \ . \end{cases}$$

**Computation of Probabilities.** The output probabilities for the CVRP decoder are computed using a single decoder layer with multi-head attention. This layer computes probabilities while incorporating a masking mechanism:

$$
p_\theta(\mathsf{y}_t = i | \mathsf{y}_{0:t-1}, x, z) = \begin{cases} \text{softmax} \left( \omega \tanh \left( \frac{q_{(c_t)}^T k_i}{\sqrt{d_k}} \right) \right) & \text{if } i \neq \mathsf{y}_s \quad \forall s < t \quad \text{and} \quad d_{i,t} \leq \widehat{D}_t \,, \\ 0 & \text{otherwise} \,, \end{cases}
$$

where the query and key are given by $q_{(c_t)} = \mathsf{MHA}\left( c_t, \{h_i^{(L)}\}_{i=1}^n, \{h_i^{(L)}\}_{i=1}^n \right)$ and $k_i = W^K h_i^{(L)}$ respectively.

## A.3 TRAINING

When setting $w^k = 1$ and $\beta = 0$ in the training objective defined in (2), the loss reduces to minimizing the expected cost over $K$ latent samples, without weighting or entropy regularization. We introduce the entropy term (controlled by $\beta$) to encourage diversity in the decoder's outputs, inspired by maximum entropy reinforcement learning (Ziebart et al., 2008; Haarnoja et al., 2018).

One could, in principle, also regularize the encoder $p_\phi$ via an entropy term. However, we found empirically that this had negligible impact on performance. Our intuition is that the decoder already induces sufficient stochasticity, and that exploration of the latent space is effectively handled during inference. Moreover, including encoder regularization would require tuning additional hyperparameters, which we deliberately avoided for the sake of simplicity.

**Justification for the weights $w^k$.**
In Bhattacharyya et al. (2018), the authors motivate the "Best-of-Many" sample objective by observing that the standard multi-sample VAE (Kingma & Welling, 2014) loss averages the reconstruction error across all latent samples, allowing even low-quality samples to influence learning. To address this, they propose training on only the best sample. However, relying solely on the best sample may overlook valuable learning signals from other informative samples.

The idea behind our introduction of weights $w^k = \exp(-C(y^k, x)/\tau)$ serves to softly prioritize lower-cost solutions among multiple latent samples drawn from $p_\phi(z|x)$. This design is inspired by importance weighting techniques used in IWAE (Burda et al., 2016), where more promising samples are assigned greater influence during training. In our setting, all samples contribute to the gradient estimator, but poor-quality samples have reduced impact, while more promising ones are emphasized.

As discussed in Section 4.2, when $w^k = 1$ ($\tau$ is large), the model treats all samples equally, encouraging exploration. In contrast, when $\tau$ is small, the weighting scheme closely resembles training on only the best sample, as in Bhattacharyya et al. (2018). In our experiments, we employ a decreasing schedule for $\tau$, which enables more stable learning in the early stages while progressively concentrating on high-quality regions of the latent space.

**Importance sampling view of the weighted objective.**
Let $q_{\theta,\phi}(z, y \mid x) = p_\phi(z \mid x)p_\theta(y \mid x, z)$ denote the joint distribution and

$$
\pi_{\theta,\phi}(z, y \mid x) \propto q_{\theta,\phi}(z, y \mid x)e^{-C(y,x)/\tau}
$$

the cost-weighted joint distribution, with normalizing constant $Z(x)$. The importance ratio $w(z, y) = \pi_{\theta,\phi}/q_{\theta,\phi} \propto e^{-C(y,x)/\tau}$ satisfies

$$
\mathbb{E}_{q_{\theta,\phi}}\big[w(z, y)C(y, x)\big] = \mathbb{E}_{\pi_{\theta,\phi}}[C(y, x)].
$$

Indeed, we have:

$$
\mathbb{E}_{q_{\theta,\phi}}\big[w(z, y)C(y, x)\big] = \int C(y, x) \frac{\pi_{\theta,\phi}(z, y \mid x)}{q_{\theta,\phi}(z, y \mid x)} q_{\theta,\phi}(z, y \mid x)dzdy
$$

$$
= \frac{1}{Z(x)} \int C(y, x)e^{-C(y,x)/\tau} q_{\theta,\phi}(z, y \mid x)dzdy
$$

$$
= \mathbb{E}_{\pi_{\theta,\phi}(\cdot|x)}\big[C(y, x)\big].
$$

Drawing $K$ i.i.d. pairs $(z^k, y^k) \sim q_{\theta,\phi}(\cdot \mid x)$, we obtain

$$\mathbb{E}_{q_{\theta,\phi}^{\otimes K}} \left[ \sum_{k=1}^{K} w^k C(y^k, x) \right] = \mathbb{E}_{\pi_{\theta,\phi}} [C(y, x)].$$

Hence, the weighted multi-sample loss can be interpreted as an importance sampling estimator of the expected cost under the cost-weighted target distribution $\pi_{\theta,\phi}$.

**Gradient computation.**

The following proposition provides the gradient of the training objective defined in (2).

**Proposition A.1.** *For all $x \in \mathsf{X}$, $\theta \in \Theta$ and $\phi \in \Phi$, we have:*

$$\nabla_\theta \mathcal{L}(\phi, \theta; x) = \sum_{k=1}^{K} \mathbb{E}_{z^k \sim p_\phi(\cdot|x)} \left[ \mathbb{E}_{y^k \sim p_\theta(\cdot|x,z^k)} \left[ w^k C(y^k, x) \nabla_\theta \log p_\theta(y^k|x, z^k) \right] \right]$$

$$- \beta \sum_{k=1}^{K} \mathbb{E}_{z^k \sim p_\phi(\cdot|x)} \left[ \mathbb{E}_{y^k \sim p_\theta(\cdot|x,z^k)} \left[ \log p_\theta(y^k|x, z^k) \nabla_\theta \log p_\theta(y^k|x, z^k) \right] \right] \,,$$

$$\nabla_\phi \mathcal{L}(\phi, \theta; x) = \sum_{k=1}^{K} \mathbb{E}_{z^k \sim p_\phi(\cdot|x)} \left[ \mathbb{E}_{y^k \sim p_\theta(\cdot|x,z^k)} \left[ w^k C(y^k, x) \right] \nabla_\phi \log p_\phi(z^k|x) \right]$$

$$+ \beta \sum_{k=1}^{K} \mathbb{E}_{z^k \sim p_\phi(\cdot|x)} \left[ \mathcal{H}(p_\theta(\cdot \mid x, z^k)) \nabla_\phi \log p_\phi(z^k|x) \right] \,.$$

*Proof.* For the gradient with respect to $\theta$, we have:

$$\nabla_\theta \mathcal{L}(\phi, \theta; x) = \sum_{k=1}^{K} \mathbb{E}_{z^k \sim p_\phi(\cdot|x)} \left[ \nabla_\theta \mathbb{E}_{y^k \sim p_\theta(\cdot|x,z^k)} \left[ w^k C(y^k, x) \right] + \beta \nabla_\theta \mathcal{H}(p_\theta(\cdot \mid x, z^k)) \right] \,.$$

For all $x \in \mathsf{X}$ and $z \in \mathsf{Z}$, applying the score-function (log-derivative trick) identity gives

$$\nabla_\theta \mathbb{E}_{y \sim p_\theta(\cdot|x,z)} [wC(y, x)] = \mathbb{E}_{y \sim p_\theta(\cdot|x,z)} [wC(y, x) \nabla_\theta \log p_\theta(y|x, z)]$$

Moreover, since $\mathcal{H}(p_\theta(\cdot \mid x, z)) = -\mathbb{E}_{y \sim p_\theta(\cdot|x,z)} [\log p_\theta(y|x, z)]$, its gradient is

$$\nabla_\theta \mathcal{H}(p_\theta(\cdot \mid x, z)) = -\mathbb{E}_{y \sim p_\theta(\cdot|x,z)} [\log p_\theta(y|x, z) \nabla_\theta \log p_\theta(y|x, z)] \,.$$

Combining these two identities yields the desired expression for the gradient with respect to $\theta$. For the gradient with respect to $\phi$, we apply the score-function identity to the function $z \mapsto \mathbb{E}_{y \sim p_\theta(\cdot|x,z)} [wC(y, x)] + \beta \mathcal{H}(p_\theta(\cdot \mid x, z))$ which does not depend on $\phi$. $\square$

The estimator of the gradient of the objective defined in (2) is computed using the Monte Carlo method:

$$\widehat{\nabla}_\theta \mathcal{L} = \frac{1}{B} \sum_{i=1}^{B} \sum_{k=1}^{K} \left( w_{(i)}^k C(y_{(i)}^k, x_{(i)}) - \beta \log p_\theta(y_{(i)}^k|x_{(i)}, z_{(i)}^k) - b(x_{(i)}) \right) \nabla_\theta \log p_\theta(y_{(i)}^k|x_{(i)}, z_{(i)}^k) \,,$$

$$\tag{7}$$

$$\widehat{\nabla}_\phi \mathcal{L} = \frac{1}{B} \sum_{i=1}^{B} \sum_{k=1}^{K} \left( w_{(i)}^k C(y_{(i)}^k, x_{(i)}) - \beta \log p_\theta(y_{(i)}^k|x_{(i)}, z_{(i)}^k) - b(x_{(i)}) \right) \nabla_\phi \log p_\phi(z_{(i)}^k|x_{(i)}) \,,$$

$$\tag{8}$$

where $b(x)$ denotes the baseline function, which is given by

$$b(x) = \frac{1}{K} \sum_{k=1}^{K} \left( w^k C(y^k, x) - \beta \log p_\theta(y^k \mid x, z^k) \right) \,.$$

This update rule has an intuitive interpretation: it adjusts the parameters $\theta$ and $\phi$ in directions that favor solutions yielding the highest reward, while simultaneously constraining the latent space to remain bounded and encouraging diversity in the sampled trajectories. The training procedure is outlined in Algorithm 2. The update step ADAM for the parameters $(\phi, \theta)$ corresponds to a single Adam update step (Kingma & Ba, 2015).

---

**Algorithm 2 REINFORCE training**

---

**Input:** Distribution over problem instances $\mathbb{P}_x$, number of training steps $N$, batch size $B$, and number of latent samples $K$.

1: Initialize the model parameters $\theta$ and $\phi$.
2: **for** epoch $= 1$ to $N$ **do**
3:     Sample problem instances $x_{(i)} \sim \mathbb{P}_x$ for $i \in \{1, \ldots, B\}$.
4:     Generate latent samples $z_{(i)}^1, \ldots, z_{(i)}^K \sim p_\phi^{\otimes K}(\cdot|x_{(i)})$ using the reparameterization trick.
5:     Sample solutions $y_{(i)}^k \sim p_\theta\left(\cdot \mid x_{(i)}, z_{(i)}^k\right)$ for all $k = 1, \ldots, K$.
6:     Compute the gradient estimates $\widehat{\nabla}_{\phi,\theta}\mathcal{L}(\phi, \theta)$ using (7) and (8).
7:     Update parameters: $(\phi, \theta) \leftarrow \text{ADAM}\left((\phi, \theta), \widehat{\nabla}_{\phi,\theta}\mathcal{L}(\phi, \theta)\right)$.
8: **end for**

**Output:** Optimized parameters $\phi$ and $\theta$.

---

## A.4   INFERENCE

The gradient of the inference objective is obtained using the log-derivative trick, in the same way as in Proposition A.1:

$$\nabla_\theta \mathcal{L}_{test}(\theta; x) = \mathbb{E}_{\pi_\theta(\cdot|x)}\left[C(y, x)\nabla_\theta \log \pi_\theta(y \mid x)\right] \ .$$

This leads to the estimator in (5), with the baseline $b(x)$ defined as the average cost over the $K$ latent samples for a given $x$:

$$b(x) = \frac{1}{K}\sum_{k=1}^{K} C(y^k, x) \ . \tag{9}$$

## B    PRELIMINARIES ON MARKOV CHAINS

In this section, we use the following definitions.

- A sequence of random variables $\{X_n, n \in \mathbb{N}\}$ is a Markov Chain with respect to the filtration $(\mathcal{F}_n)_{n \geq 0}$ with Markov kernel $P : \mathsf{X} \times \mathcal{X} \to \mathbb{R}_+$ if for any bounded measurable function $f : \mathsf{X} \to \mathbb{R}$,

$$\mathbb{E}\left[f\left(X_{n+1}\right) \mid \mathcal{F}_n\right] = Pf\left(X_n\right) = \int f(x)P\left(X_n, \mathrm{d}x\right) .$$

- Furthermore, the sequence $\{X_n, n \in \mathbb{N}\}$ is a state-dependent Markov Chain if for any bounded measurable function $f : \mathsf{X} \to \mathbb{R}$,

$$\mathbb{E}\left[f\left(X_{n+1}\right) \mid \mathcal{F}_n\right] = P_{\theta_n} f\left(X_n\right) = \int f(x)P_{\theta_n}\left(X_n, \mathrm{d}x\right) ,$$

where $P_{\theta_n} : \mathsf{X} \times \mathcal{X} \to \mathbb{R}_+$ is a Markov kernel with controlled parameters $\theta_n \in \mathbb{R}^d$.

**Definition B.1** (Invariant Probability Measure).
A probability measure $\pi$ on $(\mathsf{X}, \mathcal{X})$ is called invariant for the Markov kernel $P$ if it satisfies $\pi P = \pi$.

If $\{X_n : n \in \mathbb{N}\}$ is a Markov Chain with Markov kernel $P$ and $X_0$ is distributed according to an invariant probability measure $\pi$, then for all $n \geq 1$, we have $X_n \sim \pi$.

**Definition B.2** (Reversibility).
A Markov kernel $P$ on $(\mathsf{X}, \mathcal{X})$ is said to be reversible with respect to a probability measure $\pi$ if and only if

$$\pi(\mathrm{d}x)P(x, \mathrm{d}y) = \pi(\mathrm{d}y)P(y, \mathrm{d}x).$$

**Definition B.3** (Coupling of Probability Measures).
Let $(\mathsf{X}, \mathcal{X})$ be a measurable space and let $\mu, \nu$ be two probability measures, i.e., $\mu, \nu \in \mathsf{M}_1(\mathsf{X})$. We define $\mathcal{C}(\mu, \nu)$, the coupling set associated with $(\mu, \nu)$, as follows:

$$\mathcal{C}(\mu, \nu) = \left\{\zeta \in \mathsf{M}_1(\mathsf{X}^2) : \forall A \in \mathcal{X}, \ \zeta(A \times \mathsf{X}) = \mu(A), \ \zeta(\mathsf{X} \times A) = \nu(A)\right\} .$$

**Definition B.4** (Total Variation Distance).
Let $(\mathsf{X}, \mathcal{X})$ be a measurable space and let $\mu, \nu$ be two probability measures in $\mathsf{M}_1(\mathsf{X})$. The total variation norm between $\mu$ and $\nu$, denoted by $\|\mu - \nu\|_{\mathrm{TV}}$, is defined by

$$\|\mu - \nu\|_{\mathrm{TV}} = 2 \sup \left\{|\mu(f) - \nu(f)| : f \in \mathsf{F}(\mathsf{X}), 0 \leq f \leq 1\right\}$$
$$= 2 \inf \left\{\zeta(\Delta) : \zeta \in \mathcal{C}(\mu, \nu)\right\} ,$$

where $\Delta(x, x') = \mathbf{1}_{x \neq x'}$ for all $(x, x') \in \mathsf{X}^2$.

**Assumption 5.** Let $P$ be a Markov transition kernel on $(\mathsf{X}, \mathcal{X})$. Suppose there exists a function $V : \mathsf{X} \to [0, \infty)$ satisfying $\sup_{x \in \mathsf{X}} V(x) < \infty$, and the following conditions hold.

1. **Minorization Condition.** There exist $\mathcal{K} \in \mathcal{X}$, $\varepsilon > 0$ and a probability measure $\nu$ such that $\nu(\mathcal{K}) > 0$ and, for all $A \in \mathcal{X}$ and $x \in \mathcal{K}$,

$$P(x, A) \geq \varepsilon \nu(A) .$$

2. **Drift Condition (Foster-Lyapunov Condition).** There exist constants $\lambda \in [0, 1)$, $b \in (0, \infty)$ satisfying

$$PV(x) \leq \begin{cases} \lambda V(x), & x \notin \mathcal{K} , \\ b, & x \in \mathcal{K} . \end{cases}$$

**Theorem B.5.** *(Meyn & Tweedie, 1994; Baxendale, 2005) Let Assumption 5 hold for a function $V : \mathsf{X} \to [0, \infty)$, where $\sup_{x \in \mathsf{X}} V(x) < \infty$. Then, the Markov kernel $P$ admits a unique invariant probability measure $\pi$. Moreover, $\pi(V) < \infty$ and there exist constants $(\rho, \kappa) \in (0, 1) \times \mathbb{R}_+$ such that for all $\mu \in \mathsf{M}_1(\mathsf{X})$, and $m \in \mathbb{N}$,*

$$\|\mu P^m - \pi\|_{TV} \leq \kappa \rho^m \mu(V) .$$

This result was originally stated and proven in (Meyn & Tweedie, 1994, Theorem 2.3) with explicit formulas for $\rho$ and $\kappa$, and was later improved in (Baxendale, 2005, Section 2.1). Further details are available in (Meyn & Tweedie, 2012, Chapter 15) and (Douc et al., 2018, Chapter 15).

## C  CONVERGENCE ANALYSIS FOR FIXED $\theta$

In this section, we prove that the iterates of Algorithm 1 with $K = 1$ and fixed $\theta$ (the exact algorithm is given in Algorithm 3) form a reversible Markov Chain (Proposition 5.1) and exhibit geometric ergodicity toward the joint target distribution (Theorem 5.2). We then extend these results to the general case of arbitrary $K$ in Section C.3.

---

**Algorithm 3** Latent Guided Sampling ($K = 1$)

---

1: **Input:** Problem instance $x$, pretrained encoder $p_\phi$, pretrained decoder $p_\theta$, proposal distribution $q$, number of iterations $M$, and cost function $C$.
2: **Initialize:**
3: Sample initial particle: $z_0 \sim p_\phi(\cdot|x)$.
4: **for** $m = 0, 1, \ldots, M - 1$ **do**
5:     Propagate new particle: $\tilde{z}_{m+1} \sim q(\cdot|z_m)$.
6:     Generate new solution: $y_{m+1} \sim p_\theta(\cdot|\tilde{z}_{m+1}, x)$.
7:     Compute the acceptance probability:

$$\alpha_{m+1} = \min\left(1, e^{-\lambda(C(y_{m+1},x)-C(y_m,x))}\frac{p_\phi(\tilde{z}_{m+1}|x)}{p_\phi(z_m|x)}\right) .$$

8:     Accept $z_{m+1} = \tilde{z}_{m+1}$ with probability $\alpha_{m+1}$.
9: **end for**

---

### C.1  PROOF OF PROPOSITION 5.1

**Proposition C.1.** *The sequence $\{(Z_m, Y_m) : m \in \mathbb{N}\}$ generated by Algorithm 3 with $K = 1$ and fixed $\theta$ forms a Markov Chain with transition kernel $P_\theta$. Moreover, $P_\theta$ is $\pi_\theta$-reversible and for all $z \in \mathsf{Z}$, $y \in \mathsf{Y}$, and $A \in \mathcal{Z} \times \mathcal{Y}$,*

$$P_\theta\big((z,y), A\big) = \int_A q(dz'|z)p_\theta(dy'|z', x)\alpha(z, y, z', y') + \bar{\alpha}_\theta(z, y)\delta_{(z,y)}(A) , \tag{10}$$

*where*

$$\alpha(z, y, z', y') = \min\left(1, e^{-\lambda(C(y',x)-C(y,x))}\frac{p_\phi(z'|x)}{p_\phi(z|x)}\right) ,$$

$$\bar{\alpha}_\theta(z, y) = 1 - \int_{\mathsf{Z}\times\mathsf{Y}} q(dz'|z)p_\theta(dy'|z', x)\alpha(z, y, z', y') .$$

*Proof.* To compute the Markov kernel for the joint chain $\{(Z_m, Y_m) : m \in \mathbb{N}\}$, we introduce the filtration:

$$\mathcal{F}_m = \sigma(Z_0, Y_0, U_{1:m}) ,$$

where $U_{1:m} = (U_1, \cdots, U_m)$ denotes the sequence of uniform random variables. For all bounded or non-negative measurable function $h$ on $\mathsf{Z} \times \mathsf{Y}$ and all $m \in \mathbb{N}$,

$$\mathbb{E}\big[h(Z_{m+1}, Y_{m+1}) \mid \mathcal{F}_m\big]$$

$$= \mathbb{E}\left[1_{\left\{U_{m+1} < \alpha(Z_m, Y_m, Z'_{m+1}, Y'_{m+1})\right\}} h(Z'_{m+1}, Y'_{m+1}) \mid \mathcal{F}_m\right]$$

$$+ \mathbb{E}\left[1_{\left\{U_{m+1} \geq \alpha(Z_m, Y_m, Z'_{m+1}, Y'_{m+1})\right\}} h(Z_m, Y_m) \mid \mathcal{F}_m\right]$$

$$= \int_{\mathsf{Z}\times\mathsf{Y}} q(dz'|Z_m)p_\theta(dy'|z', x)\alpha(Z_m, Y_m, z', y')h(z, y) + \bar{\alpha}_\theta(Z_m, Y_m)h(Z_m, Y_m) ,$$

where

$$\bar{\alpha}_\theta(Z_m, Y_m) = 1 - \int_{\mathsf{Z}\times\mathsf{Y}} q(dz'|Z_m)p_\theta(dy'|z', x)\alpha(Z_m, Y_m, z', y') .$$

Thus, $\{(Z_m, Y_m) : m \in \mathbb{N}\}$ forms a Markov Chain with the transition kernel given, for all $z \in \mathsf{Z}$, $y \in \mathsf{Y}$, and $A \in \mathcal{Z} \times \mathcal{Y}$, by

$$P_\theta\big((z,y), A\big) = \int_A q(\mathrm{d}z'|z) p_\theta(\mathrm{d}y'|z',x) \alpha(z,y,z',y') + \bar{\alpha}_\theta(z,y)\delta_{(z,y)}(A) \ .$$

The reversibility of the Markov transition kernel $P_\theta$ follows directly from its construction as a Metropolis–Hastings algorithm (Douc et al., 2018). Nonetheless, we include a proof specific to our setting by verifying the detailed balance condition:

$$\pi_\theta(z,y)P_\theta\big((z,y),(z',y')\big) = \pi_\theta(z',y')P_\theta\big((z',y'),(z,y)\big) \ . \tag{11}$$

Define the ratio in the acceptance probability $\alpha$ as $r$:

$$r(z,y,z',y') = e^{-\lambda\big(C(y',x)-C(y,x)\big)}\frac{p_\phi(z'|x)}{p_\phi(z|x)} \ .$$

We separate the analysis in two cases depending on the value of $r(z,y,z',y')$.

**Case 1.** If $r(z,y,z',y') \leq 1$, then $\alpha(z,y,z',y') = r(z,y,z',y')$ and $\alpha(z',y',z,y) = 1$. Thus,

$$
\begin{aligned}
\pi_\theta(z,y)P_\theta\big((z,y),(z',y')\big) &= p_\phi(z|x)p_\theta(y|z,x)e^{-\lambda C(y,x)} P_\theta\big((z,y),(z',y')\big) \\
&= p_\phi(z|x)p_\theta(y|z,x)e^{-\lambda C(y,x)} \\
&\quad \times q(z'|z)p_\theta(y'|z',x)e^{-\lambda(C(y',x)-C(y,x))}\frac{p_\phi(z'|x)}{p_\phi(z|x)} \\
&= p_\phi(z'|x)p_\theta(y'|z',x)e^{-\lambda C(y',x)}q(z'|z)p_\theta(y|z,x) \\
&= \pi_\theta(z',y')P_\theta\big((z',y'),(z,y)\big) \ .
\end{aligned}
$$

**Case 2.**

If $r(z,y,z',y') > 1$, then $\alpha(z,y,z',y') = 1$ and $\alpha(z',y',z,y) = r(z',y',z,y)$. Similarly,

$$
\begin{aligned}
\pi_\theta(z',y')P_\theta\big((z',y'),(z,y)\big) &= p_\phi(z'|x)p_\theta(y'|z',x)e^{-\lambda C(y',x)} P_\theta\big((z',y'),(z,y)\big) \\
&= p_\phi(z'|x)p_\theta(y'|z',x)e^{-\lambda C(y',x)} \\
&\quad \times q(z|z')p_\theta(y|z,x)e^{-\lambda(C(y,x)-C(y',x))}\frac{p_\phi(z|x)}{p_\phi(z'|x)} \\
&= p_\phi(z|x)p_\theta(y|z,x)e^{-\lambda C(y,x)}q(z|z')p_\theta(y'|z',x) \\
&= \pi_\theta(z,y)P_\theta\big((z,y),(z',y')\big) \ .
\end{aligned}
$$

Since the detailed balance condition holds in both cases, we conclude that $P_\theta$ is $\pi_\theta$-reversible. $\qquad\square$

C.2  PROOF OF THEOREM 5.2 FOR $K = 1$

*Proof.* This follows from Theorem B.5, provided that we verify the minorization and drift conditions of Assumption 5. We consider the set

$$\mathcal{K} = \left\{ (z,y) \in \mathsf{Z} \times \mathsf{Y} \mid \|z\|^2 \leq R^2, C(y) \neq \max_{y' \in \mathsf{Y}} C(y') \right\}$$

which is compact since $\mathsf{Y}$ is finite. We denote by $B(c,R) = \{z \in \mathsf{Z} \mid \|z-c\|^2 \leq R^2\}$ the ball centered at $c$ with radius $R$.

**Minorization Condition.**

The Markov transition kernel is given by: for all $A \in \mathcal{Z} \times \mathcal{Y}$,

$$P_\theta((z,y), A)$$
$$= \int_A q(\mathrm{d}z'|z) p_\theta(\mathrm{d}y'|z', x) \min\left(1, e^{-\lambda(C(y',x)-C(y,x))} \frac{p_\phi(z'|x)}{p_\phi(z|x)}\right) + \bar{\alpha}_\theta(z, y)\delta_{(z,y)}(A)$$
$$\geq \int_{A\cap\mathcal{K}} q(\mathrm{d}z'|z) p_\theta(\mathrm{d}y'|z', x) \min\left(1, e^{-\lambda(C(y',x)-C(y,x))} \frac{p_\phi(z'|x)}{p_\phi(z|x)}\right) + \bar{\alpha}_\theta(z, y)\delta_{(z,y)}(A) \ .$$

We now establish a minorization by separately analyzing each term.

- Proposal Component: Since the proposal density $q$ is assumed to be positive on the compact set $B(0, R)$, for all $z, z' \in B(0, R)$,

$$q(z' \mid z) \geq \varepsilon_q := \inf_{z,z'\in B(0,R)} q(z' \mid z) > 0.$$

- Encoder Component: By Assumption 1, $p_\phi(\cdot|x)$ is positive, so that by applying a similar argument as above, we obtain the existence of $\varepsilon_e > 0$ such that

$$\frac{p_\phi(z'|x)}{p_\phi(z|x)} \geq \varepsilon_e \ .$$

- Decoder Component: By Assumption 1, the categorical transition probability satisfies:

$$\inf_{z'\in\mathsf{Z}, y'\in\mathsf{Y}} p_\theta(y'|z', x) \geq \varepsilon_d > 0 \ .$$

- Exponential Weighting: Since $e^{-\lambda(C(y',x)-C(y,x))}$ is always positive and $C$ is bounded (Assumption 1), there exits $\varepsilon_w > 0$ such that:

$$e^{-\lambda(C(y',x)-C(y,x))} \geq \varepsilon_w \ . \tag{12}$$

Combining these bounds, for all $A \in \mathcal{Z} \times \mathcal{Y}$, we get:

$$P_\theta((z,y), A)$$
$$= \int_A q(\mathrm{d}z'|z) p_\theta(\mathrm{d}y'|z', x) \min\left(1, e^{-\lambda(C(y',x)-C(y,x))} \frac{p_\phi(z'|x)}{p_\phi(z|x)}\right) + \bar{\alpha}_\theta(z, y)\delta_{(z,y)}(A)$$
$$\geq \mu^{\mathrm{Leb}}(B(0, R))|\mathsf{Y}|\varepsilon_q\varepsilon_d \min(1, \varepsilon_w\varepsilon_e) \int_A \nu_q(\mathrm{d}z')\nu_d(\mathrm{d}y') \ ,$$

where $\nu_C$ is the uniform probability measure over $\mathsf{Y}$:

$$\nu_d(\mathrm{d}y') = \frac{1}{|\mathsf{Y}|} \sum_{y\in\mathsf{Y}} \delta_y(\mathrm{d}y')$$

and

$$\nu_q(\mathrm{d}z') := \frac{\mu^{\mathrm{Leb}}(\mathrm{d}z')}{\mu^{\mathrm{Leb}}(B(0, R))} 1_{B(0,R)}(z'),$$

where $\mu^{\mathrm{Leb}}$ denotes the Lebesgue measure. Thus, the transition kernel satisfies the uniform minorization condition:

$$P_\theta((z,y), A) \geq \varepsilon\nu(A) \ , \tag{13}$$

where $\varepsilon = \mu^{\mathrm{Leb}}(B(0, R))|\mathsf{Y}|\varepsilon_q\varepsilon_d \min(1, \varepsilon_w\varepsilon_e)$ and $\nu(\mathrm{d}z', \mathrm{d}y') = \nu_q(\mathrm{d}z')\nu_d(\mathrm{d}y')$ is a probability measure.

**Drift condition.**

For a fixed $x \in \mathsf{X}$, we define the Lyapunov function for all $0 < s \leq \lambda$ as

$$V_x(z, y) = e^{sC(y,x)} \ .$$

For all $(z, y) \in \mathcal{K}$, applying $P_\theta$ to $V_x$, we have:

$$P_\theta V_x(z, y) = \int e^{sC(y', x)} \left( q(\mathrm{d}z'|z) p_\theta(\mathrm{d}y'|z', x) \alpha(z, y, z', y') + \bar{\alpha}_\theta(z, y) \delta_{(z,y)}(\mathrm{d}z', \mathrm{d}y') \right)$$

$$= \int e^{sC(y', x)} q(\mathrm{d}z'|z) p_\theta(\mathrm{d}y'|z', x) \alpha(z, y, z', y') + \bar{\alpha}_\theta(z, y) e^{sC(y, x)}$$

$$= \int \left( e^{sC(y', x)} \alpha(z, y, z', y') + e^{sC(y, x)} - \alpha(z, y, z', y') e^{sC(y, x)} \right) q(\mathrm{d}z'|z) p_\theta(\mathrm{d}y'|z', x) .$$

Since $C$ is bounded, we conclude that there exists a constant $b < \infty$ such that:

$$P_\theta V_x(z, y) \leq b .$$

For all $(z, y) \notin \mathcal{K}$,

$$\frac{P_\theta V_x(z, y)}{V_x(z, y)}$$

$$= \int \left( e^{s\left(C(y', x) - C(y, x)\right)} \alpha(z, y, z', y') + 1 - \alpha(z, y, z', y') \right) q(\mathrm{d}z'|z) p_\theta(\mathrm{d}y'|z', x)$$

$$\leq \int_{y' \in \{C(y', x) < C(y, x)\}} \left( e^{s\left(C(y', x) - C(y, x)\right)} \alpha(z, y, z', y') + 1 - \alpha(z, y, z', y') \right) q(\mathrm{d}z'|z) p_\theta(\mathrm{d}y'|z', x)$$

$$+ \int_{y' \in \{C(y', x) = C(y, x)\}} \left( e^{s\left(C(y', x) - C(y, x)\right)} \alpha(z, y, z', y') + 1 - \alpha(z, y, z', y') \right) q(\mathrm{d}z'|z) p_\theta(\mathrm{d}y'|z', x) .$$

There exists $\eta \in (0, 1)$ such that:

$$\int_{y' \in \{C(y', x) < C(y, x)\}} \left( e^{s\left(C(y', x) - C(y, x)\right)} \alpha(z, y, z', y') + 1 - \alpha(z, y, z', y') \right) q(\mathrm{d}z'|z) p_\theta(\mathrm{d}y'|z', x)$$

$$\leq \eta \int_{y' \in \{C(y', x) < C(y, x)\}} q(\mathrm{d}z'|z) p_\theta(\mathrm{d}y'|z', x) .$$

Thus,

$$I(z, y) \leq \eta + (1 - \eta) \int_{y' \in \{C(y', x) = C(y, x)\}} q(\mathrm{d}z'|z) p_\theta(\mathrm{d}y'|z', x) ,$$

which establishes the drift condition and completes the proof of geometric ergodicity in total variation distance using Theorem B.5.

For $L_2$-geometric ergodicity, note that the Markov Chain is reversible with respect to the probability measure $\pi_\theta$ (Proposition 5.1), so the Markov operator $P_\theta$ acts as a self-adjoint operator on $L_2$. The equivalence of geometric ergodicity and the existence of a spectral gap for $P_\theta$ acting on $L_2$ was established in (Roberts & Rosenthal, 1997, Theorem 2.1). Specifically, it is shown that $P_\theta$ is $L_2$-geometrically ergodic if and only if it is $\pi_\theta$-TV geometrically ergodic. As a result, there exist constants $(\rho_2, \kappa_2) \in (0, 1) \times \mathbb{R}_+$ such that for all $\mu \in \mathsf{M}_1(\mathsf{Z} \times \mathsf{Y})$, $\theta \in \Theta$, and $m \in \mathbb{N}$,

$$\|\mu P_\theta^m - \pi_\theta\|_{\mathrm{L}_2} \leq \kappa_2 \rho_2^m \|\mu - \pi_\theta\|_{\mathrm{L}_2} .$$

$\square$

### C.3 EXTENSION TO $K > 1$

Here, we show how Theorem 5.2 can be extended to the general case of $K > 1$. Notably, at each iteration, $K$ chains are generated simultaneously and independently, with interactions occurring only among the particles from the previous iteration.

*Proof.* The Markov kernel for general $K \geq 1$ is defined, for all $\mathsf{A} \in \mathcal{Z}^K \times \mathcal{Y}^K$, as

$$\mathsf{P}_\theta\left((z^{1:K}, y^{1:K}), \mathsf{A}\right)$$

$$= \int_\mathsf{A} \prod_{k=1}^K q(\mathrm{d}\tilde{z}^k|z) p_\theta(\mathrm{d}\tilde{y}^k|\tilde{z}^k, x) \alpha(z^k, y^k, \tilde{z}^k, \tilde{y}^k) + \bar{\alpha}_\theta(z^k, y^k) \delta_{(z^k, y^k)}(\mathrm{d}\tilde{z}^k, \mathrm{d}\tilde{y}^k) ,$$

where $\alpha$ and $\bar{\alpha}_\theta$ are defined in (10).

Using the same arguments as in the case $K = 1$ (cf. Section C.2), we obtain the following minorization condition:

$$\mathsf{P}_\theta\big((z^{1:K}, y^{1:K}), A\big) \geq \varepsilon^K \nu^{\otimes K}(A) \ ,$$

where $\varepsilon$ and $\nu$ are defined in (13). This completes the proof of the minorization condition. We now turn to the drift condition: for a fixed $x \in \mathsf{X}$, we define the Lyapunov function for all $0 < s \leq \lambda$ as

$$V_x^K(z^{1:K}, y^{1:K}) = \sum_{k=1}^K e^{sC(y^k, x)} \ .$$

As in the case $K = 1$, for all $\big(z^{1:K}, y^{1:K}\big) \in \mathcal{K}^K$, applying $\mathsf{P}_\theta$ to $V_x^K$ yields:

$$\mathsf{P}_\theta V_x^K(z^{1:K}, y^{1:K})$$

$$= \sum_{k=1}^K \int e^{sC(\tilde{y}^k, x)} \big(q(\mathrm{d}\tilde{z}^k|z)p_\theta(\mathrm{d}\tilde{y}^k|\tilde{z}^k, x)\alpha(z^k, y^k, \tilde{z}^k, \tilde{y}^k) + \bar{\alpha}_\theta(z^k, y^k)\delta_{(z^k, y^k)}(\mathrm{d}\tilde{z}^k, \mathrm{d}\tilde{y}^k)\big)$$

$$= \sum_{k=1}^K \int \left(\alpha(z^k, y^k, \tilde{z}^k, \tilde{y}^k)\left(e^{sC(\tilde{y}^k, x)} - e^{sC(y^k, x)}\right) + e^{sC(y^k, x^k)}\right) q(\mathrm{d}\tilde{z}^k|z)p_\theta(\mathrm{d}\tilde{y}^k|\tilde{z}^k, x) \ .$$

Using the same bounding argument as in the $K = 1$ case, we conclude that there exists a constant $b < \infty$ such that:

$$P_\theta V_x(z, y) \leq bK \ .$$

The case where the particles $(z^{1:K}, y^{1:K})$ lie outside a compact set can be handled similarly, following the same reasoning as in the proof for $K = 1$, thereby completing the extension to general $K$ for geometric ergodicity in total variation.

As in the case $K = 1$, the reversibility of the $k$-th chain can be verified. Since the Markov kernel decomposes as a product, this structure ensures that reversibility holds component-wise, which in turn implies $L_2$-geometric ergodicity for $K \geq 1$.

□

# D CONVERGENCE ANALYSIS FOR ADAPTIVE $\theta$

## D.1 CONVERGENCE ANALYSIS OF TIME-INHOMOGENEOUS MCMC ALGORITHM WITH STOCHASTIC APPROXIMATION UPDATE

In this section, we present the general results of the time-inhomogeneous MCMC algorithm, where the objective is to sample from $\pi_\theta$, which itself depends on the parameter $\theta$ updated via SA. This setting corresponds to Algorithm 1 with $K = 1$, without imposing any assumptions on the proposal density $q$, in particular without requiring symmetry. We then apply these results to our proposed inference method. We consider the following time-inhomogeneous MCMC algorithm with Stochastic Approximation update.

---

**Algorithm 4** Time-Inhomogeneous MCMC with Stochastic Approximation Update

---

1: **Input:** Number of iterations $M$, initial parameter $\theta_0$, step sizes $(\gamma_m)_{m\geq 1}$, initial distribution $\mu$, proposal distribution $q$, and target distribution $\pi_\theta$.
2: **Initialize:**
3: Sample initial particle: $x_0 \sim \mu$.
4: **for** $m = 0, 1, \ldots, M - 1$ **do**
5:     Propose a new sample: $\tilde{x}_{m+1} \sim q(\cdot|x_m)$
6:     Compute acceptance probability:

$$\alpha_{m+1} = \min\left(1, \frac{\pi_{\theta_m}(\tilde{x}_{m+1})q(x_m|\tilde{x}_{m+1})}{\pi_{\theta_m}(x_m)q(\tilde{x}_{m+1}|x_m)}\right) .$$

7:     Accept $x_{m+1} = \tilde{x}_{m+1}$ with probability $\alpha_{m+1}$.
8:     Compute the gradient estimate $H_{\theta_m}(x_{m+1})$ using previous samples.
9:     Update parameters: $\theta_{m+1} = \theta_m - \gamma_{m+1}H_{\theta_m}(x_{m+1})$.
10: **end for**

---

To analyze its convergence, we introduce the following assumptions.

**Assumption 6.** Let $(P_k)_{k\geq 1}$ be a sequence of Markov transition kernels on $(\mathsf{X}, \mathcal{X})$. Suppose there exists a function $V : \mathsf{X} \to [1, \infty)$ satisfying $\sup_{x\in\mathsf{X}} V(x) < \infty$, and the following conditions hold:

1. **Minorization condition.** There exist $\mathcal{K} \in \mathcal{X}$, $\varepsilon > 0$ and a probability measure $\nu$ such that $\nu(\mathcal{K}) > 0$ and, for all $A \in \mathcal{X}$ and $x \in \mathcal{K}$,

$$P_k(x, A) \geq \varepsilon\nu(A) .$$

2. **Drift condition.** There exist constants $\lambda \in [0, 1)$, $b \in (0, \infty)$ satisfying

$$P_k V(x) \leq \begin{cases} \lambda V(x) & x \notin \mathcal{K} , \\ b & x \in \mathcal{K} . \end{cases}$$

Assumption 6 corresponds to a minorization and drift condition similar to those used in time-homogeneous MCMC, but it holds uniformly for the sequence of kernels. While similar convergence guarantees can be established under weaker, non-uniform conditions (e.g., allowing the constants to depend on $k$), we focus on the uniform case as it aligns with our setting.

**Assumption 7.** There exists $L \in \mathsf{F}(\mathsf{X})$ such that for all $x \in \mathsf{X}$ and $\theta \in \Theta$,

$$\|\nabla_\theta \log \pi_\theta(x)\| \leq L(x) .$$

**Assumption 8.** There exists $\theta_\infty \in \Theta$ and a positive sequence $(a_m)_{m\in\mathbb{N}}$, with $a_m \to 0$ as $m \to \infty$ such that

$$\|\theta_m - \theta_\infty\|_{\mathrm{L}_2}^2 = O(a_m) .$$

Assumption 7 is similar to the assumptions considered in (Andrieu & Atchadé, 2007; Andrieu & Moulines, 2006). Notably, Assumption 8 does not require $\theta_\infty$ to be a unique minimizer; it can simply be a critical point.

**Theorem D.1.** *Let Assumptions 6 - 8 hold. Then, there exist a constant $\rho \in (0,1)$ and positive sequences $(b_m)_{m \in \mathbb{N}}$ such that for all $\mu \in \mathsf{M}_1(\mathsf{X})$, and $m \in \mathbb{N}$,*

$$\mathbb{E}\Big[ \|\mu P_{\theta_1} \cdots P_{\theta_m} - \pi_{\theta_\infty}\|_{\mathrm{TV}} \Big] = \mathcal{O}\Bigg( \rho^{b_m} + \sum_{j=m-b_m}^{m-1} \gamma_{j+1} + a_m \Bigg) .$$

*Furthermore, if $\limsup_{m \to \infty} \big( b_m^{-1} + b_m/m + b_m \gamma_m \big) = 0$, then*

$$\mathbb{E}\Big[ \|\mu P_{\theta_1} \cdots P_{\theta_m} - \pi_{\theta_\infty}\|_{\mathrm{TV}} \Big] \xrightarrow[m \to \infty]{} 0 .$$

Theorem D.1 establishes that the iterates of the time-inhomogeneous MCMC with Stochastic Approximation step converge to the target distribution $\pi_{\theta_\infty}$. To establish this result, we first present a decomposition of the error in total variation in D.1.1, followed by an upper bound on the mixing error in D.1.2. The proof of the theorem is then provided in D.1.3.

### D.1.1 ERROR DECOMPOSITION

**Lemma D.2.** *For all $1 \leq s \leq m$, we have:*

$$\|\mu P_{\theta_1} \cdots P_{\theta_m} - \pi_{\theta_\infty}\|_{\mathrm{TV}} \leq \big\|\mu P_{\theta_1} \cdots P_{\theta_m} - \pi_{\theta_s} P_{\theta_{s+1}} \cdots P_{\theta_m}\big\|_{\mathrm{TV}} + \sum_{j=s}^{m-1} \big\|\pi_{\theta_{j+1}} - \pi_{\theta_j}\big\|_{\mathrm{TV}}$$

$$+ \|\pi_{\theta_m} - \pi_{\theta_\infty}\|_{\mathrm{TV}} .$$

*Proof.* For all $m \in \mathbb{N}$, using the triangle inequality, we have:

$$\|\mu P_{\theta_1} \cdots P_{\theta_m} - \pi_{\theta_\infty}\|_{\mathrm{TV}} \leq \|\mu P_{\theta_1} \cdots P_{\theta_m} - \pi_{\theta_m}\|_{\mathrm{TV}} + \|\pi_{\theta_m} - \pi_{\theta_\infty}\|_{\mathrm{TV}} . \qquad (14)$$

Using the fact that $P_{\theta_m}$ admits $\pi_{\theta_m}$ as an invariant measure and applying the triangle inequality, for all $1 \leq s \leq m$, we have:

$$\|\mu P_{\theta_1} \cdots P_{\theta_m} - \pi_{\theta_m}\|_{\mathrm{TV}} \leq \big\|\mu P_{\theta_1} \cdots P_{\theta_m} - \pi_{\theta_s} P_{\theta_{s+1}} \cdots P_{\theta_m}\big\|_{\mathrm{TV}}$$

$$+ \sum_{j=s}^{m-1} \big\|\pi_{\theta_j} P_{\theta_{j+1}} P_{\theta_{j+2}} \cdots P_{\theta_m} - \pi_{\theta_{j+1}} P_{\theta_{j+1}} P_{\theta_{j+2}} \cdots P_{\theta_m}\big\|_{\mathrm{TV}}$$

$$\leq \big\|\mu P_{\theta_1} \cdots P_{\theta_m} - \pi_{\theta_s} P_{\theta_{s+1}} \cdots P_{\theta_m}\big\|_{\mathrm{TV}} + \sum_{j=s}^{m-1} \big\|\pi_{\theta_{j+1}} - \pi_{\theta_j}\big\|_{\mathrm{TV}} ,$$

where the last inequality follows from the fact that, for all $j$, the Markov kernels $P_{\theta_j}$ are contractions. Together with (14), this completes the proof. $\qquad \square$

This bound decomposes the total variation error into three components: $(i)$ the mixing error (first term), which measures how well the Markov chain mixes from an arbitrary initial distribution; $(ii)$ the tracking error (second term), which measures how much the stationary distributions shift over time due to changes in the parameters; and $(iii)$ the optimization error (last term), which measures the difference between the current parameters and their limiting value $\theta_\infty$.

### D.1.2 UPPER BOUND ON THE MIXING ERROR

**Proposition D.3.** *Let Assumption 6 hold for a function $V : \mathsf{X} \to [1, \infty)$, where $\sup_{x \in \mathsf{X}} V(x) < \infty$. Let $(P_k)_{k \geq 1}$ be a sequence of Markov transition kernels on $(\mathsf{X}, \mathcal{X})$. Then, there exists a constant $\rho \in (0, 1)$ such that for all $\xi, \xi' \in \mathsf{M}_1(\mathsf{X})$ and all $m \in \mathbb{N}$,*

$$\|\xi P_1 \cdots P_m - \xi' P_1 \cdots P_m\|_{\mathrm{TV}} \leq \begin{cases} 2\rho^m & \text{if } \xi, \xi' \text{ are supported on } \mathcal{K} , \\ \rho^m \big(\xi(V) + \xi'(V)\big) & \text{otherwise} . \end{cases}$$

Proposition D.3 corresponds to the mixing rate of a time-homogeneous Markov Chain and is analogous to (Douc et al., 2004, Theorem 2), though it differs in both statement and proof. In general, there are various approaches to establish the geometric ergodicity of Markov Chains. Here, we follow a coupling argument. Specifically, we adapt the proof of homogeneous Markov Chains from (Douc et al., 2018, Theorem 19.4.1) to the time-homogeneous setting.

We construct a bivariate Markov Chain $(X_k, X_k')_{k \geq 1}$ such that, marginally, $(X_k)_{k \geq 1}$ and $(X_k')_{k \geq 1}$ are Markov Chains starting from $X_1 = x$ and $X_1' = x'$, respectively, and each evolving according to the transition kernels $(P_k)_{k \geq 1}$.

To achieve this, for all $k \geq 1$, we define the modified kernel $Q_k$, defined for all $A \subset \mathcal{X}$ and $x_k \in \mathsf{X}$ as:

$$Q_k(x_k, A) = \frac{P_k(x_k, A) - \varepsilon \nu(A)}{1 - \varepsilon} ,$$

and introduce the coupling kernel $\bar{P}_k$ on $\mathsf{X}^2$, defined for all $A \times A' \subset \mathsf{X}^2$ and all $z_k = (x_k, x_k') \in \mathsf{X}^2$ by

$$\bar{P}_k(z_k, A \times A') = \mathbf{1}_{x_k = x_k'} P_k(x_k, A) \delta_{x_{k+1}}(A') + \mathbf{1}_{x_k \neq x_k'} \mathbf{1}_{z_k \notin \mathcal{K}^2} P_k(x_k, A) P_k(x_k', A')$$

$$+ \mathbf{1}_{x_k \neq x_k'} \mathbf{1}_{z_k \in \mathcal{K}^2} \left( \varepsilon \nu(A) \delta_{x_{k+1}}(A') + (1 - \varepsilon) Q_k(x_k, A) Q_k(x_k', A') \right) . \quad (15)$$

**Lemma D.4.** *Let $\left(\bar{P}_k\right)_{k \geq 0}$ be the Markov kernels on $\left(\mathsf{X}^2, \mathcal{X}^2\right)$ defined by (15). Then, for all $n \in \mathbb{N}$ and all $(x, x') \in \mathsf{X}^2$, we have*

$$\bar{P}_1 \cdots \bar{P}_m\left((x, x'), \cdot\right) \in \mathcal{C}\left(P_1 \cdots P_m(x, \cdot), P_1 \cdots P_m(x', \cdot)\right) ,$$

*where $\mathcal{C}$ denotes the set of couplings introduced in Definition B.3.*

*Proof.* We proceed by induction on $m$. By the definition of $\bar{P}_1$, we have by construction

$$\bar{P}_1\left((x, x'), \cdot\right) \in \mathcal{C}\left(P_1(x, \cdot), P_1(x', \cdot)\right) .$$

Suppose that for some $m \geq 1$, we have

$$\bar{P}_1 \cdots \bar{P}_m\left((x, x'), \cdot\right) \in \mathcal{C}\left(P_1 \cdots P_m(x, \cdot), P_1 \cdots P_m(x', \cdot)\right) .$$

Applying $\bar{P}_{m+1}$ to both sides and using the definition of composition of Markov kernels, we obtain, by definition of $\bar{P}_{m+1}$,

$$\bar{P}_1 \cdots \bar{P}_{m+1}\left((x, x'), A \times \mathsf{X}\right) = \int_{\mathsf{X} \times \mathsf{X}} \bar{P}_1 \cdots \bar{P}_m\left((x, x'), \mathrm{d}y\mathrm{d}y'\right) \bar{P}_{m+1}\left((y, y'), A \times \mathsf{X}\right)$$

$$= \int_{\mathsf{X} \times \mathsf{X}} \bar{P}_1 \cdots \bar{P}_m\left((x, x'), \mathrm{d}y\mathrm{d}y'\right) P_{m+1}(y, A)$$

$$= \int_{\mathsf{X} \times \mathsf{X}} P_1 \cdots P_m(x, \mathrm{d}y) P_{m+1}(y, A)$$

$$= P_1 \cdots P_{m+1}(x, A) ,$$

where the third equality follows from the inductive hypothesis. Similarly, we obtain

$$\bar{P}_1 \cdots \bar{P}_{m+1}\left((x, x'), \mathsf{X} \times A\right) = P_1 \cdots P_{m+1}(x', A) .$$

Thus, we conclude that

$$\bar{P}_1 \cdots \bar{P}_{m+1}\left((x, x'), \cdot\right) \in \mathcal{C}\left(P_1 \cdots P_{m+1}(x, \cdot), P_1 \cdots P_{m+1}(x', \cdot)\right) .$$

This completes the induction and proof. $\qquad \square$

**Lemma D.5.** *For all $(x, x') \in \mathsf{X}^2$ define $\Delta(x, x') = \mathbf{1}_{x \neq x'}$ and $\bar{V}(x, x') = (V(x) + V(x'))/2$. Then, for all $k \in \mathbb{N}$:*

- *If $(x, x') \in \mathcal{K}^2$, then*

$$\bar{P}_k \Delta(x, x') \leq (1 - \varepsilon)\Delta(x, x'), \quad \bar{P}_k \bar{V}(x, x') \leq b .$$

- If $(x, x') \notin \mathcal{K}^2$, then

$$\bar{P}_k \Delta(x, x') \le \Delta(x, x'), \quad \bar{P}_k \bar{V}(x, x') \le \lambda \bar{V}(x, x') .$$

*Proof.* The inequality for $\bar{P}_k \Delta$ in both cases follows immediately from the definition of $\bar{P}_k$. For the second inequality, we have:

$$\bar{P}_k \bar{V}(x, x') = \frac{P_k V(x) + P_k V(x')}{2} .$$

If $(x, x') \in \mathcal{K}^2$, then since $P_k V(x) \le b$ and $P_k V(x') \le b$, it follows that:

$$\frac{P_k V(x) + P_k V(x')}{2} \le b .$$

If $(x, x') \notin \mathcal{K}^2$, using $P_k V(x) \le \lambda V(x)$ and $P_k V(x') \le \lambda V(x')$, we obtain:

$$\bar{P}_k \bar{V}(x, x') = \frac{P_k V(x) + P_k V(x')}{2} \le \frac{\lambda V(x) + \lambda V(x')}{2} = \lambda \bar{V}(x, x') .$$

This concludes the proof. $\qquad \square$

*Proof of Proposition D.3.* For any $t \in (0, 1)$, we define

$$\varrho_t = \max \left( (1 - \varepsilon)^{1-t} b^t, \lambda^t \right) .$$

For chosen $t$, we introduce the function: $W(x, x') = \Delta^{1-t} 1_{(x,x') \in \mathcal{K}^2} + \Delta^{1-t} \bar{V}^t 1_{(x,x') \notin \mathcal{K}^2}$. Then, using Lemma D.4, we have:

$$\|P_1 \cdots P_m(x, \cdot) - P_1 \cdots P_m(x', \cdot)\|_{\mathrm{TV}} = 2 \inf \{\zeta(\Delta) : \zeta \in \mathcal{C}(P_1 \cdots P_m(x, \cdot), P_1 \cdots P_m(x', \cdot))\}$$
$$\le 2 \bar{P}_1 \cdots \bar{P}_m \Delta(x, x')$$
$$\le 2 \bar{P}_1 \cdots \bar{P}_m W(x, x') .$$

where we used $V \ge 1$. Finally, applying Hölder's inequality and using Lemma D.5, we obtain, for all $(x, x') \in \mathsf{X}^2$,

$$\bar{P}_k W(x, x') = \bar{P}_k \left( \Delta^{1-t} \bar{V}^t \right)(x, x') \le \left( \bar{P}_k \Delta(x, x') \right)^{1-t} \left( \bar{P}_k \bar{V}(x, x') \right)^t$$
$$\le \Delta^{1-t}(x, x') \times \begin{cases} (1 - \varepsilon)^{1-t} b^t & \text{if } (x, x') \in \mathcal{K}^2 \\ \lambda^t \bar{V}^t(x, x') & \text{if } (x, x') \notin \mathcal{K}^2 \end{cases}$$
$$\le \varrho_t W(x, x') .$$

This implies by induction that for all $m \in \mathbb{N}$ and all $(x, x') \in \mathsf{X}^2$,

$$\bar{P}_1 \cdots \bar{P}_m W(x, x') \le \varrho_t^m W(x, x') .$$

Then,

$$\|P_1 \cdots P_m(x, \cdot) - P_1 \cdots P_m(x', \cdot)\|_{\mathrm{TV}} \le 2 \varrho_t^m W(x, x')$$
$$\le \begin{cases} 2 \varrho_t^m & \text{if } x \in \mathcal{K} , \\ \varrho_t^m (V(x) + V(x')) & \text{if } x \notin \mathcal{K} . \end{cases}$$

This concludes the proof. $\qquad \square$

### D.1.3 PROOF OF THEOREM D.1

*Proof.* Using Lemma D.2, and taking $s = m - b_m$, we have:

$$\left\| \mu P_{\theta_1} \cdots P_{\theta_m} - \pi_{\theta_\infty} \right\|_{\mathrm{TV}} \le \left\| \mu P_{\theta_1} \cdots P_{\theta_m} - \pi_{\theta_{m-b_m}} P_{\theta_{m-b_m}} \cdots P_{\theta_m} \right\|_{\mathrm{TV}}$$

$$+ \sum_{j=m-b_m}^{m-1} \left\| \pi_{\theta_{j+1}} - \pi_{\theta_j} \right\|_{\mathrm{TV}} + \left\| \pi_{\theta_m} - \pi_{\theta_\infty} \right\|_{\mathrm{TV}}$$

$$\le \left\| \mu P_{\theta_1} \cdots P_{\theta_m} - \pi_{\theta_{m-b_m}} P_{\theta_{m-b_m}} \cdots P_{\theta_m} \right\|_{\mathrm{TV}}$$

$$+ L(x) \sum_{j=m-b_m}^{m-1} \left\| \theta_{j+1} - \theta_j \right\| + L(x) \left\| \theta_m - \theta_\infty \right\| \, ,$$

where we used the Lipschitz condition of $\pi_\theta$. For the first term, using Proposition D.3 with $\xi = \mu P_{\theta_1} \cdots P_{\theta_{m-b_m-1}}$ and $\xi' = \pi_{\theta_{m-b_m}}$, we have:

$$\left\| \mu P_{\theta_1} \cdots P_{\theta_m} - \pi_{\theta_{m-b_m}} P_{\theta_{m-b_m}} \cdots P_{\theta_m} \right\|_{\mathrm{TV}} \le \kappa \rho^{b_m} \, .$$

For the second term, using the Lipschitz condition (Assumption 7) and the recursion of $\theta_{j+1}$, we get:

$$\sum_{j=m-b_m}^{m-1} \mathbb{E}\left[ \left\| \theta_{j+1} - \theta_j \right\| \right] = \sum_{j=m-b_m}^{m-1} \gamma_{j+1} \mathbb{E}\left[ \left\| H_{\theta_m}\left( x_{m+1} \right) \right\| \right]$$

$$= \mathbb{E}\left[ L(x) \right] \sum_{j=m-b_m}^{m-1} \gamma_{j+1} \, .$$

For the last term, Using Jensen inequality and Assumption 8, we obtain:

$$\mathbb{E}\left[ \left\| \theta_m - \theta_\infty \right\| \right] \le \left\| \theta_m - \theta_\infty \right\|_{\mathrm{L}_2}^2 = \mathcal{O}\left( a_m \right) \, .$$

$$\square$$

### D.2 PROOF OF THEOREM 5.3

For a fixed $x \in \mathsf{X}$, we define the Lyapunov function for all $0 < s \le \lambda$ as

$$V_x(z, y) = 1 + e^{sC(y,x)} \, .$$

Following the procedure outlined in the proof of Theorem 5.2, it is straightforward to verify Assumption 6 (the minorization and drift condition) with $V \ge 1$. Additionally, using Assumptions 3 and 4, we can verify Assumptions 7 and 8. The proof is then concluded by applying Theorem D.1.

### D.3 CONVERGENCE RATE IN STOCHASTIC APPROXIMATION

Stochastic Approximation can be traced back to Robbins & Monro (1951). Since then, numerous variants have been proposed, including those using adaptive step sizes, such as Kingma & Ba (2015). The non-asymptotic convergence of biased SA methods has been studied in various settings. For instance, Karimi et al. (2019) analyzes the case without adaptive step sizes, while Surendran et al. (2024) extends the analysis to include adaptive schemes for non-convex smooth objectives. These works provide convergence guarantees in terms of the squared norm of the gradient of the objective function. Specifically, they show that the iterates converge to a critical point at a rate of $\mathcal{O}(\log m/\sqrt{m} + b)$, where $b$ corresponds to the bias and $m$ to the number of iterations. The analysis typically relies on standard assumptions, including the smoothness of the objective function, an assumption on the bias and variance of the gradient estimator (see Assumption H3 in Surendran et al. (2024)), and a decreasing step size.

In our setting, given that the cost is bounded, the smoothness of the test objective $\mathcal{L}_{test}$ hinges on the smoothness of the policy $p_\theta$, an assumption also used in Papini et al. (2018); Surendran et al. (2025). The stochastic update defined in (5) is bounded under Assumption 3, which allows us to verify the necessary conditions on the bias and variance. In particular, the bias is of order $\mathcal{O}(1/K)$. Therefore, with an additional smoothness assumption on $p_\theta$ and a suitable choice of step sizes, such as $\gamma_m = 1/\sqrt{m}$, Assumption 4 can be satisfied.

## E    EXTENSIVE RELATED WORK

**Other Sampling Methods.**

Besides MCMC methods, there exist works on Boltzmann sampling with non-differentiable reward functions (Fan et al., 2023; Uehara et al., 2024; Venkatraman et al., 2024; Bengio et al., 2023), which aligns well with our setting. In Fan et al. (2023); Uehara et al. (2024); Venkatraman et al. (2024), the authors propose methods for fine-tuning diffusion models to maximize potentially non-differentiable reward functions. These approaches remain applicable when adapting a generic pretrained policy. This idea is closely related to Active Search (Bello et al., 2017), where policy parameters are fine-tuned through Reinforcement Learning. Efficient Active Search (EAS) (Hottung et al., 2022) extends this approach by fine-tuning only a subset of parameters—typically by adding new layers—and by incorporating Imitation Learning (IL), which allows part of the generated samples to imitate the best solutions found. Another promising direction is GFlowNets (Bengio et al., 2023), which have also been applied to Vehicle Routing Problems (Zhang et al., 2025). In this setting, a GFlowNet generator is trained jointly with an adversarial discriminator to tackle large-scale routing problems, handling instances with up to 1,000 nodes.

# F ADDITIONAL EXPERIMENTS

## F.1 TRAINING DETAILS

In this section, we provide the details of our model and training procedure. The encoder uses multi-head attention with 8 heads and an embedding dimension of $d_h = 128$, and consists of 6 layers. The decoder includes a single multi-head attention layer with 8 heads and a key dimension of $d_k = 16$.

For both the TSP and CVRP, node coordinates $c_i$ are sampled uniformly within the unit square. In CVRP instances, customer demands $d_i$ are drawn from a uniform distribution $\mathcal{U}([1, 10])$. The vehicle capacity $D$ is set based on the number of nodes: $D = 50$ for $n = 100$, $D = 55$ for $n = 125$, and $D = 60$ for $n = 150$, following the setup used in the literature (Hottung et al., 2021).

Training is conducted only for instances with $n = 100$ nodes, using $K = 100$ latent samples. The latent space is defined as a compact space with diameter $R = 40$ and dimension $d_z = 100$. We use the Adam optimizer with a learning rate of $5 \times 10^{-4}$, a batch size of 128, and train for 2000 epochs. The momentum parameters are fixed at $\beta_1 = 0.9$ and $\beta_2 = 0.999$, with a weight decay of $1 \times 10^{-6}$. The entropic regularization parameter $\beta$ is set to 0.01, and the weights $\tau$ in the loss (2) are chosen according to an exponential decay schedule. The training loss and corresponding cost are shown in Figure 4.

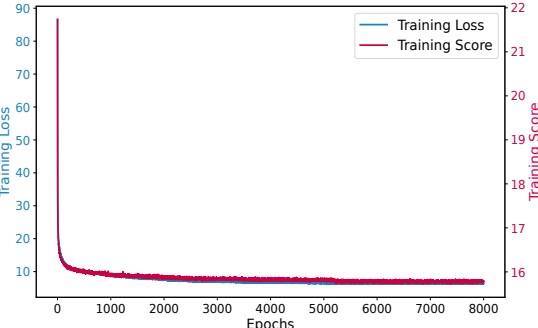

Figure 4: Training loss and score for our model trained on CVRP instances with nodes $n = 100$

## F.2 INFERENCE DETAILS AND ADDITIONAL EXPERIMENTS

### F.2.1 INFERENCE DETAILS

The inference results typically include the objective value (cost), the optimality gap, and the computation time. For example, in Tables 4 and 5, Obj. denotes the value of the cost function defined in (6) for the TSP and CVRP. The Gap indicates the percentage gap to optimality, computed as:

$$Gap(y, y^*) = \left( \frac{C(y, x)}{C(y^*, x)} - 1 \right) * 100\% , \tag{16}$$

where $y^*$ denotes the optimal solution for TSP and the near-optimal solution provided by LKH3 for CVRP.

In our experiments, we use a batch size of 200 for TSP and 100 for CVRP with $K = 600$ latent samples when the augmentation trick is not applied. When using the augmentation trick, we reduce the batch size to 100 for TSP and 50 for CVRP, and $K = 300$ latent samples.

The latent dimension is set to $d_z = 100$, and the latent space is constrained to a ball of radius $R = 40$. For MCMC, we use a Gaussian proposal distribution with density, for all $m \in \mathbb{N}$ and $1 \le k \le K$,

$$q(z_{m+1}^k \mid z_m^{1:K}) = \mathcal{N}\left( z_{m+1}^k; z_m^k + \gamma\left( z_m^{I_1} - z_m^{I_2} \right), \sigma^2 I_{d_z} \right) ,$$

where $I_1, I_2 \sim \mathcal{U}(\{1, \dots, K\})$. The variance parameter is set to $\sigma^2 = 0.01$ and the scaling factor is $\gamma = 0.319$ for TSP and $\gamma = 0.379$ for CVRP. Instead of updating the parameters at every iteration,

updates are performed at fixed intervals. The update schedule is described in the next section and illustrated in Figure 8.

### F.2.2 ADDITIONAL EXPERIMENTS

Here, we present the experimental results for TSP and CVRP with the augmentation trick. "Obj." denotes the average total cost (travel distance) over all instances, while "Time" indicates the total runtime for all 1,000 instances. "Gap" defined in (16), measures the difference from the best-known solution (Concorde for TSP and LKH3 for CVRP). Concorde is an exact solver (optimal solutions), whereas LKH3 is a heuristic (near-optimal solutions). Consequently, negative gaps indicate that the corresponding method achieves lower-cost solutions than LKH3 on CVRP. For clarity and ease of comparison, we also report the results without augmentation, as originally shown in the main paper.

Table 4: Experimental results on TSP without and with the augmentation trick

| | | Training distribution | | | Generalization | | | | | |
| | | n = 100 | | | n = 125 | | | n = 150 | | |
| | Method | Obj. | Gap | Time | Obj. | Gap | Time | Obj. | Gap | Time |
|---|---|---|---|---|---|---|---|---|---|---|
| | Concorde | 7.752 | 0.00% | 8M | 8.583 | 0.00% | 12M | 9.346 | 0.00% | 17M |
| | LKH3 | 7.752 | 0.00% | 47M | 8.583 | 0.00% | 73M | 9.346 | 0.00% | 99M |
| no aug. | POMO (greedy) | 7.785 | 0.429% | <1M | 8.640 | 0.664% | <1M | 9.442 | 1.022% | <1M |
| | POMO (sampling) | 7.768 | 0.206% | 20M | 8.614 | 0.361% | 30M | 9.406 | 0.642% | 40M |
| | CVAE-Opt | 7.779 | 0.348% | 15H | 8.646 | 0.736% | 21H | 9.482 | 1.454% | 30H |
| | EAS | 7.767 | 0.197% | 20M | 8.607 | 0.280% | 30M | 9.387 | 0.434% | 40M |
| | COMPASS | 7.753 | 0.014% | 20M | 8.586 | 0.035% | 30M | 9.358 | 0.128% | 40M |
| | ELG | 7.783 | 0.399% | 20M | 8.634 | 0.594% | 30M | 9.427 | 0.867% | 40M |
| | CNF | 7.766 | 0.181% | 20M | 8.607 | 0.279% | 30M | 9.394 | 0.514% | 40M |
| | LGS-Net (ours) | **7.752** | **0.002%** | 20M | **8.584** | **0.012%** | 30M | **9.354** | **0.081%** | 40M |
| aug. | POMO (greedy) | 7.762 | 0.132% | <1M | 8.607 | 0.280% | <1M | 9.397 | 0.541% | <1M |
| | POMO (sampling) | 7.756 | 0.051% | 40M | 8.596 | 0.156% | 60M | 9.378 | 0.342% | 90M |
| | EAS | 7.755 | 0.042% | 60M | 8.591 | 0.093% | 100M | 9.363 | 0.177% | 160M |
| | COMPASS | 7.752 | 0.002% | 40M | 8.585 | 0.024% | 60M | 9.352 | 0.059% | 90M |
| | ELG | 7.761 | 0.116% | 40M | 8.606 | 0.268% | 75M | 9.391 | 0.481% | 140M |
| | CNF | 7.756 | 0.052% | 40M | 8.595 | 0.139% | 75M | 9.377 | 0.332% | 140M |
| | LGS-Net (ours) | **7.752** | **0.000%** | 40M | **8.583** | **0.001%** | 60M | **9.349** | **0.027%** | 90M |

Table 5: Experimental results on CVRP without and with the augmentation trick

| | | Training distribution | | | Generalization | | | | | |
| | | n = 100 | | | n = 125 | | | n = 150 | | |
| | Method | Obj. | Gap | Time | Obj. | Gap | Time | Obj. | Gap | Time |
|---|---|---|---|---|---|---|---|---|---|---|
| | LKH3 | 15.54 | 0.00% | 17H | 17.50 | 0.00% | 19H | 19.22 | 0.00% | 20H |
| | OR Tools | 17.084 | 9.936% | 38M | 18.036 | 3.063% | 64M | 21.209 | 10.349% | 73M |
| no aug. | POMO (greedy) | 15.740 | 1.287% | <1M | 17.905 | 2.314% | <1M | 19.882 | 3.444% | <1M |
| | POMO (sampling) | 15.607 | 0.431% | 40M | 17.652 | 0.869% | 1H | 19.539 | 1.659% | 1H30 |
| | CVAE-Opt | 15.752 | 1.364% | 32H | 17.864 | 2.080% | 36H | 19.843 | 3.240% | 46H |
| | EAS | 15.563 | 0.148% | 40M | 17.541 | 0.234% | 1H | 19.319 | 0.515% | 1H30 |
| | COMPASS | 15.561 | 0.135% | 40M | 17.546 | 0.263% | 1H | 19.358 | 0.718% | 1H30 |
| | ELG | 15.736 | 1.261% | 40M | 17.729 | 1.308% | 1H | 19.516 | 1.540% | 1H30 |
| | CNF | 15.591 | 0.328% | 40M | 17.682 | 1.040% | 1H | 19.998 | 4.047% | 1H30 |
| | LGS-Net (ours) | **15.524** | **-0.102%** | 40M | **17.496** | **-0.022%** | 1H | **19.286** | **0.343%** | 1H30 |
| aug. | POMO (greedy) | 15.652 | 0.721% | 1M | 17.756 | 1.463% | 1M | 19.701 | 2.503% | 1M |
| | POMO (sampling) | 15.561 | 0.137% | 80M | 17.583 | 0.475% | 2H10 | 19.462 | 1.261% | 3H20 |
| | EAS | 15.508 | -0.205% | 80M | 17.466 | -0.194% | 2H10 | **19.212** | **-0.041%** | 3H20 |
| | COMPASS | 15.531 | -0.057% | 80M | 17.512 | 0.068% | 2H10 | 19.318 | 0.509% | 3H20 |
| | ELG | 15.635 | 0.611% | 90M | 17.623 | 0.703% | 2H30 | 19.421 | 1.046% | 4H15 |
| | CNF | 15.553 | 0.084% | 90M | 17.607 | 0.611% | 2H30 | 19.878 | 3.423% | 4H15 |
| | LGS-Net (ours) | **15.501** | **-0.251%** | 80M | **17.461** | **-0.223%** | 2H10 | 19.229 | 0.046% | 3H20 |

The average performance of each method is reported in Table 4 (TSP) and Table 5 (CVRP), both with and without the augmentation trick of Kwon et al. (2020). Overall, our approach achieves state-of-the-art performance across most settings. For the TSP, our method produces near-optimal solutions even without augmentation, and consistently reaches optimality when the augmentation trick is applied.

For the CVRP without augmentation, our model again outperforms all baselines, including both COMPASS and EAS. It also surpasses the performance of LKH3 on the instances with $n = 100$ and $n = 125$. When augmentation is applied, performance improves further. However, for $n = 150$, EAS outperforms our method, likely due to the reduced number of latent samples imposed by the computational budget. In this case, exploring the latent space effectively may require a higher sample count. We note that, when solving one instance at a time (thus relaxing the budget constraint), our model can achieve even stronger performance under augmentation.

**Illustration of the cost-weighted loss in training.** We compare training our model with the proposed cost-weighted loss to using the POPPY loss from Grinsztajn et al. (2023); Chalumeau et al. (2023), which focuses on the best sample. Further details on the cost-weighted loss and its mathematical justification are provided in Appendix A.3. The empirical comparison is reported in Table 6.

Table 6: Comparison of cost-weighted loss and POPPY loss when training on CVRP with $n = 100$

| Method | Obj. | Gap |
|---|---|---|
| POPPY loss | 15.527 | -0.083% |
| Cost-weighted loss | **15.524** | **-0.102%** |

Training with the cost-weighted loss (used in our main experiments) yields slightly better results than the POPPY loss, although the performance gap remains small. We attribute this to the strength of the inference procedure. In particular, we allocate a sufficiently large inference-time budget, which tends to reduce performance differences arising from the choice of training objective. Moreover, since our inference method includes parameter updates, the model can partially "forget" suboptimal training parameters and adapt at test time—an effect formally supported by our theoretical results (Theorems 5.2 and 5.3).

**Choice of the inference budget.** In our experiments, all comparisons are carried out under a fixed wall-clock budget, rather than a fixed number of rollouts or attempts, so that methods with different per-iteration costs are evaluated under comparable computational effort. This wall-clock budget is fixed for the entire test set at a given problem size. To choose this budget, we proceeded as follows. Since our architecture builds on the CVAE-Opt setting, we first ran Differential Evolution (the inference method used in Hottung et al. (2021)) on our model with the same number of iterations as in their original setup and recorded the corresponding runtime. We then used this runtime as a reference and applied the same wall-clock budget to all methods at that problem size. The resulting budgets of 20M for TSP100 and 40M for CVRP100 correspond to roughly 1 second and 2 seconds per instance, respectively, which we found to provide a reasonable trade-off between solution quality and computational cost. All methods are evaluated in the same PyTorch codebase and on the same hardware (a single NVIDIA RTX 6000 GPU).

**Comparison of our method with an instance-independent latent model.** To isolate the effect of conditioning the latent space on the instance, we additionally study a variant of our model in which the latent distribution is instance-independent. Concretely, we replace the encoder $p_\phi(z \mid x)$ with an unconditional latent distribution $p_\phi(z)$ (a standard Gaussian), while keeping the decoder architecture, training objective, and inference procedure unchanged. In this setting, the same latent space must be shared across all instances, and the search in latent space can no longer adapt its prior structure to the specific geometry of each problem instance.

We report a comparison between the instance-conditioned and instance-independent variants, as well as our inference method applied to the COMPASS model, in Table 7. First, the combined approach (COMPASS + LGS) does not improve over the original COMPASS model and remains clearly below the performance of our full method. This is reasonable, since COMPASS differs from our model not only in the lack of instance conditioning and in its loss (POPPY), but also in how the latent space is used: COMPASS draws a small number of latent samples and, for each latent sample, generates a solution with different starting points. As a result, only a limited region of the latent space is actually explored, which we believe restricts the effectiveness of our latent-space search when applied on top of a COMPASS-trained model.

The instance-independent model underperforms both the instance-conditioned variant and COMPASS. We suspect that COMPASS nevertheless works well, despite using an instance-independent latent distribution, because its latent space is learned on top of a strong pre-trained policy. Overall, these results confirm that conditioning on the instance $x$ provides a more informative latent prior for the search and yields measurable gains, in line with the intuition that an instance-dependent latent space can better capture problem-specific structure.

Table 7: Comparison of instance-conditioned and instance-independent model on CVRP with $n = 100$

| Method | Obj. | Gap |
|---|---|---|
| COMPASS + LGS | 15.579 | 0.249% |
| Instance-independent model | 15.641 | 0.648% |
| Instance-conditioned model | **15.524** | **-0.102%** |

**Comparison of deterministic and stochastic latent variables.** In the deterministic case, the encoder maps each instance to a single latent vector, $z = f_\phi(x)$, where $\phi$ denotes the encoder parameters. This collapses the latent space to a single representation per instance, so the Markov chain no longer has a meaningful space to explore. In contrast, a stochastic encoder $p_\phi(z \mid x)$ defines a distribution over the latent space that captures model uncertainty and multiple promising regions. This distribution is exactly what our method exploits: the MCMC proposals and accept/reject steps are defined with respect to a target $\pi_\theta(z \mid x)$, and exploration in latent space translates into diverse and high-quality solutions in the original solution space.

We evaluate our inference method with a deterministic latent representation (deterministic embedding from POMO). Specifically, we apply (i) a deterministic latent model with our LGS inference, and (ii) a deterministic latent model with a gradient-based method similar to EAS. The results are reported in Table 8.

Table 8: Comparison of deterministic and stochastic latent variables on CVRP with $n = 100$

| Method | Obj. | Gap |
|---|---|---|
| Deterministic latent (+LGS) | 15.572 | 0.206% |
| Deterministic latent (+SGD) | 15.564 | 0.154% |
| LGS (ours) | **15.524** | **-0.102%** |

We observe a clear degradation for both deterministic latent variants compared to the stochastic latent model, highlighting the benefit of stochasticity. In this setting, the gradient-based method performs slightly better than applying LGS directly on a deterministic latent representation, which is consistent with the fact that, without a proper target distribution in latent space, MCMC-based exploration becomes less natural. In summary, the deterministic latent model is effective for producing greedy solutions under a very small inference budget, whereas for a fixed, larger budget, a stochastic latent variable substantially improves exploration and solution quality.

**Comparison with other sampling methods.** We now compare our method to the approaches discussed in Section E: Active Search (Bello et al., 2017), which appears to be the most effective existing approach related to sampling non-differentiable reward functions (Fan et al., 2023; Uehara et al., 2024; Venkatraman et al., 2024), and the Adversarial Generative Flow Network (AGFN) (Zhang et al., 2025), which builds on GFlowNet. We evaluate these methods on CVRP instances with $n = 100$, under the same setup as in Table 2, and report the results in Table 9. The results show that Active Search outperforms POMO (both greedy and sampling) but remains inferior to our method. In contrast, AGFN performs surprisingly poorly compared to POMO-greedy, despite using the same inference budget. Notably, the original AGFN paper also reports sub-par performance on instances with $n = 200$. While the method shows improvements on extremely large-scale problems, such settings are beyond the scope of our current study but represent a valuable direction for future work.

**Complexity of MCMC + SA.** The Markov chain runs in a low-dimensional latent space, so proposals and accept/reject steps are computationally cheap. The main potential overhead comes from performing gradient updates at each iteration. To keep the inference cost moderate in our experiments, we design the implementation as follows: (i) parameter updates are applied only to

Table 9: Additional comparison of different inference methods on CVRP with $n = 100$

| Method | Obj. | Gap |
|---|---|---|
| Active Search + IL | 15.618 | 0.502% |
| AGFN (greedy) | 15.873 | 2.142% |
| LGS (ours) | **15.524** | **-0.102%** |

the last layer of the decoder (backpropagation through this layer is inexpensive), and (ii) instead of updating parameters at every iteration, updates are performed at fixed intervals. This avoids backpropagation through the full network at each step and keeps the SA component lightweight. The choice of fixed update intervals is motivated not only by computational considerations but also by the need to let the chains explore the latent space under fixed parameters, since changing the parameters at every iteration can hinder exploration. The effect of the parameter update interval is illustrated in Figure 8.

**Comparison with gradient-based methods.** Our initial motivation for using a continuous latent space was to gain the flexibility of applying continuous inference methods, particularly gradient-based techniques such as SGLD. The primary challenge in our setting is to obtain high-quality solutions within a limited computational budget. However, we found that computing gradients—especially backpropagating through the decoder in the latent space—is computationally expensive. This observation motivated us to adopt a sampling-and-learning approach instead.

In our experiments, all methods (including ours) were evaluated on batches of 1,000 problem instances. Gradient-based approaches, however, cannot process such large batches within GPU memory limits, requiring much smaller batch sizes and thereby reducing their practical efficiency. In contrast, our sampling-and-learning method updates only the decoder's final layer, and does so at fixed intervals rather than at every iteration, thereby avoiding costly end-to-end backpropagation. This design reduces computation time while preserving solution quality, underscoring the practical advantages of sampling-and-learning inference over fully gradient-based alternatives.

**Extension to larger instances ($n = 200$).** While our main experiments follow the CVAE-Opt setup with instance sizes up to $n = 150$, we additionally evaluate scalability on larger CVRP instances with $n = 200$. Using the same inference setup as for $n = 100$ (without the augmentation trick), we obtain the results reported in Table 10. LGS-Net remains competitive with strong baselines such as EAS and COMPASS on CVRP200.

Table 10: Experimental results on CVRP with $n = 200$ without the augmentation trick

| Method | Obj. | Gap | Time |
|---|---|---|---|
| LKH3 | 22.00 | 0.00% | 25H |
| POMO (greedy) | 23.296 | 5.891% | 1M |
| POMO (sampling) | 23.371 | 6.232% | 2H10 |
| EAS | 22.361 | 1.643% | 2H10 |
| COMPASS | 22.455 | 2.069% | 2H10 |
| ELG | 22.460 | 2.092% | 2H10 |
| CNF | 23.389 | 6.314% | 2H10 |
| LGS-Net (ours) | **22.284** | **1.289%** | 2H10 |

**Scalability of our method.** While our experiments are limited to $n \leq 200$, our method is in principle applicable to larger instances. The main computational cost arises from (i) propagating multiple latent particles during inference, and (ii) updating the final decoder layer via Stochastic Approximation. Both steps parallelize efficiently on GPUs, with the number of particles $K$ and the update frequency remaining constant across scales. However, the decoding horizon $T$ inevitably grows with instance size, a limitation shared by all constructive NCO methods. An interesting direction for future work is to design inference strategies that exploit problem structure by adapting the latent space at each decoding step, though this would incur significant computational cost.

### F.2.3 ILLUSTRATION OF HYPERPARAMETERS

Figure 5 illustrates solutions generated by our method, following a similar visualization style as in Perron & Furnon (2019); Kool et al. (2019). Visually, the solutions appear to be optimal.

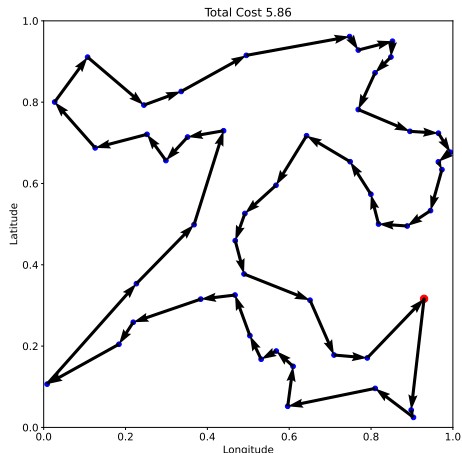 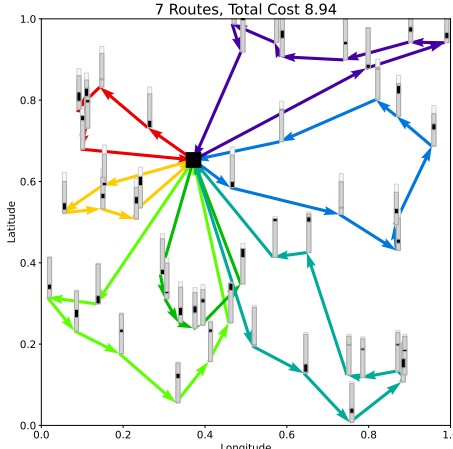

Figure 5: Solution representation produced by our model on the TSP (left) and CVRP (right) with $n = 50$. In the TSP plot, the red point denotes the starting node, and arrows indicate the visiting order. In the CVRP plot, the large square represents the depot, and each color corresponds to a distinct vehicle route. The bar illustrates vehicle capacity usage: black segments show the portion used by each customer, while white segments indicate unused capacity. The overall height of each bar reflects the total load on the corresponding route.

To highlight the impact of important hyperparameters, we evaluate our inference method while varying the number of latent samples $K$, the latent space radius $R$, the latent dimension $d_z$, and the parameter update frequency.

**Impact of the number of latent samples $K$.** We first focus on the effect of the number of latent samples $K$. Figure 6 illustrates how the average gap evolves as $K$ increases, and we observe that increasing $K$ improves the performance of our method. This can be explained by the fact that the variance of the gradient estimator in (5) is of order $\mathcal{O}(1/K)$, and also by the dependence on $K$ of the constant arising from the minorization condition in Theorem 5.3. However, beyond a certain threshold, the improvement becomes marginal. Choosing an appropriate value of $K$ is therefore crucial to balance faster mixing against computational cost.

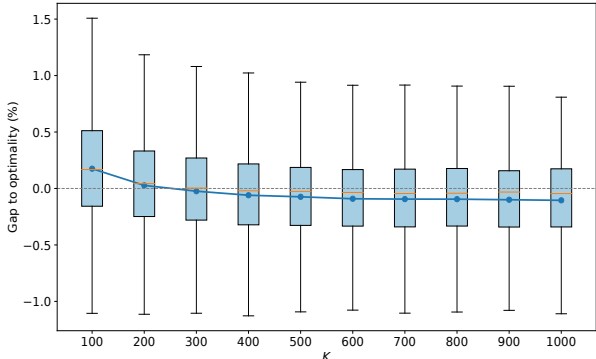

Figure 6: Average gap to optimality for different values of $K$ in the CVRP with $n = 100$

**Sensitivity to latent dimension $d_z$ and radius $R$.**  Since our latent space is, in practice, restricted to a ball of radius $R$, we study the impact of this parameter. Figure 7 illustrates the average optimality gap for different values of $R$. For small values of $R$, the model performance degrades slightly because the latent space is too constrained. For large values of $R$, the performance also deteriorates slightly, as the latent space becomes too large and harder to exploit effectively. In contrast, moderate values of $R$ strike a good balance and yield the best empirical performance. Nevertheless, across all tested values of $R$, the performance remains competitive.

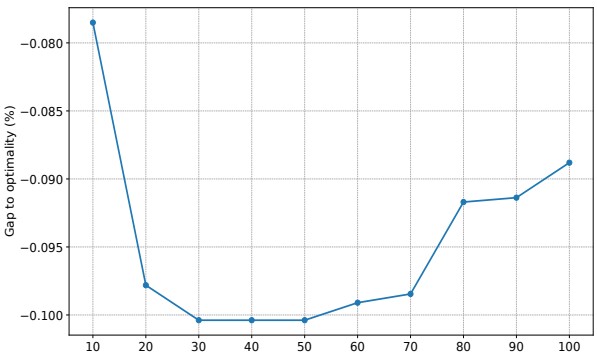

Figure 7: Average gap to optimality for different values of $R$ in the CVRP with $n = 100$

Regarding the latent dimension, we train our model with the same configuration while varying the latent dimension $d_z \in \{2, 50, 100, 150\}$, and we use the same inference procedure and budget as in our other experiments on CVRP with $n = 100$. The results are reported in Table 11. As expected, the very low-dimensional model with $d_z = 2$ performs noticeably worse, since such a small latent space cannot capture the structure of routing problems. For $d_z = 50$, the model performs competitively but slightly worse than $d_z = 100$. Increasing the dimension further to $d_z = 150$ does not bring additional gains: performance degrades marginally, likely because the higher-dimensional latent space is harder to explore efficiently under a fixed inference budget.

Table 11: Effect of latent dimension $d_z$ in our method on CVRP with $n = 100$

| Method | Obj. | Gap |
|---|---|---|
| $d_z = 2$ | 16.189 | 4.176% |
| $d_z = 50$ | 15.529 | -0.071% |
| $d_z = 100$ | **15.524** | **-0.102%** |
| $d_z = 150$ | 15.543 | 0.019% |

**Impact of the SA step frequency.**  We now discuss the SA step, focusing on how frequently the parameters should be updated. Our initial motivation for not updating the parameters at every iteration is twofold: to reduce computational cost and to better explore the latent space given the current parameter distribution. Consequently, parameter updates can be performed at regular intervals rather than every iteration.

Since the initial parameters are typically far from optimal, allowing long exploration intervals early in training is often unnecessary. Instead, we begin with short exploration intervals and gradually increase them to enable more thorough exploration as learning progresses. Selecting these intervals is non-trivial; we chose them manually without extensive hyperparameter tuning. The update schedule used in our experiments is $[1, 1, 5, 15, 25, 100, 150]$. Optimizing this schedule remains an open question and presents an interesting direction for future work.

Figure 8 illustrates the average gap to optimality for different choices of parameter update frequency, including both regular intervals and the increasing schedule. Here, $M_0$ denotes the update frequency, with $M_0 = 50$ indicating that parameters are updated every 50 iterations. We observe that while regular intervals produce similar results overall, $M_0 = 75$ yields slightly better performance. Notably, the increasing update schedule achieves a clear improvement over the fixed schedules, highlighting the potential benefits of adaptive strategies.

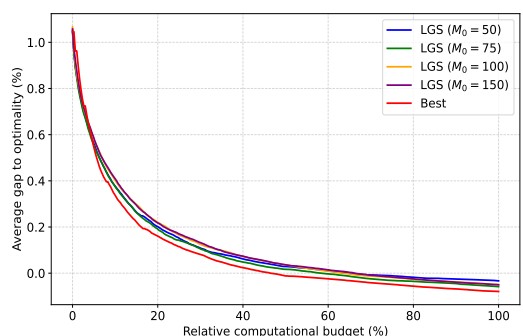

Figure 8: Performance comparison of various SA step update frequencies on CVRP with $n = 100$

### F.2.4 GENERALIZATION TO OTHER PROBLEMS

We now illustrate how our method can be extended beyond routing problems, using the classical 0/1 knapsack problem (Pisinger & Toth, 1998). Given a set of $n$ items indexed by $i \in \{1, \ldots, n\}$, each item is associated with a weight $w_i$ and a value $v_i$, and the knapsack has a capacity $D$ (we denote the capacity by $D$ for consistency with our routing notation). The 0/1 knapsack problem consists in maximizing the total value of the selected items subject to the capacity constraint:

$$\min_{a \in \{0,1\}^n} \sum_{i=1}^n a_i v_i \, , \quad \text{subject to } \sum_{i=1}^n a_i w_i \leq D \, ,$$

where $a_i = 1$ indicates that item $i$ is included in the knapsack and $a_i = 0$ otherwise.

We apply our LGS-net model with the encoder described in A.2.1, using input features $x_i = (w_i, v_i) \in \mathbb{R}^2$. The decoder follows the same architecture as the CVRP decoder in A.2.3 and keeps track of the remaining load capacity. The only difference lies in the termination condition: in the knapsack setting, an episode ends as soon as the remaining capacity reaches zero, whereas in the CVRP setting the decoder runs until all nodes have been visited.

We follow the experimental protocol of Bello et al. (2017). We generate two datasets, Knapsack50 and Knapsack100, each containing 1000 instances, with weights and values drawn independently and uniformly at random in $[0, 1]$. The knapsack capacity $D$ is set to 12.5 for Knapsack50 and 25 for Knapsack100. We compare our inference method (LGS) with a sampling-based decoding from the same trained model (*Sampling*), OR-Tools (Perron & Furnon, 2019), and the optimal dynamic-programming solutions; the results are reported in Table 12. We observe that LGS matches the optimal solutions and OR-Tools on average, whereas the sampling-based method struggles to reach this optimal performance.

Table 12: Experimental results on knapsack instances (average objective value; higher is better).

| **Method** | Sampling | LGS (ours) | OR-Tools | Optimal |
|---|---|---|---|---|
| Knapsack50 | 20.012 | **20.018** | **20.018** | **20.018** |
| Knapsack100 | 40.501 | **40.512** | **40.512** | **40.512** |

