# OpenReview forum: "Latent Guided Sampling for Combinatorial Optimization"
_ICLR.cc/2026/Conference — Submitted to ICLR 2026_

### Official Review · Reviewer_TFKR · 2025-10-30

**Soundness:** 3
**Presentation:** 4
**Contribution:** 1
**Rating:** 4
**Confidence:** 4

**Summary:**

The paper proposes LGS-Net, a latent-space model for Neural Combinatorial Optimization (NCO). The approach trains an encoder to map a problem instance into a distribution for a latent variable.  Based on a sample of this distribution, the decoder generates a distribution over the solution space which then is sampled to obtain solutions. At inference time, the authors introduce Latent Guided Sampling (LGS) — an MCMC-based method augmented with stochastic approximation updates to the decoder parameters, allowing test-time adaptation. Theoretical analysis establishes convergence guarantees for both fixed and adaptive parameter settings. Experiments on TSP and CVRP benchmarks show that LGS-Net achieves state-of-the-art or near-optimal performance, sometimes improving upon strong baselines such as COMPASS and EAS by small margins.

**Strengths:**

- The paper addresses recognized limitations of prior NCO approaches (e.g., data requirements, lack of inference-time optimization) and positions the work within a clear conceptual framework.

- Principled combination of latent modeling and RL. The unification of stochastic approximation with MCMC inference in a learned latent space is elegant and theoretically grounded.

- Theoretical soundness. The convergence proofs are rigorous and build on established MCMC theory, adapted to a time-inhomogeneous setting.

- Comprehensive experimental comparison. The authors benchmark against numerous recent NCO baselines and demonstrate consistent improvements, confirming the competitiveness of the approach.

**Weaknesses:**

- Unclear role of the concept of the latent space: During inference this method employs a local search based on MCMC updates and weight updates of the decoder. This iterative search evaluates the solution quality of intermediate samples in solution space. Hence this method can hardly be considered a latent CO method since it does not learn to solve CO problems via latent space dynamics but relies on generating samples in solution space and evaluating the cost function for them. More generally, the role of the stochastic latent variable is not clear.

- Incremental novelty. The method extends earlier latent-space NCO works such as CVAE-Opt and COMPASS. The architectural and training components are similar, with the main novelty residing in the inference mechanism.

- Limited empirical diversity. All experiments are on Euclidean routing problems (TSP/CVRP ≤ 150 nodes). The approach’s generality to other combinatorial settings (e.g., knapsack, scheduling) remains untested.

- Small empirical margins. Performance gains over prior SOTA are modest (typically < 0.2 % gap). It is unclear whether these differences are statistically significant and practically meaningful given the added complexity of MCMC + SA inference.

**Questions:**

- Why is the latent variable needed?
- Why not use only a decoder model that takes as input the problem instance x instead of the latent z(x)?
- Why is the latent variable stochastic and why couldn't a deterministic representation work as well?
- Can you show the benefit of having an encoder and the stochastic variable?

**Details Of Ethics Concerns:**

-

---

> ### Author Response · Authors · 2025-11-20
>
> We thank the reviewer for the detailed and thoughtful feedback, and for recognizing the conceptual framework and the theoretical soundness. We address each of the raised points below and have incorporated the corresponding clarifications and additions into the revised manuscript.
>
> **Role of the latent space.**
>
> ***(i) Motivation of the latent space (W1 and Q1).***
>
> Latent-space models have proven effective across diverse tasks, in particular for image and text generation. The use of a continuous latent space has also been explored for discrete routing problems (CVAE-Opt, COMPASS). The motivation for learning a latent representation in our setting is to gain more flexibility at inference time: once a continuous latent space is available, a wide range of inference methods becomes applicable, as opposed to operating directly in the discrete solution space (see (ii)). Moreover, a learned continuous latent space offers a smoother optimization landscape and exposes smooth directions for proposal moves, enabling gradient-based or covariance-adapted samplers that cannot operate directly on discrete structures.
>
> We respectfully disagree with the statement that LGS-Net “can hardly be considered a latent CO method.” In our method, the Markov chain evolves in latent space: proposals and accept/reject steps are defined on $z$, and solutions $y$ are obtained only by decoding $(x,z)$. Evaluating the cost on decoded solutions is standard in NCO and does not imply that the search itself happens in solution space. Our method explicitly follows prior latent formulations such as CVAE-Opt and COMPASS in the sense that inference-time search is carried out in latent space via a specified procedure, and only then are the resulting latent samples decoded to evaluate the cost. This is precisely the sense in which these methods, including ours, “learn to solve CO problems via latent space dynamics.”
>
> ***(ii) Why not only a decoder $p_{\theta}(x|y)$? (Q2).***
>
> A decoder-only model that maps an instance $x$ directly to a distribution over solutions corresponds to existing RL-based NCO methods (e.g., AM [1], POMO [2]), which are included among our baselines. In this case, once the model is trained, inference relies on repeated stochastic sampling to improve solution quality. Recent approaches, such as Active Search and EAS, go further by adapting model parameters during inference to enhance performance, which is costly and requires many gradient steps. All these methods are already part of our baselines in the experiments.
>
> In contrast, our method uses the latent variable $z$ as an instance-specific control variable: interacting MCMC runs in the low-dimensional latent space while the decoder parameters (restricted to the last layer to avoid computational overhead) are updated more conservatively via stochastic approximation. Empirically, our method improves over strong decoder-only baselines and also over other latent-space models under a comparable inference budget.
>
> ***(iii) Why stochastic, not deterministic latent variable? (Q3 and Q4).***
>
> A deterministic encoder $z = f_{\phi}(x)$ would collapse the latent space to a single representation per instance, so the Markov chain would no longer have a meaningful space to explore. In contrast, a stochastic encoder $q_{\phi}(z \mid x)$ defines a distribution over the latent space that captures model uncertainty and multiple promising regions. This distribution is exactly what our method exploits: the MCMC proposals and accept/reject steps are defined with respect to a target $\pi_{\theta}(z \mid x)$, and exploration in latent space translates into diverse and high-quality solutions in the original solution space.
>
> We also evaluate our inference method with a deterministic latent representation. Specifically, we apply:
> (i) a deterministic latent model with our LGS inference, and
> (ii) a deterministic latent model with a gradient-based method similar to EAS [3].
> The results (also reported in Table 8 in the revised manuscript) are:
>
> | Method                      |   Obj.   |    Gap     |
> |-----------------------------|---------:|-----------:|
> | Deterministic latent (+LGS) |  15.572  |   0.206%   |
> | Deterministic latent (+SGD) |  15.564  |   0.154%   |
> | LGS (ours)                  | **15.524** | **-0.102%** |
>
> We observe a clear degradation for both deterministic latent variants compared to the stochastic latent model, highlighting the benefit of stochasticity. In this setting, the gradient-based method performs slightly better than applying LGS directly on a deterministic latent representation, which is consistent with the fact that, without a proper target distribution in latent space, MCMC-based exploration becomes less natural. In summary, the deterministic latent model is effective for producing greedy solutions under a very small inference budget, whereas for a fixed, larger budget, a stochastic latent variable substantially improves exploration and solution quality.

---

> ### Author Response · Authors · 2025-11-20
>
> **Novelty compared to CVAE-Opt / COMPASS (W2).**
>
> The main contribution of our work lies in the design of the inference scheme for latent-space models and in providing theoretical guarantees for this scheme. Existing latent models (CVAE-Opt, COMPASS) are not directly tailored to our inference method:
>
> - CVAE-Opt requires supervised data (pre-computed solutions) and is trained with a conditional VAE objective, whereas our method is trained purely via RL without labels.
>
> - COMPASS lacks instance-dependent structure in its latent space: it is defined via a fixed prior in latent space and does not adapt to the specific problem instance, which limits its ability to define instance-specific search regions in latent space (we show how an instance-conditioned latent structure improves upon such instance-independent models in Table 7).
>
> To effectively couple training and inference, we therefore propose a model architecture that (i) does not rely on labeled data, and (ii) learns a structured latent space conditioned on the problem instance, built on top of the existing encoder–decoder architecture of AM [1]. We further introduce a cost-weighted training loss (different from the POPPY loss used in COMPASS) that aligns well with our RL formulation and admits a clean importance-sampling interpretation (see Appendix A.3). This loss is not the main focus of our contribution, so it is not emphasized in the main text, but for completeness we compare it empirically to POPPY:
>
> | Method              | **Obj.**   | **Gap**       |
> |---------------------|-----------:|--------------:|
> | POPPY loss          | 15.527     | -0.083%       |
> | Cost-weighted loss  | **15.524** | **-0.102%**   |
>
> Training with the weighted loss (used in our main experiments) yields slightly better results than POPPY, although the performance gap remains small, which we attribute to the effectiveness of our inference method and the generous inference-time budget. In particular, since our inference method involves parameter updates, the model can partially “forget” suboptimal training parameters and adapt at test time—an effect that is formally supported by our theoretical results (see Theorems 5.2 and 5.3).
>
>
> **Generality beyond routing problems (W3).**
>
> Regarding the concern about generality beyond routing problems, the experiments in the paper focus on TSP and CVRP because these are the dominant benchmarks in the Neural Combinatorial Optimization literature and allow direct comparison with prior work. Conceptually, however, the latent formulation and LGS inference are not specific to routing: the method only requires a cost function and a decoder defining a stochastic policy over feasible solutions. Moreover, the convergence analysis is fully general in the sense that it is agnostic to the policy architecture and to the choice of inference parameters.
>
> We also conduct additional experiments on the 0-1 Knapsack problem, following the setup of [4]. Due to time constraints and the importance of a careful training protocol, these experiments compare our inference method against a simple sampling baseline rather than retraining all NCO baselines. We additionally include the OR-Tools solver and the optimal solution obtained via dynamic programming. On this problem, our inference method recovers optimal solutions on the test instances, matching the optimal values, whereas the sampling method using our model also gives good results but does not reach optimality. The results are reported in the table below and are also added in Appendix F.2.4 in the revised manuscript, together with details on the problem, model, and dataset used for these experiments. This provides further evidence that the proposed latent formulation and LGS inference extend beyond routing problems.
>
> *Table: Experimental results on knapsack instances (average objective value; higher is better).*
>
> | Method        |   Sampling   |   LGS (ours)   |   OR-Tools   |   Optimal   |
> |--------------|:------------:|:--------------:|:------------:|:-----------:|
> | Knapsack50   |  20.012      |  **20.018**    |  **20.018**  |  **20.018** |
> | Knapsack100  |  40.501      |  **40.512**    |  **40.512**  |  **40.512** |

---

> ### Author Response · Authors · 2025-11-20
>
> **Magnitude and significance of empirical gains (W4).**
>
> In NCO on standard TSP/CVRP benchmarks, sub-percent improvements over strong baselines are generally considered meaningful, especially when competing methods already operate very close to high-quality heuristics such as LKH3. The strength of the empirical gains over other baselines is also noted by other reviewers. Furthermore, in some CVRP settings, our method achieves negative gaps relative to LKH3, indicating that it produces better solutions than this heuristic on average under the same time budget.
>
> Moreover, the added complexity of MCMC + SA is mitigated by the design of the implementation as follows: (i) parameter updates are applied only to the last layer of the decoder (backpropagation through this layer is inexpensive), and (ii) instead of updating parameters at every iteration, updates are performed at fixed intervals. This avoids backpropagation through the full network at each step and keeps the SA component lightweight. The choice of fixed update intervals is motivated not only by computational considerations but also by the need to let the chains explore the latent space under fixed parameters; changing the parameters at every iteration can hinder exploration. Figure 8 reports the effect of varying the update interval. Furthermore, in Table 2, we show that methods such as Stochastic Gradient Langevin Dynamics (SGLD) and Adam, which perform gradient computation in the latent space at every step, perform only moderately compared to our method, due to this additional computational overhead.
>
> Finally, in our experiments, all baselines are run under the same time budget to ensure a fair comparison. Under these matched budgets, our method consistently achieves better or comparable solution quality, suggesting that the added algorithmic complexity is manageable relative to existing approaches.
>
>
> We hope that these explanations and additions—regarding the advantages of having a stochastic latent space over a deterministic one, the new experiments on knapsack, and the clarification of our contributions—fully address your comments and further strengthen the paper. We greatly appreciate your valuable feedback, hope these changes meet your expectations, and are happy to clarify any remaining questions.
>
>
> ### References
>
> [1] Wouter Kool, Herke Van Hoof, and Max Welling. Attention, learn to solve routing problems! In International Conference on Learning Representations, 2019.
>
> [2] Yeong-Dae Kwon, Jinho Choo, Byoungjip Kim, Iljoo Yoon, Youngjune Gwon, and Seungjai Min. POMO: Policy optimization with multiple optima for reinforcement learning. In Advances in Neural Information Processing Systems, volume 33, pp. 21188–21198, 2020.
>
> [3] André Hottung, Yeong-Dae Kwon, and Kevin Tierney. Efficient active search for combinatorial optimization problems. In International Conference on Learning Representations, 2022.
>
> [4] Irwan Bello, Hieu Pham, Quoc V. Le, Mohammad Norouzi, and Samy Bengio. Neural combinatorial optimization with reinforcement learning. In International Conference on Learning Representations, Workshop Track, 2017.

---

> > ### Comment · Reviewer_TFKR · 2025-11-21
> > **W1 and Q1**
> >
> > Thanks for the detailed response. I am satisfied with most of the authors’ clarifications, but I still find the claims about the role of the latent space unclear with respect to points W1 and Q1.
> >
> > The rebuttal argues that the method “learns to solve CO problems via latent space dynamics”. However, looking at the inference procedure, the method performs a joint search that depends critically on both the latent variables and the decoded solutions. At every iteration, the latent variable is updated, but the decoded solution is also created and its cost is evaluated. The acceptance decision depends directly on these solution costs. In other words, the search relies just as much on evaluations in the solution space as on movements in the latent space.
> >
> > This also shows up in the paper’s own ablation results. The authors highlight that including the solution-cost guidance term is essential for good performance, as shown in Figure 2 and discussed in the accompanying text about the reweighting factor from solution space C(y,x) in line 458:  “Figure 2 highlights the advantage of incorporating this reweighting factor”. That suggests that the method’s search behaviour is strongly shaped by information coming from the decoded solutions, not only by properties of the latent space.
> >
> > For these reasons, I do not find it accurate to describe the method as a latent CO method. The optimisation behaviour emerges from updates and evaluations that occur in both spaces at every step, and the solution-space cost evaluations appear to play an indispensable role.

---

> > > ### Author Response · Authors · 2025-11-21
> > >
> > > We thank the reviewer for the careful follow-up and for pushing us to clarify the terminology around the role of the latent space.
> > >
> > > We fully agree that our inference scheme depends critically on both the latent variables and the decoded solutions: at every iteration we decode $y = g_\theta(x,z)$, evaluate the cost $C(y,x)$, and use this cost in the acceptance decision. This solution space feedback is indispensable, and we did not intend to suggest otherwise.
> > >
> > > Our use of the term “latent (CO) method” was meant in the same sense as in CVAE-Opt and COMPASS:
> > >
> > > (i) In CVAE-Opt, once the conditional VAE has been trained in a supervised manner, inference is performed by running Differential Evolution in the latent space: candidate latent samples are updated, then decoded to solutions, and the corresponding solution costs are used to guide the evolution.
> > >
> > > (ii) COMPASS follows the same paradigm, but uses Covariance Matrix Adaptation in the latent space: at each iteration, latent samples are drawn, decoded to solutions, and the solution costs drive the CMA-ES update.
> > >
> > > Our inference scheme follows the same overall pattern of searching in latent space: the Markov chain itself evolves in latent space (its state is $z$, not a tour), and all proposals and accept/reject steps are defined on $z$. The cost $C(y,x)$ enters through the induced target density, which in turn is used in the MCMC acceptance rule.
> > >
> > > That said, we agree that our wording “learns to solve CO problems via latent space dynamics” can be read as downplaying the role of solution space evaluations, and we will avoid this terminology. In the revised manuscript, we will therefore soften and clarify the phrasing. Concretely, we will:
> > >
> > > - avoid phrases such as “latent CO method” and “learns to solve CO problems via latent space dynamics”, and instead use formulations like “latent space guided CO method” or “MCMC in latent space guided by solution costs”; and
> > > - add a sentence in Section 4.3 explicitly stating that LGS performs MCMC updates in latent space while using decoded solution costs as guidance at every step.
> > >
> > > We hope this clarification makes our position precise: our contribution is to design and analyze an MCMC+SA procedure whose state space is latent, but whose driving signal is the solution cost. We believe this still falls squarely within the family of latent space models, in line with existing approaches such as CVAE-Opt and COMPASS. We are happy to further adjust the wording if there is still any ambiguity.

---

> ### Comment · Reviewer_TFKR · 2025-11-24
>
> Thank you for the response. The provided clarifications resolve my remaining main concern about the roles of the latent space and the solution space. I have adjusted my rating accordingly.

---

### Official Review · Reviewer_BuX2 · 2025-10-31

**Soundness:** 4
**Presentation:** 3
**Contribution:** 3
**Rating:** 6
**Confidence:** 3

**Summary:**

The paper introduces LGS-Net, a latent-space neural model for combinatorial optimization (e.g., TSP, CVRP). Similar to [Chalumeau et al., 2023], it learns instance-conditioned latent representations using reinforcement learning, yet performs inference through a new scheme based on MCMC called Latent Guided Sampling (LGS).

Contributions:
- A new latent-space NCO model (LGS-Net) that conditions on problem instances.
- A provably convergent inference method (LGS) integrating MCMC and stochastic approximation (SA).
- SOTA results on TSP and CVRP benchmarks (in distribution and slightly out of distribution), outperforming prior methods like POMO, COMPASS, and EAS, with strong generalization to unseen problem sizes.

**Strengths:**

The paper is overall well-written, gives good credit to related works, and performs decent experiments and ablations.
- Strong theoretical grounding regarding convergence.
- Standard experiments on TSP and CVRP: n in [100, 125, 150].
- SOTA results on these benchmarks, even beaten the solver for CVRP 100-125.

**Weaknesses:**

The paper and method suffer from a few weaknesses.
- Additional complexity of the inference scheme: MCMC and SA must bring a significant overhead.
- Limited experimental scope: I assume this would transfer well to other combinatorial problems like job-shop scheduling or graph problems, but the paper only tests on TSP and CVRP.

**Questions:**

1. How would you think about applying such a training and inference scheme outside of this NCO setup?

2. How realistic with respect to real problems are the assumptions made in the convergence analysis?

---

> ### Author Response · Authors · 2025-11-20
>
> We thank the reviewer for the positive and constructive feedback, and for highlighting both the theoretical grounding and the empirical performance of the method. We address each of the raised points below and have incorporated the corresponding clarifications and additions into the revised manuscript.
>
> **Complexity and overhead of MCMC + SA (W1).**
>
> The Markov chain runs in a low-dimensional latent space, so proposals and accept/reject steps are cheap. The main potential overhead comes from performing gradient steps at each iteration. To keep inference overhead moderate in our experiments, we designed the implementation as follows: (i) parameter updates are applied only to the last layer of the decoder (backpropagation through this layer is inexpensive), and (ii) instead of updating parameters at every iteration, updates are performed at fixed intervals. This avoids backpropagation through the full network at each step and keeps the SA component lightweight. The choice of fixed update intervals is motivated not only by computational considerations but also by the need to let the chains explore the latent space under fixed parameters; changing the parameters at every iteration can hinder exploration. Figure 7 reports the effect of varying the update interval. Furthermore, in Table 2, we show that methods such as Stochastic Gradient Langevin Dynamics (SGLD) and Adam, which perform gradient computation in the latent space at every step, perform only moderately compared to our method, due to this additional computational overhead.
>
> Finally, in our experiments, all baselines are run under the same time budget to ensure a fair comparison. Under these matched budgets, our method consistently achieves better or comparable solution quality, suggesting that the added algorithmic complexity is manageable relative to existing approaches.
>
> **Generality beyond routing problems (W2 and Q1).**
>
> Regarding the concern about generality beyond routing problems, the experiments in the paper focus on TSP and CVRP because these are the dominant benchmarks in the Neural Combinatorial Optimization literature and allow direct comparison with prior work. Conceptually, however, the latent formulation and LGS inference are not specific to routing: the method only requires a cost function and a decoder defining a stochastic policy over feasible solutions. Moreover, the convergence analysis is fully general in the sense that it is agnostic to the policy architecture and to the choice of inference parameters.
>
> We also conduct additional experiments on the 0-1 Knapsack problem, following the setup of [1]. Due to time constraints and the importance of a careful training protocol, these experiments compare our inference method against a simple sampling baseline rather than retraining all NCO baselines. We additionally include the OR-Tools solver and the optimal solution obtained via dynamic programming. On this problem, our inference method recovers optimal solutions on the test instances, matching the optimal values, whereas the sampling method using our model also gives good results but does not reach optimality. The results are reported in the table below and are also added in Appendix F.2.4 in the revised manuscript, together with details on the problem, model, and dataset used for these experiments. This provides further evidence that the proposed latent formulation and LGS inference extend beyond routing problems.
>
> *Table: Experimental results on knapsack instances (average objective value; higher is better).*
>
> | Method        |   Sampling   |   LGS (ours)   |   OR-Tools   |   Optimal   |
> |--------------|:------------:|:--------------:|:------------:|:-----------:|
> | Knapsack50   |  20.012      |  **20.018**    |  **20.018**  |  **20.018** |
> | Knapsack100  |  40.501      |  **40.512**    |  **40.512**  |  **40.512** |
>
> Beyond classical NCO, the same training/inference scheme can be applied whenever there is a policy to learn and only a black-box cost or reward is available. This is particularly meaningful when the solution space is discrete (a latent space is less necessary if the decision space is already continuous). This setting includes, for instance, structured prediction tasks with combinatorial outputs. In all such cases, the latent model can be trained with the same cost-weighted RL objective and optimized at test time using our latent space LGS procedure.

---

> ### Author Response · Authors · 2025-11-20
>
> **Assumptions in the convergence analysis (Q2).**
>
> The assumptions used in the convergence analysis are standard in the MCMC and Stochastic Approximation literature, and it is natural to ask how realistic they are for real-world problems.
>
> First, the conditions on the proposal distribution, encoder, and decoder in Assumptions 1 and 2 can be verified for our chosen distributions. Assumption 3 concerns the regularity of the policy: neural network decoders with bounded inputs and parameters naturally satisfy the required Lipschitz-type conditions. Assumption 4 is also independent of the underlying combinatorial problem and depends only on the choice of inference parameters (step-size schedule, number of chains $K$, etc.), so it can be enforced by design. In practice, the main sufficient conditions reduce to bounded costs and bounded inputs, which are satisfied in most real-world problems.
>
> We hope that our revisions and responses—namely, the new experiments on knapsack, the explanation of how we control the overhead of MCMC + SA, and the discussion of the theoretical assumptions—sufficiently address your concerns. We thank you again for your time and effort in reviewing our work and remain happy to clarify any remaining questions or discuss further improvements.
>
>
> ### References
>
> [1] Irwan Bello, Hieu Pham, Quoc V. Le, Mohammad Norouzi, and Samy Bengio. Neural combinatorial optimization with reinforcement learning. In International Conference on Learning Representations, Workshop Track, 2017.

---

### Official Review · Reviewer_eBRY · 2025-11-01

**Soundness:** 2
**Presentation:** 2
**Contribution:** 2
**Rating:** 4
**Confidence:** 4

**Summary:**

This paper introduces a new inference strategy for neural solvers applied to combinatorial optimization, with a primary focus on routing problems but applicability beyond them. The approach is based on latent-space models coupled with a subsequent latent-space search. It includes a training phase that anticipates the inference-time search budget and presents a coupled training–inference procedure. The method establishes close connections with prior inference strategies in the literature, namely EAS, COMPASS, and CVAE-Opt. The paper provides both mathematical guarantees and empirical results on two widely studied combinatorial optimization problems (TSP & CVRP) evaluated in- and out-of-distribution.

**Strengths:**

Significance: The use of inference-time strategies for combinatorial optimization (CO) problems is highly relevant. The proposed method achieves state-of-the-art performance with statistically significant improvements on reference benchmarks, evaluated on two well-studied problems (TSP and CVRP), both in- and out-of-distribution. I appreciate that the paper provides not only SOTA results but also mathematical justification for the proposed approach.

Originality: Most underlying concepts already exist, and the general idea of latent-space search is not new (which the authors acknowledge clearly), but the combination of latent-space modeling, MCMC sampling, and gradient-based updates is novel and supported by solid mathematical analysis. While there is no major paradigm shift, the proposed approach presents an interesting and meaningful contribution.

Clarity: The paper makes a commendable effort to explain the method, and it presents the mathematical components and proofs clearly. The main ideas are easy to follow, and the inclusion of figures, algorithms, and plots effectively supports the reader’s understanding.

Quality: The paper engages well with the existing literature, being transparent about its close connections to COMPASS, EAS, and CVAE-Opt. It presents the mathematical aspects rigorously and includes results and extended materials in the appendix. Overall, the paper demonstrates good quality, though there remains room for improvement (see weaknesses).

**Weaknesses:**

**W1. Weak motivation and unclear justification of contributions.**

The main motivation of the work remains somewhat weak. Lines 51–54 and 57–60 present the limitations of prior work (EAS and COMPASS) that LGS is supposed to address (lines 60–62). This section is crucial, as it motivates the entire paper, yet it is not fully convincing.
(i) The authors claim that EAS “fine-tunes” the policy, which poses computational challenges, but LGS also performs fine-tuning.
 (ii) The statement that “COMPASS enforces independence between the problem instance and the latent space structure” would benefit from clarification, specifically, how limiting this assumption truly is, and why relaxing it is expected to yield substantial performance gains.

Moreover, the results and ablation studies do not provide evidence that the “instance dependence” introduced in LGS is responsible for the observed performance improvements. The only ablation reported in the main paper concerns the inference-time search/sampling component. One could therefore wonder whether applying the LGS sampling and parameter-update mechanism to a COMPASS checkpoint would suffice. Overall, the paper does not convincingly establish that the proposed modifications address the stated weaknesses of prior methods. Clearer intuitions about why the proposed approach constitutes a conceptual improvement, and corresponding empirical validation through targeted ablations, would strengthen the work considerably.

Another motivation mentioned in the introduction is the removal of reliance on augmentation tricks. However, it is not clear how LGS achieves this, as other methods such as COMPASS and EAS do not fundamentally rely on augmentation either. They may use it, but they function well without it. The authors should clarify what makes LGS different in this regard; if there is no clear distinction, this point should not be used as a central motivation.

**W2. Missing ablation studies.**

Related to the previous point, some important ablations are missing. The only ablation presented concerns the sampling method, whereas one of the main motivations, conditioning on the instance, is not ablated. Ideally, this should be tested explicitly. Alternatively, the authors could revise their narrative to avoid presenting this aspect as a central feature of the method without empirical support.

**W3. Questionable timing results.**

The timing results reported in Tables 1, 4, and 5 appear inconsistent, which raises concerns about the validity of the experimental study. The times reported for POMO, EAS, and COMPASS do not align with expectations, and the reported runtime for LGS is also surprising.

First, the times for POMO and COMPASS should be roughly identical, as both share the same bottlenecks, the forward pass and environment rollouts. Their only difference is the CMA-ES update, which is negligible (see the original COMPASS paper). Therefore, one should expect similar runtimes across the benchmark, which is not the case (e.g., 10M vs. 20M on TSP100; 1h30 vs. 40M on TSP150).

Second, the reported time comparison between POMO and the other methods lacks coherence, as POMO is sometimes faster and sometimes slower, although it should consistently be equal or faster.

Third, EAS and COMPASS are reported to take the same time, which is implausible, since EAS involves backpropagation steps that should make it slower.

Finally, LGS-Net is reported to have the same runtime as COMPASS, even though it includes regular parameter updates. It should therefore be slower (perhaps only slightly if updates are infrequent), but this needs discussion if the times are indeed identical.

Overall, I am confident that there are issues with the reported timings. The authors should carefully verify and correct these results, and provide more detailed information on the time performance of their algorithm. Beyond hindering fair comparison between COMPASS and LGS-Net, these inconsistencies undermine confidence in the experimental section as a whole.

**W4. Clarity and presentation.**

The clarity of the paper could be improved. Figure 1 could include more detail to illustrate key components. It is somewhat confusing how the encoding interacts with the existing encoding that AM and POMO architectures are already using; the authors should be explicit about what differs in LGS, and how the latent space relates (or not) to the original embedding space. Additionally, certain elements, such as the “proposal distribution” (used in Algorithm 1 but not discussed in Section 4), should be better introduced.

Minor suggestions:
- “To learn solution strategies” (line 14) sounds awkward.
- It is acceptable to use “NCO” without specifying “learning-based NCO,” as the term already implies learning; this would simplify some sentences (e.g., lines 24 and 39).
- Line 52: stating that EAS is a SOTA search method is inaccurate, as it is outperformed on several benchmarks by COMPASS, PolyNet, and MEMENTO. The sentence should be softened to “among the leading methods for search-based RL.” MEMENTO [1] (NeurIPS 2025, Spotlight) could also be added to the related work.
- Line 62: “rely on the augmentation trick” is misleading, most methods are agnostic to it, even if some experiments make use of it.
- Line 113: COMPASS does not learn the latent space; it uses a fixed prior, and the policy is trained to exploit it effectively.
- Line 116: “removes the need for a pre-trained policy” should be clarified, why does this hold?
- Line 251. Small suggestion to swap theta and phi to follow the order in which encoder and decoder are mentioned.
- Line 382. I believe the usual dataset used in the NCO literature for TSP100 has 10000 instances, and TSP125 and TSP150 have 1000 instances.

[1] Memory-Enhanced Neural Solvers for Routing Problems, Neurips 2025

**Questions:**

It would be valuable to provide stronger motivation for the design choices introduced in the proposed method. For instance, why was the POPPY loss from COMPASS removed? Why was a latent-space encoder introduced? And why are all decoder parameters updated if the learned latent space is already expected to capture much of the variability? I can anticipate some of the rationale behind these decisions, but I would really appreciate hearing the authors’ explicit perspective. Clarifying these points, ideally supported by additional ablation studies, could substantially strengthen the manuscript.

In Section 4.1, it is also worth noting that the baseline AM/POMO architecture already includes an encoder. It would therefore help to clarify what the new encoder adds, and what type of information is expected to be captured in the latent space that is not already represented in the existing embedding.

It would also be helpful to report the dimensionality of the latent space used in the experiments.

Regarding the reported timings: could the authors confirm whether these values are accurate? Given the inconsistencies mentioned earlier, I am concerned about the validity of the experimental results.

Finally, on line 116, the paper states that LGS “removes the need for a pre-trained policy.” Could the authors elaborate on why this is the case?

Overall, I find the work promising and I am genuinely keen to support it. However, several concerns remain that need to be addressed to lean toward acceptance. I sincerely hope that the authors will be able to provide clarifications and additional results during the rebuttal phase to help advance this discussion.

**Details Of Ethics Concerns:**

-

---

> ### Author Response · Authors · 2025-11-20
>
> We appreciate the reviewer for taking the time to provide detailed and constructive comments, and for highlighting the significance, originality, clarity, and quality of the work, while also pointing out weaknesses that we believe have now been addressed and that helped improve the revised version. We address each of the raised points below and have incorporated the corresponding clarifications and additions into the manuscript.
>
> **Motivation and our contributions.**
>
> ***(i) Our method compared to EAS and gradient-based methods (W1).***
>
> EAS performs gradient updates on a subset of parameters (often by adding extra parameters and then fine-tuning them) during inference for each instance or small batch of instances. This leads to a substantial computational cost. Similarly, if we were to update all or even a subset of parameters at every iteration, our method would also become expensive.
>
> To avoid this overhead, the implementation is designed as follows: (i) parameter updates are applied only to the last layer of the decoder (backpropagation through this layer is inexpensive), and (ii) instead of updating parameters at every iteration, updates are performed at fixed intervals. This avoids backpropagation through the full network at each step and keeps the SA component lightweight. The choice of fixed update intervals is motivated not only by computational considerations but also by the need to let the chains explore the latent space under fixed parameters; changing the parameters at every iteration can hinder exploration. Figure 8 reports the effect of varying the update interval. Furthermore, in Table 2, we show that methods such as Stochastic Gradient Langevin Dynamics (SGLD) and Adam, which perform gradient computation in the latent space at every step, perform only moderately compared to our method, due to this additional computational overhead.
>
> In this sense, the “fine-tuning” in our method is both structurally different and much lighter than in EAS: most adaptation happens via interacting MCMC in latent space under a fixed amortized policy, and SA is used only to refine that policy in a targeted and inexpensive way. We have also added a paragraph in Appendix F.2.2 of the revised paper that explains in more detail how our design avoids the overhead of gradient computation.
>
> ***(ii) Clarifying the “independence” in COMPASS and ablation on instance dependence in LGS (W1 and W2).***
>
> COMPASS uses a fixed latent distribution that is not conditioned on the instance $x$, so it lacks instance-dependent structure in its latent space, which limits its ability to define instance-specific search regions in latent space. By contrast, our method learns an instance-conditioned distribution $q_\phi(z \mid x)$, so that the latent space can adapt to each problem instance and provide a more informative prior for the search.
>
> Intuitively, sampling $z$ from $q_\phi(z \mid x)$ gives a better starting distribution over latent space than sampling from a fixed prior, which helps guide the search from the beginning. We agree that this distinction should be supported empirically. Following the reviewer’s suggestion, we performed two additional experiments:
>
> (i) We applied our inference procedure on top of a COMPASS checkpoint and obtained a gap of 0.249\% with objective value $15.579$. In this configuration, the combined approach does not improve over the COMPASS model and remains clearly below the performance of our full method. This is reasonable, since COMPASS differs from our model not only in the lack of instance conditioning and in its loss (POPPY), but also in how the latent space is used: COMPASS draws a small number of latent samples and, for each latent sample, generates a solution with different starting points. As a result, only a limited region of the latent space is actually explored, which we believe restricts the effectiveness of our latent-space search when applied on top of a COMPASS-trained model. Our architecture is specifically designed and trained to couple the latent space with our inference scheme, so that the two components work effectively together.
>
> (ii) We trained a version of our model without conditioning on the instance, i.e., using an unconditional latent distribution $q_\phi(z)$, while keeping all other components identical. This variant performs moderately compared to the instance-conditioned model, confirming that conditioning on $x$ contributes to the performance gains. We suspect that COMPASS nevertheless works well, despite using an instance-independent latent distribution, because its latent space is learned on top of a strong pre-trained policy.
>
> These results are reported below and in Table 7 in the revised manuscript, and the narrative in the main text has been softened accordingly.

---

> ### Author Response · Authors · 2025-11-20
>
> Table: Comparison of instance-conditioned and instance-independent model on CVRP with $n = 100$
>
> | Method                    |   Obj.   |    Gap     |
> |---------------------------|---------:|-----------:|
> | COMPASS + LGS             |  15.579  |   0.249%   |
> | Instance-independent model|  15.641  |   0.648%   |
> | Instance-conditioned model| **15.524** | **-0.102%** |
>
>
> ***(iii) Augmentation as a motivation (W1).***
>
> We agree that the phrasing “rely on augmentation tricks” was too strong and potentially misleading. Our intention was to suggest that the performance of methods such as POMO and EAS fundamentally depends on the augmentation trick. In contrast, latent-variable RL approaches, including COMPASS and our method, already achieve strong performance without augmentation and improve when augmentation is applied.
>
> This point is now stated more carefully and is no longer used as a central motivation of the paper; instead, it is replaced by the motivation of an RL-based latent-space model (see Line 55-65). The main contribution of our work lies in the design of the inference scheme, and we model our architecture to exploit instance-conditioned latent spaces for this inference, together with providing theoretical guarantees for our inference method.
>
> **Timing results and consistency (W3 and Q4).**
>
> For POMO sampling, we initially used a termination condition that stopped sampling early if there was no further improvement, which explains some inconsistencies in the originally reported runtimes and also the relatively large gap compared to the numbers reported in the COMPASS paper. We later found that this was not the best option for fair comparison and have now changed the protocol so that POMO sampling is allowed the same wall-clock time as our method, leading to a more coherent and fair timing comparison.
>
> For a fixed number of iterations, we agree that EAS is slower than COMPASS and our method, due to backpropagation through the network at each step, as already noted in the TSP setting with augmentation. In that augmented setting for TSP, we allow EAS a slightly larger time budget than COMPASS and our method in order to reproduce its best reported results and highlight its strengths and weaknesses. In the main-text setting without augmentation, we match wall-clock time instead: EAS, COMPASS, and our method are all run under the same time budget, which implies that the number of iterations differs across methods.
>
> For CVRP, since all three methods (EAS, COMPASS, and ours) are competitive, we consistently use the same wall-clock time with and without augmentation, ensuring a fair comparison in terms of runtime. Note that this differs from the original COMPASS paper, where EAS was given a larger time budget; this explains why our reported gap for EAS on TSP and CVRP is larger than in their results. Our protocol enforces a common time budget across methods to enable a fair comparison.
> We have now updated Tables 1, 4, and 5 accordingly in the revised manuscript.
>
>
>
> **Clarity and presentation (W3).**
>
> We appreciate these suggestions and have updated the presentation accordingly. We added the recent work MEMENTO [2] to the related work and now list both COMPASS and MEMENTO among the SOTA baselines. We also clarify that the dataset used in our experiments is the same as in CVAE-Opt. In addition, we now explicitly report the latent dimension used in all experiments, $d_z = 100$, in the Experiments section. Further inference details, including the proposal distribution, are provided in Appendix F.2.1, and we intend to move part of this material into the main text if an extra page is granted upon acceptance.

---

> ### Author Response · Authors · 2025-11-20
>
> **Answers to specific questions.**
>
> ***“Why was the POPPY loss from COMPASS removed?” (Q1)***
>
> The POPPY loss aims to train on the best sample. However, relying solely on the best sample may overlook valuable learning signals from other informative samples. The idea behind our introduction of weights
> $w^{k} = \exp(- C(y^{k}, x) / \tau)$
> is to softly prioritize lower-cost solutions among multiple latent samples drawn from $p_{\phi}(z \mid x)$.
>
> This design is inspired by importance-weighting techniques used in IWAE [1], where more promising samples are assigned greater influence during training. In our setting, all samples contribute to the gradient estimator, but poor-quality samples have reduced impact, while high-quality ones are emphasized. Furthermore, this setup has a mathematical justification: our loss can be interpreted as an importance-sampling estimator of the expected cost under the cost-weighted target distribution, as explained in more detail in Appendix A.3.
>
> As discussed in the paper, when $w^k = 1$ (i.e., $\tau$ is large), the model treats all samples equally, encouraging exploration. In contrast, when $\tau$ is small, the weighting scheme closely resembles training on only the best sample, similarly to the POPPY loss. In our experiments, we employ a decreasing schedule for $\tau$, which enables more stable learning in the early stages while progressively concentrating on high-quality regions of the latent space.
>
> Our formulation of the weighted cost is, to the best of our knowledge, novel in the context of latent-variable models for combinatorial optimization. However, since this is not the main focus of our contribution, we chose not to highlight it prominently in the main text.
>
> We empirically compare our approach to the POPPY training objective and the results are summarized below:
>
> | Method              | **Obj.**   | **Gap**       |
> |---------------------|-----------:|--------------:|
> | POPPY loss          | 15.527     | -0.083%       |
> | Cost-weighted loss  | **15.524** | **-0.102%**   |
>
> We observe that training with the weighted loss (used in our main experiments) yields slightly better results than POPPY loss, although the performance gap remains small, which we attribute to the effectiveness of our inference method. In particular, we allocate a sufficient inference-time budget, which helps reduce performance differences arising from the choice of training objective. Moreover, since our inference method involves parameter updates, the model can partially “forget” suboptimal training parameters and adapt at test time—an effect that is formally supported by our theoretical results (see Theorems 5.2 and 5.3).
>
> ***“Why was a latent-space encoder introduced?” / “The baseline AM/POMO architecture already includes an encoder… what does the new encoder add?” (Q1 and Q2)***
>
> Our latent space encoder is described in detail in Section A.2.1. Concretely, it is built on top of the AM/POMO encoder: we reuse the same instance embedding architecture and add a few additional layers to compute the mean and variance of the latent distribution. In the main text, we state that “our encoder architecture follows the general structure of Kool et al. (2019) but includes additional layers to compute the parameters of the encoder distribution,” and we keep the full architectural details in the appendix (with the intention to move part of them to the main text if an extra page is granted upon acceptance).
>
> The purpose of these additional encoder layers is to obtain a latent space conditioned on the instance, so that we can sample $z \sim q_{\phi}(z \mid x)$ and run our latent space search in an instance-specific region (see our answer to (W1) above). By contrast, using a deterministic embedding such as $z = f_{\phi}(x)$ would collapse the latent space to a single representation per instance, so the Markov chain would no longer have a meaningful space to explore. A stochastic encoder $q_{\phi}(z \mid x)$ instead defines a distribution over the latent space that captures model uncertainty and multiple promising regions. Additional empirical comparisons between deterministic and stochastic encoders are reported in Table 8, together with a short discussion.

---

> ### Author Response · Authors · 2025-11-20
>
> ***“Why are all decoder parameters updated if the learned latent space is already expected to capture much of the variability?” (Q1)***
>
> As clarified in our answer on complexity in (i), we do not update all decoder parameters: to keep the gradient step lightweight, we update only the last layer of the decoder, and we do so at fixed intervals rather than at every MCMC iteration.
>
> To answer why we perform these updates at all:
>
> First, if training does not go perfectly well, the Markov chain at inference would converge to a distribution determined by the initial decoder parameters. Allowing parameter updates lets the model correct for this mismatch and adapt the decoder towards lower test-time cost. This is particularly important when there is a train–test mismatch, for example when training on instances with $n = 100$ and testing on larger instances: the effective distribution over solutions changes, and adapting the parameters helps recover a suitable policy while exploring the latent space. Even though the latent-space search already introduces variability, these parameter updates ensure that this variability is aligned with good solutions.
>
> Second, under a moderate time budget, the Markov chains may get trapped in local modes in latent space under a fixed target distribution (i.e., fixed parameters), which can limit solution quality after a fixed budget. Small, targeted parameter updates help reshape the landscape seen by the chain and can facilitate escaping such local modes, leading to further improvements.
>
>
> ***It would also be helpful to report the dimensionality of the latent space used in the experiments (Q3)***
>
> We now explicitly report the latent dimension used in all experiments, $d_z = 100$, in the Experiments section. Further inference details, including the proposal distribution, are provided in Appendix F.2.1, and we intend to move part of this material into the main text if an extra page is granted upon acceptance.
>
>
> ***“On line 116, the paper states that LGS ‘removes the need for a pre-trained policy.’” (Q5)***
>
> Our intention was to convey that our method can be trained from scratch, without relying on a separately pre-trained policy or a supervised pre-training stage. We acknowledge that the original phrasing was misleading, and we have revised it accordingly in the manuscript.
>
>
> We thank the reviewer once more for their insightful and detailed comments. The revisions above—clarifying the motivation and contributions of our work, adding the requested (and some additional) ablation studies, and revising and explaining the timing protocol—directly address all the points raised and, we believe, further strengthen the manuscript. We hope these changes meet your expectations, and we remain happy to clarify any remaining questions.
>
>
> ### References
>
> [1] Yuri Burda, Roger Grosse, and Ruslan Salakhutdinov. Importance weighted autoencoders. In International Conference on Learning Representations, 2016.
>
> [2] Felix Chalumeau, Refiloe Shabe, Noah De Nicola, Arnu Pretorius, Thomas D Barrett, and Nathan Grinsztajn. Memory-enhanced neural solvers for routing problems. In Advances in Neural Information Processing Systems, 2025.

---

> > ### Comment · Reviewer_eBRY · 2025-11-27
> >
> > I thank the authors for their detailed replies, the additional experiments, and the clarifications made to the manuscript. Several of my initial concerns have been addressed, and I appreciate the effort invested in improving the submission. Below, I summarise the remaining points where clarification would still be appreciated.
> >
> > 1. Evaluation protocol
> >
> > In the rebuttal and in responses to other reviewers, the authors state that the “evaluation protocol follows standard practice in NCO.” I would encourage more precision, as the protocol differs from common setups in a few ways:
> >
> > (i) Dataset choice.
> >
> > Experiments use the CVAE-Opt dataset, whereas most RL-based NCO baselines (AM, POMO, EAS, COMPASS) evaluate on the AM or EAS datasets. This is not an issue, but it should be stated explicitly so readers do not assume direct comparability.
> >
> > (ii) Definition of budget.
> >
> > It now appears the comparison is based on wall-clock time, whereas much prior work uses number of attempts / rollouts. Either choice is fine, but it should be clearly stated in the main text.
> >
> > Given this, I have a few follow-up questions:
> > - Is the time budget applied per instance or for the entire evaluation set?
> > - Are all methods implemented in the same codebase / infrastructure? Otherwise differences in implementation speed could affect results.
> > - How was the time budget chosen? Since it affects performance, a short justification (e.g., alignment with typical budgets or with EAS/COMPASS iterations) would be helpful.
> >
> > Regarding the original POMO timings: if early stopping was used and the budget was time-based, the reported runtime should have always been the smallest, hence the fact that it was reported to be the largest on TSP150 is still unexplained. I appreciate the correction in the revised table; I am simply trying to understand the original setup.
> >
> > Finally, I would encourage avoiding situations where some methods receive a different time budget (e.g., “we allow EAS a slightly larger time budget”). Even if well-intentioned, this complicates reproducibility.
> >
> > >  "Note that this differs from the original COMPASS paper, where EAS was given a larger time budget"
> >
> > In prior work (EAS, COMPASS), runtime differences stemmed from fixing the number of attempts rather than the wall-clock budget. Both conventions are valid, but ensuring clarity and consistency is important.
> >
> > 2. Instance independence / dependence
> >
> > The rebuttal provides results for COMPASS + LGS, the instance-independent model, the instance-conditioned model, and the deterministic-latent variant. These are useful additions, but a few questions remain:
> >
> > (i) Why is COMPASS + LGS restricted to “a small number of latent samples”? LGS could in principle draw as many latent samples as needed. Similarly, it is not obvious why applying LGS on top of COMPASS would require using multiple starting points per latent. These constraints are not inherent to COMPASS, hence could the authors just choose the set-up that enable to make the most of LGS?
> >
> > (ii) The ordering between COMPASS+LGS and the instance-independent model is surprising; any intuition here would help.
> >
> > (iii) The deterministic-latent variant outperforming the instance-independent one (15.572 vs. 15.641) is also unexpected. Intuitively, one would expect deterministic to be outperformed by instance-independent. Can the authors propose an explanation?
> >
> > (iv) My earlier question on the encoder was not only about its implementation but about what structure is actually learned by conditioning the latent space on the instance, compared to using a fixed latent distribution. What behaviour should one expect to emerge?
> >
> > 3. Instance sizes
> >
> > A final question I omitted earlier: why were larger instances (e.g., TSP/CVRP 200 as in EAS and COMPASS) not included? Scaling to larger sizes is a central challenge in RL4CO, and several recent works highlight the importance of evaluating at these scales. Is there a reason why these sizes were not considered in the paper?
> >
> > Overall, while the rebuttal clarifies several of my initial questions, a number of points remain uncertain to me. The comments above highlight the areas where further explanation would be appreciated.

---

> > > ### Author Response · Authors · 2025-12-01
> > >
> > > We thank the reviewer for the very helpful follow-up and for the concrete suggestions on clarifying the experimental protocol and the role of instance dependence. We address each point below and have already incorporated the corresponding clarifications into the revised manuscript.
> > >
> > > **1. Evaluation protocol.**
> > >
> > > ***(i) Dataset choice***
> > >
> > > We agree that our dataset choice for $n=100$ differs from most RL-based NCO papers; however, for evaluating generalization to larger sizes, we use the same datasets as most NCO methods. As mentioned and as you noted, we follow the CVAE-Opt setup and use the dataset of [1], which provides 1000 instances per problem size. We chose this setting in order to work with a more compact benchmark while keeping the same number of instances across sizes and avoiding the larger 10,000-instance datasets. In the revised manuscript, we now state this explicitly in Section 5 and in the discussion of the experimental results.
> > >
> > > ***(ii) Definition of budget***
> > >
> > > We confirm that our comparisons are based on a wall-clock budget rather than on the number of attempts, and that this budget is fixed for the entire evaluation set (not per instance). We chose this because different methods have very different computational costs per attempt, so fixing the number of attempts can lead to substantially different runtimes, which makes direct comparison harder. Using a time budget instead ensures that all methods are compared under a comparable computational effort.
> > >
> > > In our experiments, we fix a total wall-clock budget for evaluating the entire test set at a given problem size and, since all methods are run in mini-batches, this induces a corresponding budget for each batch of instances. All methods are evaluated in the same PyTorch codebase and on the same hardware (a single NVIDIA RTX 6000 GPU), so that infrastructure differences do not bias the comparison.
> > >
> > > To choose the budget, we proceeded as follows. Since our model is based on the CVAE-Opt setting, we first ran Differential Evolution (the inference method used in [1]) with the same number of iterations as in their setup on our model and recorded its runtime. We then used this runtime as a reference and applied the same wall-clock budget to all methods at that problem size. The resulting budgets of 20M for TSP100 and 40M for CVRP100 correspond to roughly 1 second and 2 seconds per instance, respectively, which we found to be a reasonable trade-off between solution quality and computational cost, and therefore kept for all methods.
> > >
> > > Regarding the original POMO sampling timings: the anomalies on TSP125 and TSP150 indeed came from a mistake in an early version of our timing code. For most settings, POMO used a maximum number of iterations together with an early-stopping condition (stop if no improvement), so the effective runtime was often shorter than the maximum. For TSP125 and TSP150, however, the early-stopping condition was accidentally commented out, so POMO always ran for the full number of iterations. This resulted in larger runtimes for these two cases, leading to inconsistencies with the other timing settings, even though the reported objective values themselves were correct (e.g., $8.595$ in 50M vs $8.614$ in 30M for TSP125, and $9.377$ in 1h30 vs $9.406$ in 40M for TSP150). We apologize for this inconsistency and have corrected it in the revised version by enforcing the same time budget as for our method (see Tables 1, 4, and 5).
> > >
> > > For the TSP setting with the augmentation trick, our initial goal was to reproduce the best reported EAS results while also highlighting its weaknesses, not to disadvantage other methods. In the revised manuscript, we will instead enforce an equal wall-clock budget across all methods. Since we are currently running all methods on instances with $n = 200$ as you suggested (see Point 3 below for more details), we will update the corresponding tables once these runs are completed. Finally, we have added a paragraph in Section F.2.2 to explain this choice of budget.

---

> > > ### Author Response · Authors · 2025-12-01
> > >
> > > ***(iv) What structure is actually learned by conditioning on the instance?***
> > >
> > > Conceptually, the instance-conditioned encoder is designed to learn a low-dimensional summary of the problem instance that is most relevant for constructing good solutions. Concretely, it outputs the mean and variance of the latent distribution $p_\phi(z \mid x)$ and is trained to place mass in regions of latent space that tend to decode into low-cost solutions for that particular instance. In other words, the encoder learns how instance-specific features translate into a latent prior that concentrates probability around promising search regions.
> > >
> > > At inference time, conditioning $p_\phi(z \mid x)$ on $x$ allows the MCMC search to start and evolve in an instance-specific region of latent space rather than in a single, shared prior region. This is precisely what our instance-independent ablation suggests: when $p_\phi(z)$ is unconditional, the model loses this adaptation and performance deteriorates.
> > >
> > > A natural direction for future work would be to enforce that similar instances are mapped to nearby regions in latent space, for example by adding a regularizer that encourages the Kullback–Leibler divergence between $p_\phi(z \mid x)$ and $p_\phi(z \mid x')$ to be small whenever the corresponding instances $x$ and $x'$ are close under some predefined metric.
> > >
> > >
> > > **3. Choice of instance sizes.**
> > >
> > > We agree that scaling to larger problem sizes (e.g., TSP/CVRP with $n=200$) is an important aspect in RL4CO, and we appreciate the reminder. Our initial choice to focus on sizes up to 150 nodes was driven by the CVAE-Opt dataset setup, which only provides instances up to $n=150$.
> > >
> > > Due to time constraints during the rebuttal period, we were able to run additional experiments for CVRP with $n=200$ without the augmentation trick; these preliminary results, reported below, indicate that our method remains competitive at this scale. We are currently running further experiments for TSP and for settings with the augmentation trick, and we will include these additional results in the revised manuscript upon acceptance. The current $n=200$ results are reported in Section F.2.2, and we plan to move them into Tables 1, 4, and 5 once all experiments are completed.
> > >
> > > ***Table: Experimental results on CVRP with $n = 200$ without the augmentation trick***
> > >
> > > | Method             |    Obj.    |     Gap      |  Time  |
> > > |--------------------|-----------:|------------:|:------:|
> > > | LKH3               |   22.00    |    0.00%    |  25H   |
> > > | POMO (greedy)      |   23.296   |    5.891%   |   1M   |
> > > | POMO (sampling)    |   23.371   |    6.232%   |  2H10  |
> > > | EAS                |   22.361   |    1.643%   |  2H10  |
> > > | COMPASS            |   22.455   |    2.069%   |  2H10  |
> > > | ELG                |   22.460   |    2.092%   |  2H10  |
> > > | CNF                |   23.389   |    6.314%   |  2H10  |
> > > | LGS-Net (ours)     | **22.284** | **1.289%**  |  2H10  |
> > >
> > > We thank the reviewer again for this careful follow-up. The additional clarifications you raised—on the evaluation protocol, the role of instance conditioning, and the additional experiments at instance size $n=200$—have led to concrete improvements in the presentation and positioning of the paper. We hope these clarifications and changes address your remaining concerns, and we remain happy to clarify any further questions.
> > >
> > >
> > > ### References
> > >
> > > [1] André Hottung, Bhanu Bhandari, and Kevin Tierney. Learning a latent search space for routing problems using variational autoencoders. In International Conference on Learning Representations, 2021.

---

> ### Author Response · Authors · 2025-12-01
>
> **2. Instance independence vs dependence.**
>
> We appreciate the opportunity to clarify the design choices and the ablations.
>
> ***(i) Why small $K$ and multiple starting points for COMPASS+LGS?***
>
> Conceptually, LGS does not require a small number of latent samples nor multiple starting points per latent sample, and in principle could be combined with COMPASS using a larger $K$ and a single starting point. In our COMPASS+LGS ablation, we initially adopted the original COMPASS configuration (small $K$ with multiple starting points), because this is the regime in which the model was trained and reported to perform best. We also conducted experiments with a variant that uses a single starting point per latent and increases $K$, but this configuration performed worse (with a gap of 0.459\%) than the original COMPASS setup when combined with LGS. Our interpretation is that the COMPASS model has learned to exploit the specific coupling between latent samples and multiple starting tours used during training, so changing this structure at inference time without retraining degrades performance. A proper evaluation of alternative configurations (e.g., a single starting point with larger $K$) would require retraining COMPASS under those settings, which was unfortunately beyond the available time and computational budget for the rebuttal.
>
> ***(ii) Ordering between COMPASS+LGS and the instance-independent model.***
>
> We believe this behaviour can be explained by how the models are trained. COMPASS learns its latent space on top of a strong pre-trained policy, whose parameters have already been shaped to produce good solutions. Even though the latent distribution is instance-independent, the underlying policy already encodes rich problem-dependent structure that the inference procedure can partially exploit. By contrast, our instance-independent variant used in the ablations must learn both the policy and the latent space jointly from scratch via RL, without instance conditioning, which appears more challenging under the same training budget and leads to weaker performance. Overall, this comparison supports our main claim: conditioning the latent distribution on the instance $x$ provides a more informative prior for the search and yields measurable gains over both COMPASS+LGS and the instance-independent variant, in line with the intuition that an instance-dependent latent space better captures problem-specific structure.
>
>
> ***(iii) Deterministic latent outperforming the instance-independent model.***
>
> It is indeed somewhat surprising that the deterministic latent variant outperforms the instance-independent model. However, note that the deterministic latent representation is still instance-conditioned through $z = f_\phi(x)$ and is built on top of a well-trained POMO model. Although this collapses the latent space to a single point per instance, it provides a strong, geometry-aware representation adapted to each input.
>
> This yields a good starting point for greedy decoding and, when combined with light exploration in latent space via an adapted decoder, can produce better solutions than an instance-independent model that must share a single latent prior across all instances. What the deterministic variant lacks is the ability to explore multiple promising latent regions for a fixed instance, which is precisely where the stochastic latent model gains an advantage under a larger inference budget (see Table 8).
>
> We now add a short clarification in the appendix to make this distinction explicit: deterministic latent variables provide strong instance-specific summaries but limit exploration, whereas stochastic latents enable exploration of multiple modes per instance.

---

### Official Review · Reviewer_4pVj · 2025-11-01

**Soundness:** 3
**Presentation:** 4
**Contribution:** 3
**Rating:** 6
**Confidence:** 3

**Summary:**

LGS-Net is an RL-based latent variable neural combinatorial optimization model that requires no labels. It uses instance-conditioned latent embeddings and trains via a cost-weighted, entropy-regularized objective function. The key innovation is the Latent Guided Sampling (LGS) inference procedure, which updates decoder parameters through stochastic approximation while running interacting MCMC chains in latent space. It provides convergence guarantees for both fixed and adaptive parameters. The model achieves learning-based SOTA on TSP and CVRP benchmarks, and even outperforms LKH3 on some CVRP settings.

**Strengths:**

The instance-conditioned latent model eliminates the need for pre-computed solutions or pre-trained policies unlike CVAE-Opt and COMPASS, and demonstrates clear performance improvements. LGS inference combines interacting MCMC with real-time parameter updates, consistently outperforming all alternatives including DE, CMA-ES, active search, and gradient-based finetuning. The convergence proof for time-inhomogeneous Markov chains is a non-trivial theoretical contribution for this problem class. The experiments include multiple datasets, comparisons with and without augmentation, runtime reports, and detailed hyperparameters, demonstrating excellent reproducibility.

**Weaknesses:**

The adaptive chain convergence relies on Assumption 4, but the paper only provides high-level justification. There is a lack of sufficient conditions linking Algorithm 1's specific step-size choices and gradient-variance conditions, creating a gap between theory and implementation.
The empirical study is limited to Euclidean routing (TSP/CVRP), restricting the demonstration of generality. Experiments on non-Euclidean problems or other combinatorial domains would help establish the versatility of the latent formulation.
Although negative gaps versus LKH3 on CVRP are reported, verification via exact solvers or optimality certificates is absent, making the evaluation protocol ambiguous.

**Questions:**

What is the sensitivity to the choice of latent dimension d_z and bounded diameter R? The ablations only vary K and update schedules, not d_z or R.

---

> ### Author Response · Authors · 2025-11-20
>
> We thank the reviewer for the positive and thoughtful feedback on the paper, in particular for recognizing both the theoretical contributions and the empirical strength of the approach. We address each of the raised points below and have incorporated the corresponding clarifications and additions into the revised manuscript.
>
> **Assumption 4 and link with implementation (W1).**
>
> The connection between Assumption 4 and the practical choice of step sizes can indeed be made more explicit. In Appendix D.3, we provide sufficient conditions (smoothness of the test loss, a bounded variance gradient estimator, and a decreasing step-size schedule) under which Assumption 4 holds, following standard results from Stochastic Approximation theory. In our setting, verifying Assumption 4 essentially reduces to jointly controlling the variance of the gradient estimator (typically of order $\mathcal{O}(1/K)$) and the step-size sequence $(\gamma_m)_{m \geq 0}$. The influence of $K$ on performance, and thus on the variance of this estimator, is illustrated empirically in Figure 6.
>
> Algorithm 1 is written in a way that is fully compatible with these conditions (plain SA updates with step sizes $\gamma_m$). In practice, however, updating all model parameters at every MCMC iteration would be computationally expensive and would not allow the chains to sufficiently explore the latent space between updates. For this reason, the implementation performs parameter updates at fixed intervals and restricts updates to the last layer of the decoder, for which backpropagation is significantly cheaper, keeping our method efficient and avoiding excessive MCMC+SA overhead. Figure 8 in the revised manuscript reports the effect of varying the update interval, showing that updating parameters at fixed intervals leads to better performance.
>
> If a larger computational budget is available, one can move closer to the ideal SA setting considered in the theory by updating the parameters at every iteration with a decaying step-size schedule, as in standard SA.
>
> **Generality beyond routing problems (W2).**
>
> Regarding the concern about generality beyond routing problems, the experiments in the paper focus on TSP and CVRP because these are the dominant benchmarks in the Neural Combinatorial Optimization literature and allow direct comparison with prior work. Conceptually, however, the latent formulation and LGS inference are not specific to routing: the method only requires a cost function and a decoder defining a stochastic policy over feasible solutions. Moreover, the convergence analysis is fully general in the sense that it is agnostic to the policy architecture and to the choice of inference parameters.
>
> We also conduct additional experiments on the 0-1 Knapsack problem, following the setup of [1]. Due to time constraints and the importance of a careful training protocol, these experiments compare our inference method against a simple sampling baseline rather than retraining all NCO baselines. We additionally include the OR-Tools solver and the optimal solution obtained via dynamic programming. On this problem, our inference method recovers optimal solutions on the test instances, matching the optimal values, whereas the sampling method using our model also gives good results but does not reach optimality. The results are reported in the table below and are also added in Appendix F.2.4 in the revised manuscript, together with details on the problem, model, and dataset used for these experiments. This provides further evidence that the proposed latent formulation and LGS inference extend beyond routing problems.
>
> *Table: Experimental results on knapsack instances (average objective value; higher is better).*
>
> | Method        |   Sampling   |   LGS (ours)   |   OR-Tools   |   Optimal   |
> |--------------|:------------:|:--------------:|:------------:|:-----------:|
> | Knapsack50   |  20.012      |  **20.018**    |  **20.018**  |  **20.018** |
> | Knapsack100  |  40.501      |  **40.512**    |  **40.512**  |  **40.512** |

---

> ### Author Response · Authors · 2025-11-20
>
> **Evaluation protocol and negative gaps vs LKH3 (W3).**
>
> The evaluation protocol follows standard practice in the NCO literature. For TSP, Concorde is used as the reference (exact) solver, so the reported “gap” is an actual optimality gap. For CVRP, LKH3 is used as a strong heuristic baseline and the gap is defined as in Eq. (16) with respect to the LKH3 solution, which is the common choice in prior work. Since running exact solvers at scale is computationally prohibitive, prior work also relies on heuristic methods such as LKH3 (e.g., EAS, COMPASS), for which negative gaps are frequently reported. In this setting, negative gaps on CVRP indicate that LGS-Net finds solutions of lower cost than LKH3, whereas no negative gaps arise for TSP because the reference solutions are exact. This interpretation is currently detailed in Appendix F.2 and will be stated explicitly in the main text to avoid any ambiguity. For knapsack, where exact solutions are cheap to compute, we additionally report optimal values as shown above.
>
>
> **Sensitivity to latent dimension $d_z$ and radius $R$ (Q1).**
>
> We appreciate this question. In our paper, we fixed $d_z = 100$ and $R = 40$ for all experiments (Appendix F.2.1) to avoid introducing too many degrees of freedom. As suggested, we performed an ablation on the radius $R$ of the latent space; the results are now reported in Figure 7 of the revised manuscript, together with a short discussion. We observe that the model performance degrades slightly for small values of $R$, since the latent space is too constrained, and also deteriorates for very large values of $R$, as the space becomes harder to exploit effectively. In contrast, moderate values of $R$ strike a good balance and yield the best empirical performance. Nevertheless, across all tested values of $R$, the performance remains competitive.
>
> Regarding the latent dimension, we already trained a model with $d_z = 2$ to illustrate the latent space search in Figure 3, and, as expected, this very low-dimensional model performs noticeably worse on the test instances (gap of 4.176\% with objective value $16.189$), since such a small latent space cannot capture the structure of realistic routing problems. We are currently running additional experiments with $d_z = 50$ and $d_z = 150$ under the same protocol, and we will include the corresponding results in the appendix once these runs are completed.
>
> We hope that our revisions and responses sufficiently address your concerns. We thank you again for your time and effort in reviewing our work, and we remain open to further feedback.
>
>
> ### References
>
> [1] Irwan Bello, Hieu Pham, Quoc V. Le, Mohammad Norouzi, and Samy Bengio. Neural combinatorial optimization with reinforcement learning. In International Conference on Learning Representations, Workshop Track, 2017.

---

> ### Author Response · Authors · 2025-12-01
>
> **Follow-up on Q1: sensitivity to $d_z$**
>
> As a follow-up to our earlier response, we have now completed the additional experiments with latent dimensions $d_z = 50$ and $d_z = 150$ under the same protocol as for $d_z = 100$. The results are reported below and also in Table 11 of the revised manuscript.
>
> ***Table: Effect of latent dimension $d_z$ in our method on CVRP with $n = 100$***
>
> |      Method       |    Obj.     |      Gap        |
> |-------------------|------------:|----------------:|
> | $d_z = 2$         |   16.189    |  4.176\%      |
> | $d_z = 50$        |   15.529    |  -0.071\%     |
> | $d_z = 100$       | **15.524**  | -**0.102\%**  |
> | $d_z = 150$       |   15.543    |  0.019\%      |
>
>
> As expected, the very low-dimensional model with $d_z = 2$ performs noticeably worse, since such a small latent space cannot capture the structure of routing problems. For $d_z = 50$, the model performs competitively but slightly worse than $d_z = 100$. Increasing the dimension further to $d_z = 150$ does not bring additional gains: performance degrades marginally, likely because the higher-dimensional latent space is harder to explore efficiently under a fixed inference budget.
>
> We have updated the corresponding paragraph in the revised manuscript to reflect these results. We thank the reviewer again for suggesting this sensitivity analysis.

---

### Author Response · Authors · 2025-12-02
**Summary of Rebuttal and Manuscript Revisions**

We thank the reviewers and the program committee for their time and constructive feedback. Below, we briefly clarify our contributions and summarize the main changes and additional experiments conducted during the rebuttal period.

**Main contributions.**

Our contributions have two parallel components:

- ***Model and inference:*** We propose a latent space model trained via RL that learns a structured, instance-conditioned latent representation for NCO. On top of this model, we introduce an interacting MCMC+SA latent space inference scheme that avoids full-network backpropagation at every step, making it more efficient than fully gradient-based methods in low-budget regimes.

- ***Theory:*** We provide, to the best of our knowledge, the first convergence-rate result for a time-inhomogeneous MCMC whose target and proposal distributions evolve via SA (Theorems D.1 and 5.3). These results apply directly to our adaptive inference method.

**Changes in the revised manuscript.**

***Clarifying motivation and the role of the latent space.***
Following the comments of reviewers eBRY and TFKR, we now explicitly illustrate the effect of using a stochastic, instance-dependent latent space. We added two ablation studies: Table 7 compares our instance-conditioned latent model to an instance-independent variant, and Table 8 compares deterministic and stochastic latent variables. These results show that (i) conditioning the latent distribution on the instance improves over an instance-independent latent model, and (ii) a stochastic latent space further improves over a deterministic latent representation. The additional questions raised by reviewer eBRY on these ablations are addressed in our final rebuttal response. We also compare our cost-weighted loss (explained in Appendix A.3) with the POPPY loss used in COMPASS in Table 6.

***Complexity of MCMC + SA.***
As highlighted by reviewers BuX2 and TFKR, we have clarified in the revised manuscript how we control the overhead of combining MCMC and SA. The main potential cost comes from performing gradient steps at every iteration. To keep the inference overhead moderate, we update only the last layer of the decoder and do so at fixed intervals rather than at every step. This avoids full-network backpropagation and keeps the SA component lightweight, while also allowing the chains to explore the latent space under fixed parameters. Figure 8 reports the effect of varying the update interval. In addition, Table 2 shows that methods such as SGLD and Adam, which compute gradients in latent space at every step, perform only moderately compared to our method, due to this additional computational overhead.

***Hyperparameter and sensitivity analyses.***
As suggested by reviewer 4pVj, we conducted additional sensitivity analyses on key latent-space hyperparameters. We study the effect of the latent radius and latent dimension, and report the corresponding results in Figure 7 and Table 11.

***Evaluation protocol.***
We added a dedicated paragraph in the revised manuscript explaining how we choose the inference-time budget and how we keep this budget comparable across all methods to ensure a fair comparison. As requested by reviewer eBRY, we also corrected the timing inconsistencies for POMO sampling in Tables 1, 4, and 5; all methods are now evaluated under a similar time budget as our approach.

***Extension to larger instance sizes.***
In response to reviewer eBRY’s rebuttal comments, we conducted additional experiments on CVRP with $n = 200$. The results, reported in Table 10, show that our method remains competitive at this scale compared to other methods. We are currently running analogous experiments for TSP, and we plan to include these additional results in a future version of the manuscript.

***Generalization to other problems.***
Finally, as requested by several reviewers, we demonstrate that our method can be applied beyond routing problems by reporting experiments on 0–1 knapsack instances in Section F.2.4.

In summary, reviewers 4pVj and BuX2 acknowledged the value of the work, and we have addressed their concerns in the rebuttal. Reviewer TFKR raised thoughtful questions about the role of the latent space and the use of a stochastic latent representation; we responded with targeted ablations and clarifications, after which they increased their score. Finally, we engaged in a detailed discussion with reviewer eBRY, who expressed appreciation for the work but raised several detailed questions—particularly about the evaluation protocol and instance dependence—which we directly addressed in our final response.

We thank the area chair and senior area chair for considering our work and for overseeing the review process under the unusual circumstances this year. We hope that the clarifications, additional experiments, and revisions above convey the robustness and scope of our contribution. Thank you very much for your time and consideration.

Sincerely,

The authors

---

### Meta-Review · Area_Chair_KePp · 2025-12-24

**Summary:**

Initially, the reviewers are concerned about 1) the insufficient conditions linking the specific step-size choices and gradient-variance conditions, 2) limitation to Euclidean VRPs, 3) missing ablation studies, 4) longer inference time, 5) small problem size, 6) generalization to non-routing problems.

The authors made massive rebuttal, which are acknowledged by most reviewers. However, from my perspective, I feel surprised no reviewers are concerned about the small problem size even if the authors raise it up to 200 nodes. In the literature of NCO for routing problems, 200 nodes are really small, where 500+ or thousands of nodes would be regular. Moreover, the authors only test KP as other COPs, which is quite simple and similar to routing, different from the scheduling problem suggested by reviewers. Lastly, the inference time is really long compared with many NCO baselines (which were not included in the experiment). To summarize, I do believe it is a beautiful paper with theoretic proof, but suffering from implicit yet critical limitations.

**Reviewer Concerns:**

From the reviewer perspective, I guess they may acknowledge all the concerns are addressed. But from my perspective, some concerns  are critical and are still outstanding as mentioned above, although they are not raised by reviewers.

**Reviewer Scores:**

From what I read and observe, I guess the final score might change to 6,6,6,6 from 4, 4, 6, 6. However, positive scores do not always mean acceptance, which happened to this submission in my opinion.

---

### Decision · Program_Chairs · 2026-01-26

Reject